# TMEM251 loss-induced autophagy dysfunction in the anterior cingulate cortex contributes to chronic postoperative pain

Yaowei Xu [iD] [1,2,5], Fei Xing [1,2,5], Xin Wei [1], Xiaoling Wang [2,3], Xiaoshan Shi [1,2], Zhongyu Wang [1], Na Xing [1], Jingjing Yuan [1], Zhisong Li [iD] [2,4 ✉] & Wei Zhang [iD] [1,2 ✉]

## Abstract

Macroautophagy/autophagy plays a crucial role in maintaining nervous system homeostasis but its role in chronic postoperative pain (CPOP) remains poorly understood. Here, we identify impaired autophagy and the accumulation of synaptic proteins in the anterior cingulate cortex (ACC) during the maintenance of CPOP after skin/muscle incision and retraction (SMIR). Lysosomal hydrolase levels are reduced upon SMIR, accompanied by a deficiency of the lysosomal trafficking protein transmembrane protein 251 (TMEM251, also named LYSET). TMEM251 overexpression alleviates impaired autophagy, accumulation of synaptic proteins within autophagy substrates, and maintenance of CPOP in SMIR mice. Conversely, TMEM251 knockdown induces autophagy impairment, accumulation of synaptic proteins, and chronic pain phenotypes in naive mice. Autophagy dysfunction is most pronounced in CaMKIIα-positive neurons in the ACC post-surgery, resulting in their activation, which is mitigated by TMEM251 overexpression. Chemogenetic activation of CaMKIIα neurons exacerbates autophagy impairment and CPOP, while their inhibition rescues SMIR-induced autophagy and pain phenotypes. Taken together, our study highlights the close relationship between impaired autophagy and neuronal activation in the promotion of chronic postoperative pain.

**Keywords** Anterior Cingulate Cortex; Autophagy; Chronic Postoperative Pain; Synaptic Plasticity; Transmembrane Protein 251
**Subject Categories** Autophagy & Cell Death; Molecular Biology of Disease; Neuroscience

## Introduction

Chronic postoperative pain (CPOP) lasts beyond the normal wound healing period (≥3 months), and represents a significant clinical challenge in modern surgical practice (Fregoso et al, 2019).

Epidemiological studies indicate that more than 320 million surgical procedures are performed annually worldwide, and ~10% of patients develop CPOP, with incidence rates increasing to 48–85% following major surgeries such as cardiothoracic procedures and amputations (Wylde et al, 2017). Current therapeutic strategies, primarily relying on opioid analgesics and nonsteroidal anti-inflammatory drugs, are limited by their adverse effects and incomplete efficacy (Rikard et al, 2023), underscoring the urgent need for novel mechanistic insights and therapeutic targets.

Central sensitization, characterized by maladaptive neuroplastic changes in higher brain regions, has emerged as a critical mechanism in CPOP pathogenesis (Treede, 2023). Among numerous brain regions, the anterior cingulate cortex (ACC), a key component of the limbic system, plays a dual role in emotional functions and pain modulation (Alejandro and Holroyd, 2024; Guo et al, 2024). Neurophysiological studies have demonstrated that ACC hyperexcitability facilitates nociceptive signal transmission (Bliss et al, 2016), with preclinical models revealing an excitation/inhibition imbalance in ACC neurons during inflammatory pain (Wei et al, 2024). In addition to its sensory dimensions, experimental evidence indicates that ACC-mediated mechanisms are crucial for pain-related negative emotional states (Becker et al, 2023). However, its specific contributions to CPOP remain poorly understood.

Autophagy, an evolutionarily conserved lysosomal degradation pathway, serves as a fundamental homeostatic mechanism for maintaining brain physiology (Nixon and Rubinsztein, 2024). This intracellular process has gained increasing recognition for its essential roles in neuronal function, particularly in regulating synaptic plasticity and modulating neurotransmitter dynamics (Fleming et al, 2022; Kim et al, 2017). Previous study has demonstrated impaired autophagy predominantly in astrocytes, during the maintenance phase of neuropathic pain (Li et al, 2021a). The lysosomal system serves as the terminal effector of autophagy, and its function is critically dependent on proper lysosomal enzyme trafficking (Holland et al, 2020). Lysosomal trafficking protein Transmembrane protein 251 (TMEM251), a Golgi-associated protein, has been identified as a key regulator of lysosomal enzyme

[1]Department of Anesthesiology, Pain and Perioperative Medicine, The First Affiliated Hospital of Zhengzhou University, Zhengzhou, Henan, China. [2]Neuroscience Research Institute, Zhengzhou University Academy of Medical Sciences, Zhengzhou, Henan, China. [3]Department of Anesthesiology and Perioperative Medicine, People's Hospital of Zhengzhou University, Henan Provincial People's Hospital, Zhengzhou, Henan, China. [4]Department of Anesthesiology and Perioperative Medicine, The Second Affiliated Hospital of Zhengzhou University, Zhengzhou, Henan, China. [5]These authors contributed equally: Yaowei Xu, Fei Xing. ✉E-mail: lzszd@126.com; zhangw571012@126.com

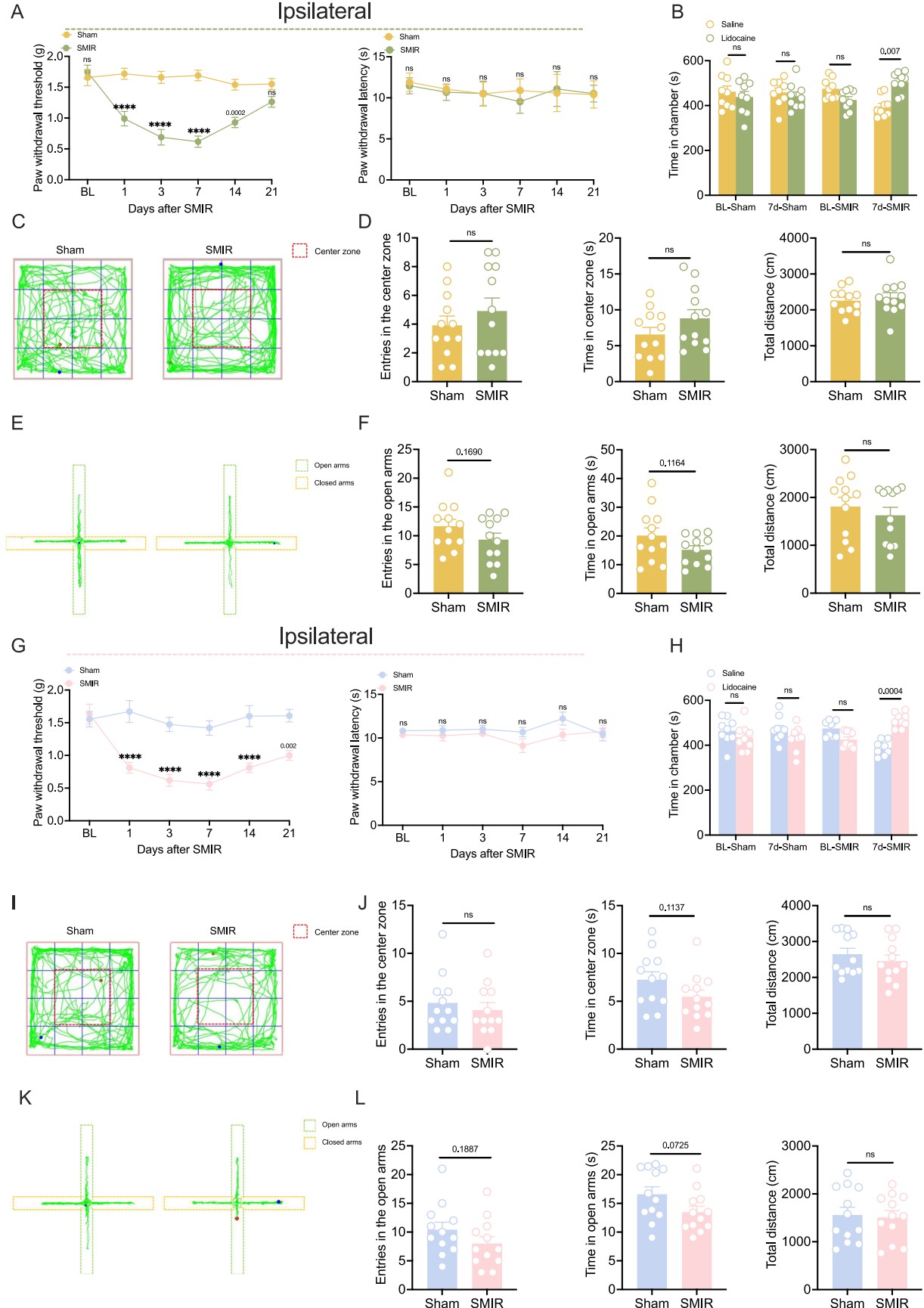

**Figure 1. Persistent postoperative pain in SMIR mice.**

(A) PWT (left) and PWL (right) of the ipsilateral paws in the sham and SMIR groups of male mice. $n = 12$ per group; ****$P < 0.0001$ vs. the sham group; two-way repeated-measures ANOVA with Bonferroni post hoc correction. (B) The CPP analysis showing increased lidocaine-paired chamber residence time in male SMIR mice at POD7. $n = 9$ per group; multiple paired $t$ tests with Bonferroni post hoc correction. (C, D) Representative movement trajectories (C) and quantitative analysis (D) of male mice in the OFT. $n = 12$ per group. Two-tailed unpaired Student's $t$ tests. (E, F) Representative motion trajectories (E) and quantitative analysis (F) of male mice in the EPM. $n = 12$ per group. Two-tailed unpaired Student's $t$ tests. (G) PWT (left) and PWL (right) of the ipsilateral paws in the sham and SMIR groups of female mice. $n = 8$ per group; ****$P < 0.0001$ vs. the sham group; two-way repeated-measures ANOVA with Bonferroni post hoc correction. (H) The CPP analysis showing increased lidocaine-paired chamber residence time in female SMIR mice at POD7. $n = 9$ per group; multiple paired $t$ tests with Bonferroni post hoc correction. (I, J) Representative motion trajectories (I) and quantitative analysis (J) of female mice in the OFT. $n = 12$ per group. Two-tailed unpaired Student's $t$ tests. (K, L) Representative motion trajectories (K) and quantitative analysis (L) of female mice in the EPM. $n = 12$ per group. Two-tailed unpaired Student's $t$ tests. All data are presented as the mean ± SEM. Source data are available online for this figure.

transport through the mannose-6-phosphate (M6P) pathway (Pechincha et al, 2022; Qiao et al, 2023). TMEM251 deficiency disrupts M6P-dependent lysosomal enzyme targeting, leading to impaired lysosomal function and autophagy (Zhang et al, 2023; Zhang et al, 2022b). While the role of TMEM251 in autophagy has been established, its specific involvement in the context of CPOP remains unexplored.

In this study, we revealed impaired autophagy in the ACC, accompanied by the accumulation of synaptic proteins in autophagy substrates, during the maintenance of CPOP. We further demonstrated that TMEM251 deficiency underlay this autophagy impairment. Notably, autophagy dysfunction specifically activated CaMKIIα-positive neurons in the ACC. Moreover, sustained activation of CaMKIIα-positive neurons also induced autophagy damage. Hence, these findings not only advance our understanding of the relationship among autophagy, neuronal sensitization and CPOP pathogenesis, but also highlight autophagy regulation as a promising therapeutic target for postoperative pain management.

## Results

### The mice exhibit chronic mechanical allodynia after SMIR surgery

In this study, we employed the skin/muscle incision and retraction (SMIR) model to establish chronic postoperative pain, mirroring clinical surgical conditions. Appendix Fig. S1A,B illustrated the detailed experimental timeline and surgical protocol. The behavioral assessments focused on male mice were firstly implemented. Specifically, paw withdrawal threshold (PWT) demonstrated a marked reduction in ipsilateral paws of SMIR mice at postoperative day 1 (POD1), peaking at POD7, and persisting through POD14 (Fig. 1A). No significant hypersensitivity was observed in the contralateral hindpaws of SMIR mice (Appendix Fig. S1C). Notably, paw withdrawal latency (PWL) showed no significant alterations in either ipsilateral or contralateral paws following SMIR surgery (Fig. 1A; Appendix Fig. S1C). To assess affective-motivational pain behaviors, we conducted conditioned place preference (CPP) test at the peak of evoked pain response (POD7). The CPP analysis revealed that SMIR mice exhibited significantly prolonged dwelling time in the lidocaine-paired chamber, compared to the saline-paired chamber (Fig. 1B). Furthermore, the rotarod test was used to evaluate the motor ability of mice at POD7. The results revealed that there was no

significant difference in the latency to fall between SMIR group and sham group (Appendix Fig. S1D). Given the multidimensional nature of pain, we further evaluated anxiety-like behaviors in SMIR mice. Both open field test (OFT) and elevated plus maze (EPM) analyses indicated that SMIR surgery did not significantly affect anxiety-like behaviors in male mice (Fig. 1C–F).

Consistent with previous reports demonstrating sex-dependent differences in pain perception (Bartley and Fillingim, 2013), we extended our investigation to evaluate the effects of SMIR surgery on female mice. Similar to our observations in male mice, SMIR surgery significantly reduced the PWT in the ipsilateral paws, but not the contralateral paws of female mice (Fig. 1G; Appendix Fig. S1E). Notably, no significant alterations were detected in the PWL on either side (Fig. 1G; Appendix Fig. S1E). However, comparative analysis revealed that female mice maintained significantly reduced PWT in the ipsilateral paws at POD21, a time point at which male mice showed recovery (Fig. 1A,G). Behavioral assessment using the CPP test at POD7 confirmed the presence of affective-motivational pain behaviors in female mice (Fig. 1H). The rotarod tests revealed that SMIR surgery did not affect the latency to fall of female mice (Appendix Fig. S1F). Moreover, OFT and EPM analyses showed SMIR surgery also had no significant effect on anxiety-like behavior in female mice (Fig. 1I–L). Collectively, these findings demonstrate that SMIR surgery induces chronic postoperative pain, characterized primarily by mechanical allodynia in both male and female mice, with subtle sex-specific differences in temporal progression. Importantly, SMIR surgery has no significant impact on anxiety-like behaviors or motor functions in either sex.

### Autophagy is impaired and mainly exists in CaMKIIα-positive neurons during the maintenance of CPOP

To investigate the role of autophagy in CPOP, we analyzed key cortical and subcortical brain regions, including the somatosensory cortex (S1), insular cortex (IC), ACC, nucleus accumbens (NAc), hippocampus (Hip), and basolateral amygdala (BLA). Autophagy activity was assessed via the following established molecular markers: autophagy-related 7/5 (ATG7/5), sequestosome 1 (SQSTM1) and PE-conjugated microtubule-associated protein 1 light chain 3 (LC3B). Immunoblotting analysis revealed significant upregulation of ATG7, ATG5, SQSTM1, and LC3B-II in the ACC of both female and male mice following SMIR surgery (Appendix Fig. S2A–D). This consistent autophagy response across sexes prompted us to focus on the ACC for subsequent investigations. Further temporal analysis demonstrated that SMIR surgery induced

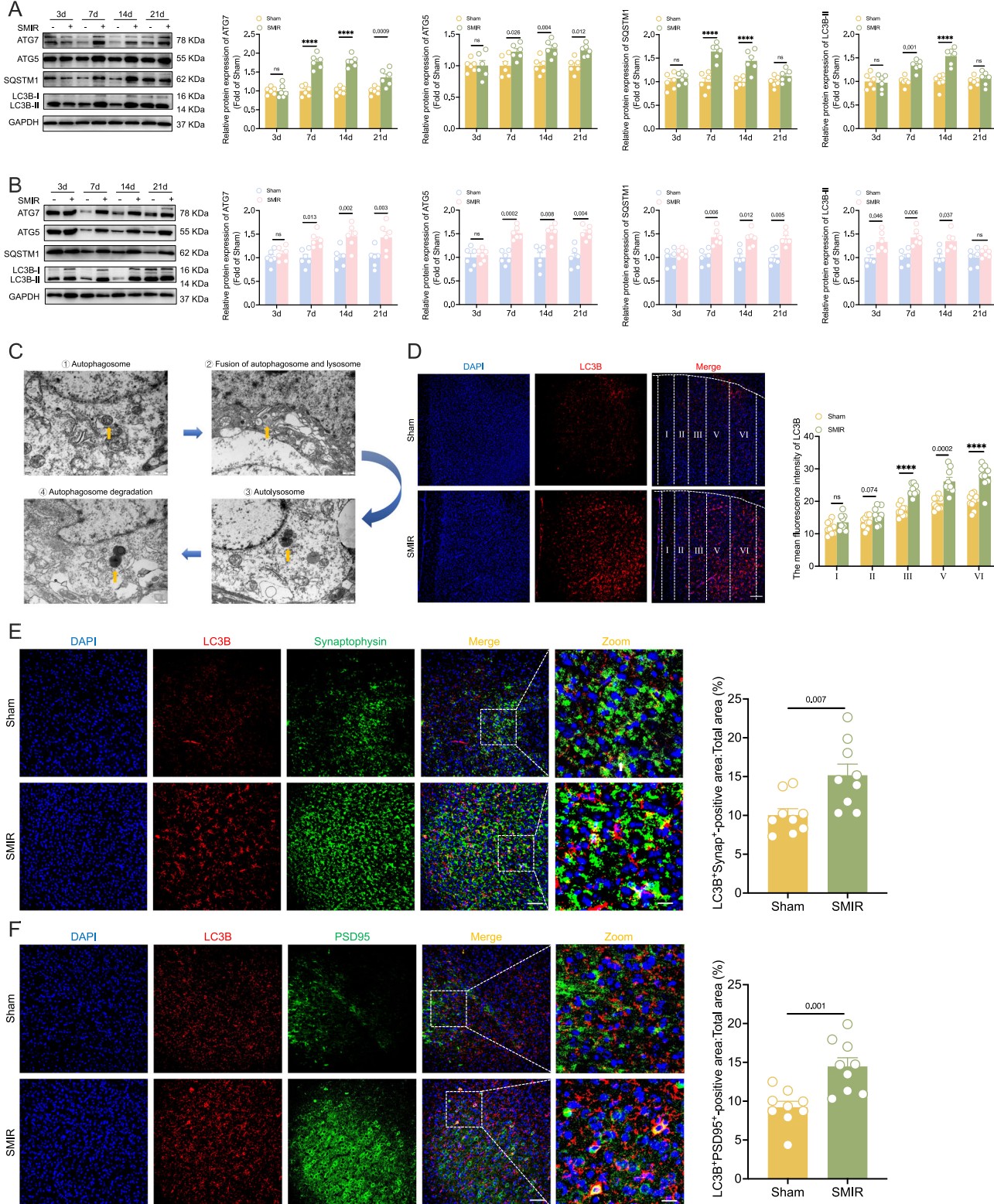

**Figure 2.   Increased autophagy levels in the ACC of SMIR mice.**

(A) Expression levels of autophagy-related proteins (ATG7, ATG5, SQSTM1, and LC3B) at POD3, POD7, POD14 and POD21 of male mice. $n = 6$ per group; ****$P < 0.0001$ vs. the sham group; two-way repeated-measures ANOVA with Bonferroni post hoc correction. (B) Expression levels of autophagy-related proteins (ATG7, ATG5, SQSTM1, and LC3B) at POD3, POD7, POD14 and POD21 of female mice. $n = 6$ per group; two-way repeated-measures ANOVA with Bonferroni post hoc correction. (C) TEM images illustrating the progression of autophagy (yellow arrows). Scale bar: 0.5 μm. (D) SMIR mice showed a significant increase in LC3B, mainly concentrated in the III, V, and VI layers of the ACC. Representative immunofluorescence images (left) and statistical analysis (right) are shown. The white dashed lines are the boundary of the ACC brain region. $n = 9$ per group; ****$P < 0.0001$ vs. the sham group; multiple unpaired $t$ test with Bonferroni post hoc correction. Scale bars: 200 μm. (E) Representative immunofluorescence images (left) and statistical analysis (right) of LC3B colocalization with synaptophysin in the ACC. $n = 9$ per group; two-tailed unpaired Student's $t$ tests. Scale bars: 100 μm and 20 μm. (F) Representative immunofluorescence images (left) and statistical analysis (right) of the colocalization of LC3B with PSD95 in the ACC. $n = 9$ per group; two-tailed unpaired Student's $t$ tests. Scale bars: 100 μm and 20 μm. All data are presented as the mean ± SEM. Source data are available online for this figure.

the upregulation of these autophagy markers in the ACC of male and female mice from POD7 to POD21 (Fig. 2A,B). Given the similar autophagy dynamics in the ACC of both sexes, we selected male mice for subsequent experiments to minimize animal usage while maintaining experimental rigor. Ultrastructural analysis using transmission electron microscopy (TEM) confirmed characteristic autophagy structures (Fig. 2C). Furthermore, immunofluorescence analysis revealed that increased LC3B was mainly concentrated in layers III, V, and VI of ACC in the SMIR group (Fig. 2D). Importantly, we observed a significant increase in the colocalization signal of LC3B and synaptophysin in the SMIR group (Fig. 2E). SMIR surgery increased the expression of postsynaptic density protein-95 (PSD95) and vesicular glutamate transporter 1 (VGLUT1) in the ACC, but did not affect the expression of Gephyrin and vesicular GABA transporter (VGAT) (Appendix Fig. S2E). Consistently, the colocalization signals of LC3B and PSD95, rather than those of Gephyrin, significantly increased in the SMIR group (Fig. 2F; Appendix Fig. S2G).

During the maintenance of CPOP, the expression of SQSTM1 in the ACC of male and female mice significantly increased (Fig. 2A,B), indicating that autophagy flux may be impaired. We further performed immunofluorescence co-staining for SQSTM1 and lysosomal-associated membrane protein 1 (LAMP1). Our results revealed increased aggregation of SQSTM1 in lysosomes of SMIR mice (Fig. 3A). Quantitative immunofluorescence analysis demonstrated that ~74.77% of SQSTM1 aggregates were localized to neurons, while 10.11% and 9.21% were found in astrocytes and microglia, respectively (Fig. 3B; Appendix Fig. S2F). Notably, within the ACC of SMIR mice, SQSTM1 predominantly accumulated in calcium/calmodulin-dependent protein kinase II (CaMKII)-positive neurons, with minimal colocalization observed in glutamate decarboxylase 67 (GAD67)-positive neurons (Fig. 3C,D). TEM analysis also revealed a significant increase in the number of autophagosomes and a marked reduction in the number of autolysosomes in the SMIR group (Fig. 3E). To further quantify autophagy flux, we utilized the pLenti-EGFP-mCherry-LC3B reporter system (Fig. 3F,G). In the ACC of SMIR mice, we observed increased aggregation of EGFP-mCherry-LC3B (Fig. 3H). Notably, similar to SQSTM1 accumulation, merge-positive signals in CaMKIIα-positive neurons also significantly increased (Fig. 3I,J). Collectively, these findings demonstrate that autophagy flux in the ACC is impaired during the maintenance of CPOP. Notably, this effect is particularly pronounced in CaMKIIα-positive neurons. Furthermore, synaptic proteins, particularly the excitatory postsynaptic protein PSD-95, appear to be one of these accumulated autophagy substrates.

## The decrease in hydrolases in lysosomes is accompanied by decreased TMEM251 expression in SMIR mice

To investigate the causal relationship between impaired autophagy and CPOP, we employed bafilomycin A1 (BafA1), a specific inhibitor of lysosomal vacuolar-ATPase (Patergnani et al, 2024). The blocking effect of BafA1 on autophagy flux was validated in cellular experiments (Appendix Fig. S3A). Microinfusion cannulas were surgically implanted into the ACC, followed by the administration of BafA1 at varying concentrations via a microinfusion pump (Appendix Fig. S3B). Quantitative analysis revealed a significant reduction in the PWT from day 3 to day 10 postinfusion in 10 ng and 50 ng groups (Appendix Fig. S3C,D). PWL analysis showed significant decreases in the 10 ng and 50 ng groups from day 3 to day 7 (Appendix Fig. S3C,D). Notably, BafA1 administration had no significant effect on motor coordination or anxiety-like behaviors of mice (Appendix Fig. S3E,F). Quantitative analysis revealed that BafA1 administration (10 ng and 50 ng doses) significantly upregulated the autophagy levels and inhibited autophagy flux (Appendix Fig. S3G,H). Notably, SQSTM1 expression exhibited a temporal pattern of upregulation, with significant increases observed between day 3 and day 7, rather than on day 14, which paralleled the PWT changes in the 10 ng treatment group (Appendix Fig. S3H). Furthermore, regression analysis demonstrated a significant negative correlation between SQSTM1 levels and left PWT in the 10 ng group (Appendix Fig. S3I). Fluorescence analysis also revealed significantly more EGFP-mCherry-LC3B puncta in BafA1 group than in the control group (Appendix Fig. S3J,K). These findings provide that obstructive autophagy flux may be an important factor in inducing pain hypersensitivity.

Lysosomes serve as the primary degradation sites for autophagosomes. To elucidate the mechanism underlying impaired autophagy in the ACC of SMIR mice, we assessed lysosomal activity via the following key molecular markers: LAMP1, ATPase $H^+$ transporting V0 subunit D1 (ATP6V0D1, lysosome acidification), lysosomal associated membrane protein 2A (LAMP2A, lysosome membrane fusion), and cathepsin B (CTSB, lysosome hydrolase). Quantitative analysis revealed a significant increase in LAMP1 expression in the ACC of SMIR mice, while other markers remained unchanged (Fig. 4A). Subsequent western blot analysis of lysosomal proteins demonstrated a substantial reduction in CTSB levels in lysosomes of SMIR group (Fig. 4B). Immunofluorescence analysis further confirmed a significant decrease in CTSB and LAMP1 colocalization signals in the SMIR group, independent of the CTSB expression level (Fig. 4C). These results suggest the incorrect sorting of lysosomal hydrolases in the ACC of SMIR mice.

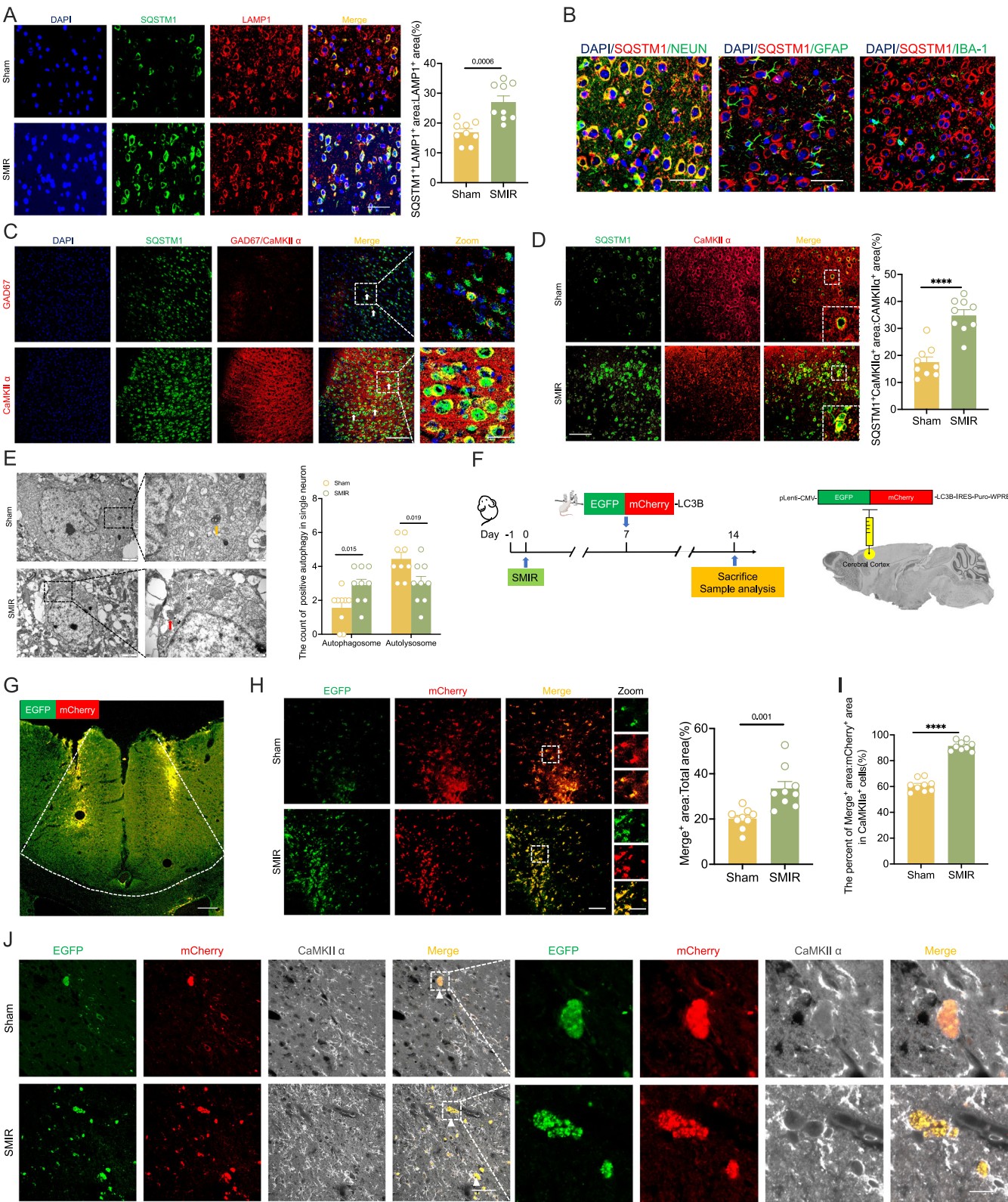

**Figure 3.  Impaired autophagy flux in the ACC of SMIR mice.**

(**A**) Representative immunofluorescence images (left) and statistical analysis (right) of SQSTM1 colocalization with LAMP1 in the ACC of SMIR mice. $n = 9$ per group; two-tailed unpaired Student's $t$ tests. Scale bars: 50 µm. (**B**) Distribution of SQSTM1 in neurons, astrocytes, and microglia in the ACC of SMIR mice. Scale bar: 20 µm.
(**C**) Distribution of SQSTM1 in GAD67-positive (white arrows) and CaMKIIα-positive neurons (white arrows) in the ACC of SMIR mice. Scale bar: 100 µm and 20 µm.
(**D**) Representative images (left) and increased immunofluorescence colocalization (right) of SQSTM1 with CaMKIIα in the ACC of SMIR mice. $n = 9$ per group; ****$P < 0.0001$ vs. the sham group; two-tailed unpaired Student's $t$ tests; Scale bar: 50 µm. (**E**) TEM analysis revealed a significant increase in autophagosomes (red arrows) and a decrease in autolysosomes (yellow arrows) in ACC neurons of SMIR mice. Representative images (left) and statistical analysis (right) are shown. $n = 9$ per group; multiple unpaired $t$ tests with Bonferroni post hoc correction. Scale bar: 0.5 µm. (**F**) Experimental design (left) and schematic diagram (right) for pLenti-EGFP-mCherry-LC3B assays. pLenti-EGFP-mCherry-LC3B was microinfused into the ACC on POD7. (**G**) Successful transfection of pLenti-EGFP-mCherry-LC3B in the ACC. The white dashed lines are the boundary of the ACC brain region. Scale bar: 200 µm. (**H**) Representative images (left) and statistical analysis (right) of merge-positive area in the ACC. $n = 9$ per group; two-tailed unpaired Student's $t$ tests. Scale bar: 50  µm and 20 µm. (**I, J**) Immunofluorescence analysis (**I**) and representative images (**J**) of pLenti-EGFP-mCherry-LC3B (white arrows) in the CaMKIIα-positive neurons. $n = 9$ per group; ****$P < 0.0001$ vs. the sham group; two-tailed unpaired Student's $t$ tests. Scale bar: 50 µm and 20 µm. All data are presented as mean ± SEM. Source data are available online for this figure.

On the basis of previous studies demonstrating the role of TMEM251 in mediating lysosomal hydrolases sorting (Pechincha et al, 2022; Qiao et al, 2023), we investigated its potential involvement in autophagy impairment under CPOP conditions. Quantitative PCR analysis revealed a significant reduction in *Tmem251* mRNA levels in the ACC of SMIR mice at POD7 and POD14, paralleling alterations in autophagy (Fig. 4D). Furthermore, in situ hybridization analysis demonstrated SMIR surgery decreased TMEM251 expression, specifically in CaMKIIα-positive neurons, but not in GAD67-positive neurons (Fig. 4E,F). The change of TMEM251 expression was consistent with the accumulation of autophagy substrates in the ACC of SMIR mice.

## The overexpression of TMEM251 within the ACC alleviates impaired autophagy and prevents chronicity of postoperative pain in SMIR mice

To investigate the role of TMEM251 in autophagy and CPOP, we microinjected AAV-DIO-TMEM251 virus into the ACC of CaMKII-Cre mice, which targeted the overexpression of TMEM251 in CaMKIIα-positive neurons (Fig. 5A; Appendix Fig. S4E). In situ hybridization and qPCR results showed that TMEM251 expression in the ACC was significantly increased in the AAV-TMEM251 group, especially in mCherry-positive cells (Fig. 5B,C). Notably, TMEM251 upregulation enhanced the colocalization of CTSB and LAMP1 without altering CTSB expression levels (Fig. 5D; Appendix Fig. S4D). Behavioral assessments demonstrated that scrambled virus-treated SMIR mice exhibited a sustained reduction in ipsilateral PWT from POD1 to POD14 (Fig. 5E,F). In contrast, AAV-TMEM251-transfected SMIR mice presented a transient decrease in PWT during POD1-POD3, followed by a significant recovery from POD5 to POD14, indicating that TMEM251 overexpression attenuated the maintenance of postoperative pain without preventing its initial occurrence (Fig. 5E,F). Furthermore, TMEM251 overexpression significantly reduced the lidocaine-paired chamber dwelling time in SMIR mice (Appendix Fig. S4A). Importantly, this intervention did not affect locomotor activity or anxiety-like behaviors in SMIR mice, as demonstrated by OFT and EPM (Appendix Fig. S4B,C).

Western blot analysis revealed that TMEM251 overexpression significantly attenuated the SMIR-induced elevation in the levels of ATG5, SQSTM1, and LC3B-II (Fig. 5G). Consistent with these findings, immunofluorescence analysis revealed reduced SQSTM1 expression in mCherry-positive neurons following TMEM251

overexpression (Fig. 5H). Ultrastructural analysis using TEM also demonstrated that TMEM251 overexpression markedly reduced the number of autophagosomes while increasing autolysosomes in ACC neurons (Appendix Fig. S4F). Western blot analysis also demonstrated that TMEM251 overexpression significantly reduced the expression of PSD95 and VGLUT1 in SMIR group (Appendix Fig. S4G). Furthermore, the overexpression of TMEM251 reduced the accumulation of synaptophysin and PSD95 in autophagy substrates of SMIR mice (Fig. 5I; Appendix Fig. S4H,I). These results collectively demonstrate that the overexpression of TMEM251 prevented the chronicity but not the occurrence of postoperative pain in SMIR mice. Mechanistically, the overexpression of TMEM251 increases lysosomal localization of CTSB, alleviates autophagy disorders, and reduces the accumulation of synaptic proteins in autophagy substrates.

## TMEM251 knockdown in the ACC disrupts autophagy and induces chronic pain phenotypes in naive mice

To elucidate the regulatory role of TMEM251 in autophagy, we employed RNAi technology to deplete TMEM251 expression in SH-SY5Y cells (Appendix Fig. S5A). Immunofluorescence revealed that TMEM251 depletion led to a more dispersed cytoplasmic distribution of CTSB and significantly reduced its colocalization with LAMP1 (Fig. 6A). Notably, TMEM251-knockdown cells exhibited marked upregulation of ATG7, ATG5, SQSTM1, and LC3B-II (Fig. 6B). Subsequent transfection with pLenti-EGFP-mCherry-LC3B demonstrated increased EGFP-mCherry-LC3B puncta formation with enhanced aggregation patterns in TMEM251- knockdown cells (Fig. 6C,D). To further investigate the functional consequences of TMEM251 in the ACC, we performed stereotaxic microinjection of RNAi-*Tmem251* constructs into the ACC of naive mice (Fig. 6E). Quantitative PCR analysis confirmed significant downregulation of *Tmem251* mRNA in the ACC of RNAi-treated animals (Fig. 6F). Autophagy assessment revealed the accumulation of autophagosomes in TMEM251-deficient mice, as indicated by a significant increase in the EGFP-mCherry-LC3B puncta (Fig. 6G). Behavioral assessment revealed that TMEM251 knockdown in the ACC significantly reduced the PWT and PWL of both hind paws, from post-RNAi day 3 through day 10 (Fig. 6H,I). However, anxiety-like behaviors and damaged locomotion were not observed in the OFT or EPM test (Appendix Fig. S5B,C). Western blot analysis demonstrated that TMEM251 knockdown significantly increased the expression

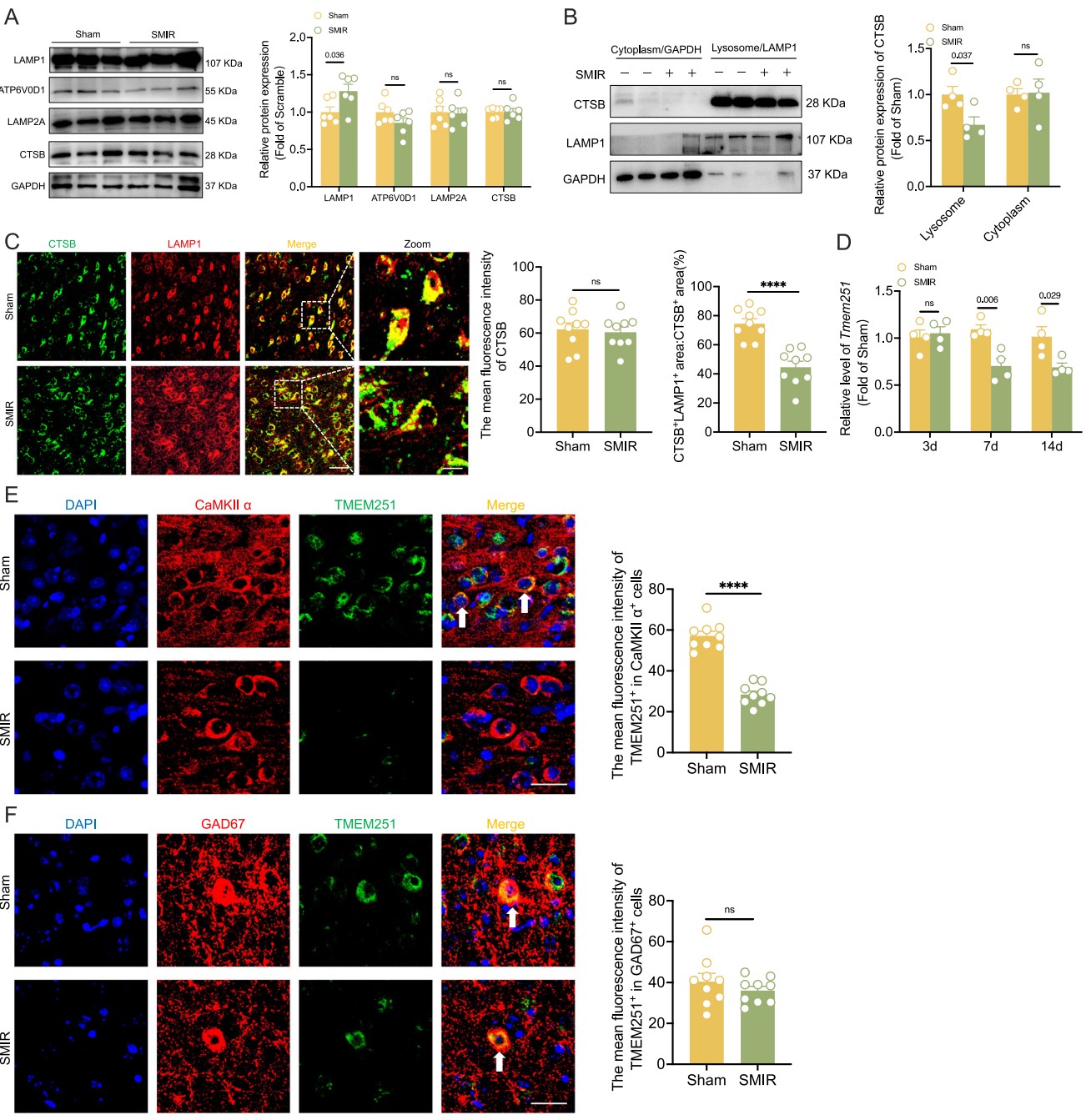

**Figure 4. Reduced lysosomal localization of CTSB and expression of TMEM251.**

(A) Representative images (left) and statistical analysis (right) of the expression of LAMP1, ATP6V0D1, LAMP2A, and CTSB in the ACC. $n = 6$ per group; multiple unpaired $t$ tests with Bonferroni post hoc correction. (B) Representative images (left) and statistical analysis (right) of the expression of CTSB in lysosomal proteins within ACC tissue. $n = 4$ per group; multiple unpaired $t$ tests with Bonferroni post hoc correction. (C) Immunofluorescence colocalization images (left) and statistical analysis (right) of CTSB with LAMP1 in the ACC of sham and SMIR mice. $n = 9$ per group; ****$P < 0.0001$ vs. the sham group; two-tailed unpaired Student's $t$ tests. Scale bar: 20 μm and 20 μm. (D) Temporal expression profile of *Tmem251* mRNA in the ACC of SMIR mice. Significant downregulation was observed on POD7 and POD14, but not on POD3. $n = 4$ per group; two-tailed unpaired Student's $t$ test. (E) Reduced TMEM251 expression in CaMKIIα-positive neurons (white arrows) of SMIR mice. Representative images (left) and statistical analysis (right) are shown. $n = 9$ per group; ****$P < 0.0001$ vs. the sham group; two-tailed unpaired Student's $t$ tests. Scale bar: 5 μm. (F) TMEM251 expression did not show significant changes in GAD67-positive cells (white arrows) of SMIR mice, representative images (left) and statistical analysis (right) are shown. $n = 9$ per group; two-tailed unpaired Student's $t$ tests. Scale bar: 5 μm. All data are presented as the mean ± SEM. Source data are available online for this figure.

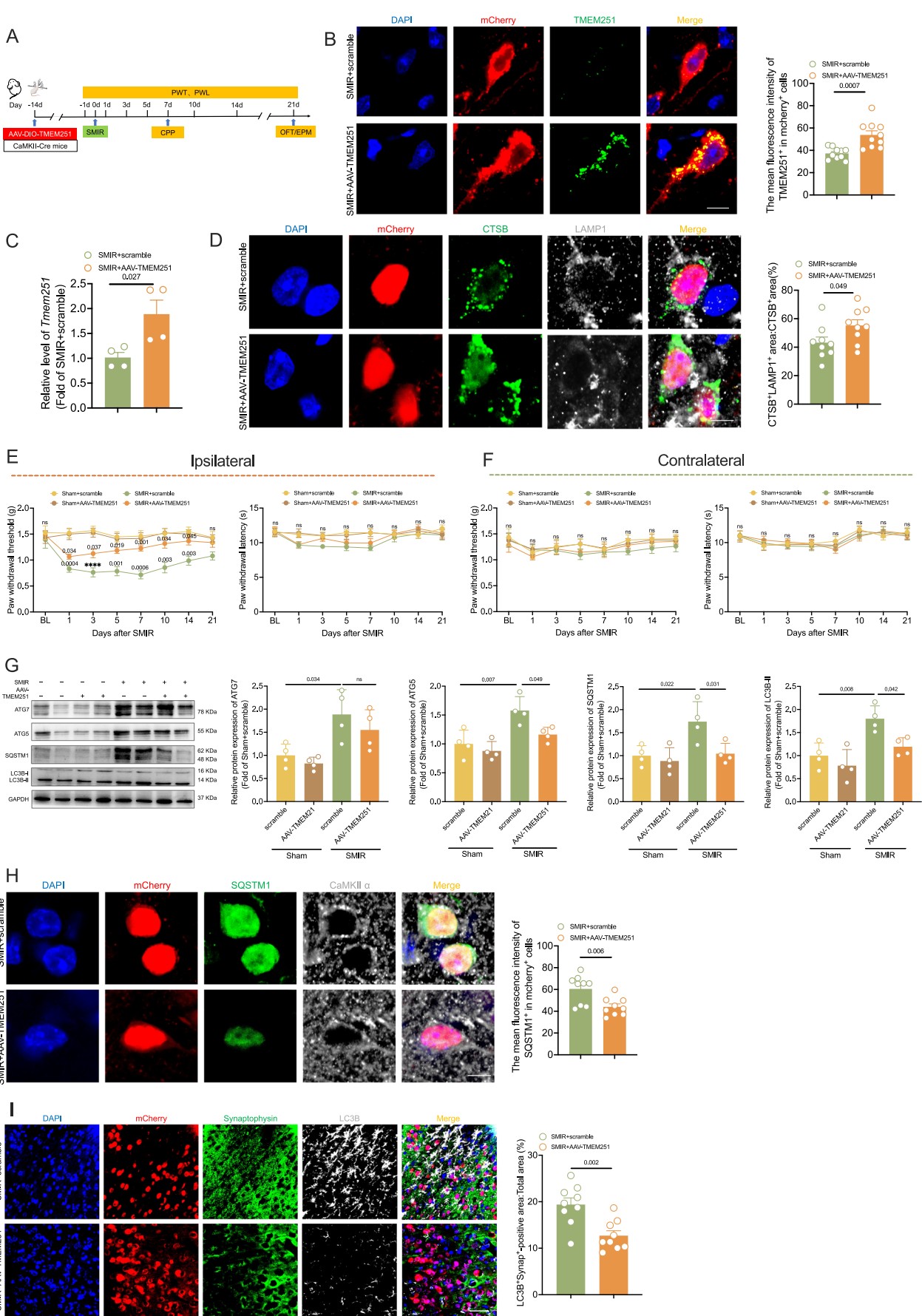

**Figure 5.  The overexpression of TMEM251 within the ACC inhibits the maintenance of postoperative pain in SMIR mice.**

(A) Experimental design for TMEM251 overexpression in the ACC of SMIR mice. AAV-TMEM251-DIO virus was microinjected into the ACC of CaMKII-Cre mice 14 days before SMIR surgery. PWT and PWL were measured at baseline (day -1) and at POD1, 3, 5, 7, 10, 14 and 21. The CPP was performed on POD7. The OFT and EPM were performed on POD21. (B) In situ hybridization images (left) and statistical analysis (right) of TMEM251 expression in mCherry-positive cells. $n = 10$ per group; two-tailed unpaired Student's $t$ tests. Scale bar: 5 μm. (C) The mRNA level of *Tmem251* was elevated in the ACC of the SMIR + AAV-TMEM251 group. $n = 4$ per group; two-tailed unpaired Student's $t$ tests. (D) Representative immunofluorescence images (left) and statistical analysis (right) of CTSB and LAMP1 colocalization in mCherry-positive cells. $n = 9$ per group; two-tailed unpaired Student's $t$ tests. Scale bar: 5 μm. (E) Behavioral analysis of ipsilateral PWT (left) and PWL (right) across four groups. $n = 8$ per group; ****$P < 0.0001$ vs. the sham+scramble group; $P = 0.0004$, 0.0001, 0.0006, 0.003 and 0.003 vs. the sham+scramble group; $P = 0.034$ and 0.037 vs. the sham+ AAV-TMEM251 group; $P = 0.019$, 0.001, 0.034 and 0.045 vs. the SMIR+scramble group; two-way repeated-measures ANOVA with Bonferroni post hoc correction. (F) Behavioral analysis of contralateral PWT (left) and PWL (right) across four groups. $n = 8$ per group; two-way repeated measure ANOVA with Bonferroni's post hoc test. (G) Western blot analysis of autophagy-related proteins (ATG7, ATG5, SQSTM1, and LC3B) in the ACC with TMEM251 overexpression. Representative images (left) and statistical analysis (right) are shown. $n = 4$ per group; one-way repeated-measures ANOVA with Bonferroni post hoc correction. (H) Immunofluorescence analysis of SQSTM1 in mCherry-positive neurons. Representative images (left) and statistical analysis (right) are shown. $n = 9$ per group; two-tailed unpaired Student's $t$ tests. Scale bar: 5 μm. (I) Representative immunofluorescence images (left) and statistical analysis (right) of LC3B colocalization with synaptophysin in the ACC with TMEM251 overexpression. $n = 9$ per group; two-tailed unpaired Student's $t$ tests. Scale bars: 50 μm. All data are presented as mean ± SEM. Source data are available online for this figure.

of PSD95 and VGLUT1 in naive mice (Appendix Fig. S5D). Furthermore, RNAi-TMEM251 also induced the accumulation of synaptophysin and PSD95 in autophagy substrates, which is consistent with the findings in SMIR mice (Fig. 6J; Appendix Fig. S5E,F). These findings collectively demonstrate that TMEM251-deficiency in the ACC disrupts autophagy, promotes the accumulation of synaptic proteins in autophagy substrates, and contributes to the development of chronic pain phenotypes in naive mice.

## Disruption of autophagy triggers the activation of CaMKIIα-positive neurons in the ACC

To investigate the impact of autophagy disorders on neuronal activity, C-FOS was used to label activated neurons. Immunofluorescence analysis demonstrated a significant increase in the number of C-FOS-positive cells in the ACC of the SMIR group (Appendix Fig. S6A). Notably, the number of activated CaMKIIα-positive cells significantly increased in the SMIR group, could be reduced by the overexpression of TMEM251 (Fig. 7A,B). However, there was no significant change in the number of activated GAD67-positive cells (Appendix Fig. S6B). Furthermore, quantitative analysis revealed that 41.30% ± 12.42% of C-FOS-positive cells were also positive for SQSTM1, and 39.39% ± 12.39% of SQSTM1-positive cells expressed C-FOS, indicating a close relationship between impaired autophagy and neuronal activation (Appendix Fig. S6C). Next, we employed fiber photometry to monitor neuronal activity in CaMKII-Cre mice (Fig. 7C,D). Mechanical stimulation of hind paws via von Frey filaments during photometric recordings revealed that SMIR surgery significantly enhanced intracellular calcium ($Ca^{2+}$) dynamics ($\Delta F/F_0$) in CaMKIIα-positive neurons, responding to both non-nociceptive (0.6 g) and nociceptive (2 g) stimuli (Fig. 7E–J). Notably, TMEM251 overexpression attenuated this neuronal hyperactivation in SMIR mice (Fig. 7E–J).

Complementary experiments demonstrated that TMEM251 knockdown recapitulated the SMIR-induced CaMKIIα-positive neuronal activation phenotype, as evidenced by both immunofluorescence (Appendix Fig. S6D,E) and photometry analyses (Fig. 7K–R). Quantitative analysis also showed that the count of C-FOS-positive cells was significant increased, mainly colocalized with CaMKIIα in the ACC of BafA1-treated mice (Appendix

Fig. S6F–H). These collective findings demonstrate that impaired autophagy primarily activate CaMKIIα-positive neurons in the ACC, whereas TMEM251 overexpression effectively mitigated this activation, establishing a mechanistic link between autophagy regulation and neuronal activity in the ACC.

## Sustained activation of CaMKIIα-positive neurons in the ACC induces impaired autophagy

To elucidate the role of CaMKIIα-positive neurons in SMIR-induced CPOP, we employed chemogenetic approaches to modulate CaMKIIα-positive neuronal activity in SMIR mice (Fig. 8A,B). Notably, chemogenetic inhibition of CaMKIIα-positive neurons effectively increased the ipsilateral PWT and reduced affective-motivational pain behaviors in SMIR mice (Fig. 8C,D; Appendix Fig. S7A), without affecting motor coordination or anxiety-like behaviors (Appendix Fig. S7B,C). Similarly, chemogenetic inhibition of CaMKIIα-positive neurons significantly attenuated nociceptive hypersensitivity in siTMEM251 and BafA1 mice (Appendix Fig. S7D–M). Amino-5-phosphonovaleric acid (AP-5) and cyano-2,6-cyano-7-nitroquinoxaline-2,3-dione (CNQX) are important excitatory synaptic transmission blockers. The behavioral data indicated that microinjection of AP-5 or CNQX into the ACC also significantly alleviated mechanical allodynia of SMIR mice at POD7 (Appendix Fig. S7N,O).

Immunofluorescence analysis revealed that inhibition of activated CaMKIIα-positive neurons significantly attenuated SQSTM1 and LC3B accumulation in the ACC of SMIR mice (Fig. 8E). To further investigate the association between CaMKIIα-positive neuron activation and autophagy dysregulation, we employed chemogenetic approaches to activate CaMKIIα-positive neurons in naive mice (Fig. 8F,G). CNO administration markedly increased the number of C-FOS-positive cells in the ACC (Fig. 8H). Notably, the CNO-treated group demonstrated significant SQSTM1 accumulation in C-FOS-positive neurons (Fig. 8H). Similarly, the expression of both ATG5 and LC3B-II was significantly elevated in the ACC of CNO-3d group (Fig. 8I). Intriguingly, prolonged activation (CNO-7d group) resulted in additional upregulation of ATG7 and SQSTM1 expression (Fig. 8I). TEM analysis showed a significant increase in the number of autophagosomes in the ACC neurons of CNO group (Appendix Fig. S8A). Immunofluorescence analysis further demonstrated that sustained CaMKIIα-positive

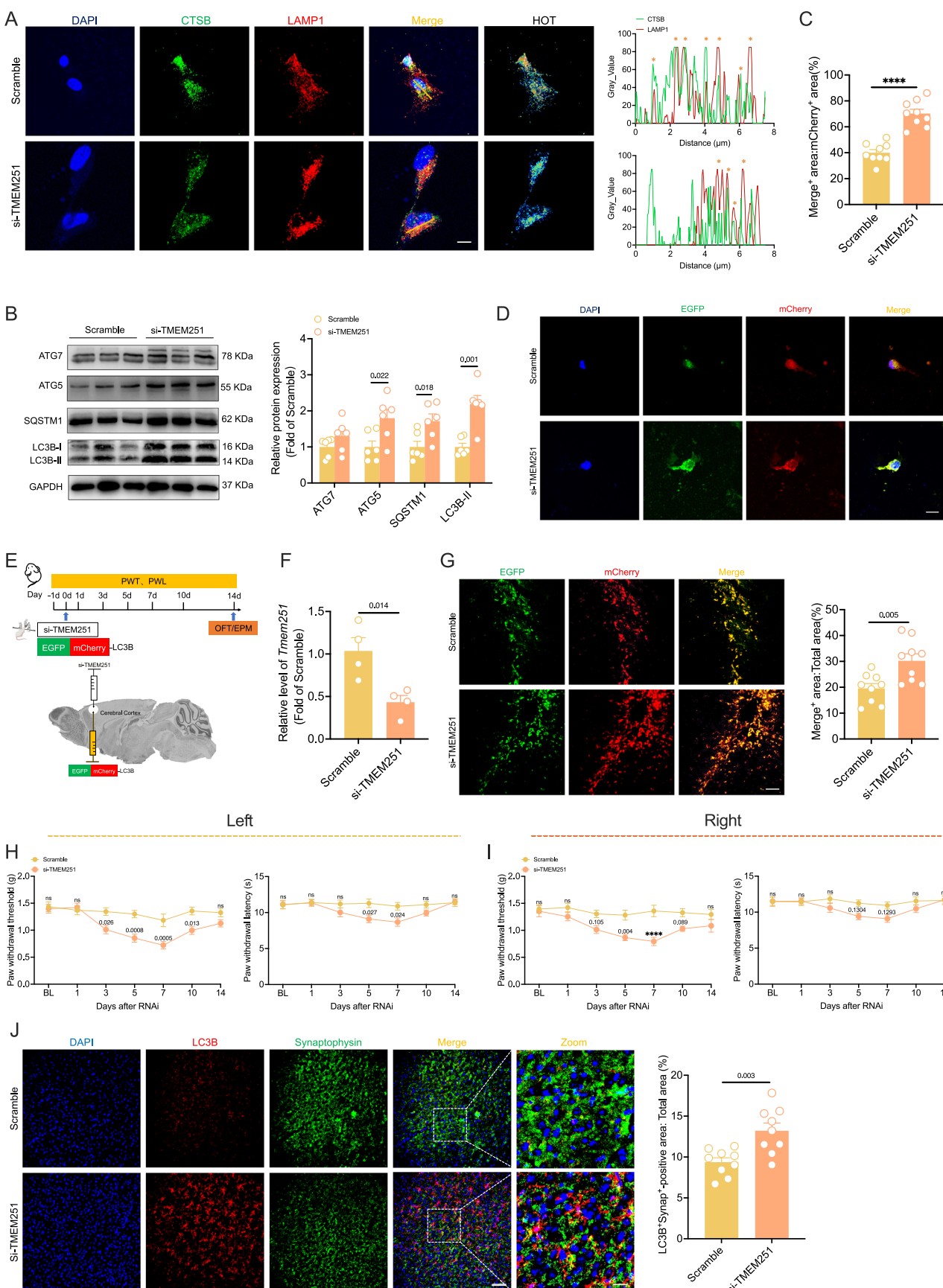

◄ **Figure 6. Knockdown of TMEM251 in the ACC disrupts autophagy and induces chronic pain phenotypes in naive mice.**

(A) Representative immunofluorescence images (left) and quantitative analysis (right) showing reduced colocalization of CTSB with LAMP1 in TMEM251-depleted cells. Scale bar: 5 μm. (B) Western blot analysis demonstrating upregulated expression of autophagy-related proteins (ATG5, SQSTM1, and LC3B) following TMEM251 knockdown. $n = 6$ per group; multiple unpaired $t$ tests with Bonferroni post hoc correction. (C, D) Quantitative analysis of EGFP-mCherry-LC3B colocalization signals (C) and representative fluorescence images (D) of transfected cells. $n = 9$ per group; ****$P < 0.0001$ vs. the scramble group; two-tailed unpaired Student's $t$ tests; scale bar: 5 μm. (E) Experimental timeline (top) and stereotaxic injection schematic (bottom) for ACC-specific TMEM251 knockdown. Behavioral assessments included PWT and PWL measurements at indicated time points, with OFT and EPM performed on day 14. (F) Quantitative PCR analysis showing a significant reduction in *Tmem251* mRNA levels in the ACC of si-TMEM251 mice. $n = 4$ per group; two-tailed unpaired Student's $t$ tests. (G) Representative images (left) and quantitative analysis (right) of pLenti-EGFP-mCherry-LC3B showing increased EGFP-mCherry-LC3B puncta in TMEM251-konckdown mice. $n = 9$ per group; two-tailed unpaired Student's $t$ tests. Scale bar: 50 μm. (H) PWT (left) and PWL (right) of the left paws of the mice in the scramble and siTMEM251 groups. $n = 12$ per group; two-way repeated-measures ANOVA with Bonferroni post hoc correction. (I) PWT (left) and PWL (right) of the right paws of the mice in the scramble and siTMEM251 groups. $n = 12$ per group; ****$P < 0.0001$ vs. the scramble group; two-way repeated-measures ANOVA with Bonferroni post hoc correction. (J) Representative immunofluorescence images (left) and statistical analysis (right) of LC3B colocalization with synaptophysin in the ACC with TMEM251 knockdown. $n = 9$ per group; two-tailed unpaired Student's $t$ tests. Scale bars: 100 μm and 20 μm. All data are presented as the mean ± SEM. Source data are available online for this figure.

neuron activation enhanced the accumulation of synaptophysin and PSD95 in autophagy substrates, while gephyrin levels remained unaffected (Fig. 8J; Appendix Fig. S8B,C). These findings collectively suggest that chronic activation of CaMKIIα-positive neurons elevates autophagy activity, impairs autophagy flux, and promotes synaptic protein accumulation within autophagy substrates, thereby establishing a link between neuronal hyperactivation and autophagy dysfunction.

## Discussion

Maladaptive neural plasticity in the central nervous system is a key underlying mechanism of CPOP (Widder et al, 2024). As an essential cellular process for maintaining neuronal homeostasis, autophagy plays a pivotal role in preserving neuronal structure and function (Kallergi et al, 2022). Our study revealed that impaired autophagy, particularly in CaMKIIα-positive neurons of the ACC, serves as a critical driver of CPOP progression. We observed significant accumulation of synaptic proteins in autophagy substrates. Notably, TMEM251 overexpression in the ACC effectively rescued autophagy impairment, reduced synaptic protein accumulation in autophagy substrates, and alleviated the maintenance of CPOP. Conversely, knocking down TMEM251 in the ACC induced chronic pain phenotype and impaired autophagy in naive mice. Furthermore, we demonstrated that autophagy dysregulation specifically activates CaMKIIα-positive neurons in the ACC, whereas sustained activation of these neurons conversely induces autophagy impairment. These findings establish a crucial role for autophagy in CPOP pathogenesis and identify TMEM251 as a promising therapeutic target for postoperative pain management.

The post-thoracotomy model and SMIR model are the most commonly employed models to investigate the pathogenesis of CPOP. However, due to the substantial trauma and high postoperative mortality associated with the post-thoracotomy model in mice (Zhu et al, 2023), we selected the SMIR model for this study. Previous reports indicate that SMIR-induced CPOP can persist for 22 days (Flatters, 2008). In contrast, our results demonstrated significant pain hypersensitivity in SMIR mice from POD1 to POD14, with recovery at POD21. This discrepancy may stem from the variations in surgical retraction duration and distance. Consistent with previous findings (Li et al, 2020), we observed

chronic mechanical allodynia but no thermal hypersensitivity in SMIR mice. This may be due to the mechanical force induced by SMIR surgery preferentially activating mechanical pressure-sensitive channels, not the temperature-sensitive channels, in peripheral primary sensory neurons (Arora et al, 2021; Coste et al, 2010). Interestingly, the mechanical allodynia induced by SMIR surgery lasted longer in female mice. Similarly, abundant evidence from epidemiologic studies has demonstrated that women are at substantially greater risk for many clinical pain conditions (Pieretti et al, 2016). Potential biological mechanisms include hormonal influences and sex-dependent immune cell activation (Sorge et al, 2015). Although SMIR mice exhibited a tendency toward anxiety-like behaviors, this effect did not reach statistical significance, possibly because of the limited intensity and duration of CPOP induced by the SMIR surgery.

The cortical regions play important roles in nociceptive signal processing and sorting. The insular cortex, for example, integrates pain-related emotional and proprioceptive information (Li et al, 2024; Zhang et al, 2022a). By contrast, the ACC is involved in processing the affective dimensions of pain and associated emotional responses (Zhang et al, 2024b; Zhu et al, 2022). Autophagy plays an important role in neuronal homeostasis by clearing damaged organelles, maintaining energy balance, responding to oxidative stress, regulating synaptic plasticity, and providing neuroprotection (Cao et al, 2019; Liu et al, 2021). Emerging evidence has highlighted that autophagy is significantly upregulated in the spinal dorsal horn of mice with neuropathic pain and inflammatory pain, particularly within neurons (Xu et al, 2024). The autophagy changes in the brain under chronic pain still need to be explored. Our findings revealed impaired autophagy flux in the ACC of male and female mice in SMIR group, predominantly in CaMKIIα-positive neurons. Importantly, this phenomenon was observed during the maintenance of CPOP, rather than at initiation. Consistently, the administration of BafA1, a blocker of autophagy flux, in the ACC induced chronic pain phenotype in naive mice. These observations align with previous reports demonstrating that impaired neuronal autophagy in the prelimbic cortex contributes to anxiety-like behaviors induced by neuropathic pain (Fu et al, 2024). These results indicate that impaired autophagy flux in central neurons is an important factor for chronic pain.

In the present study, we further observed that the autophagy alterations in the ACC were primarily localized to layers III, V, and

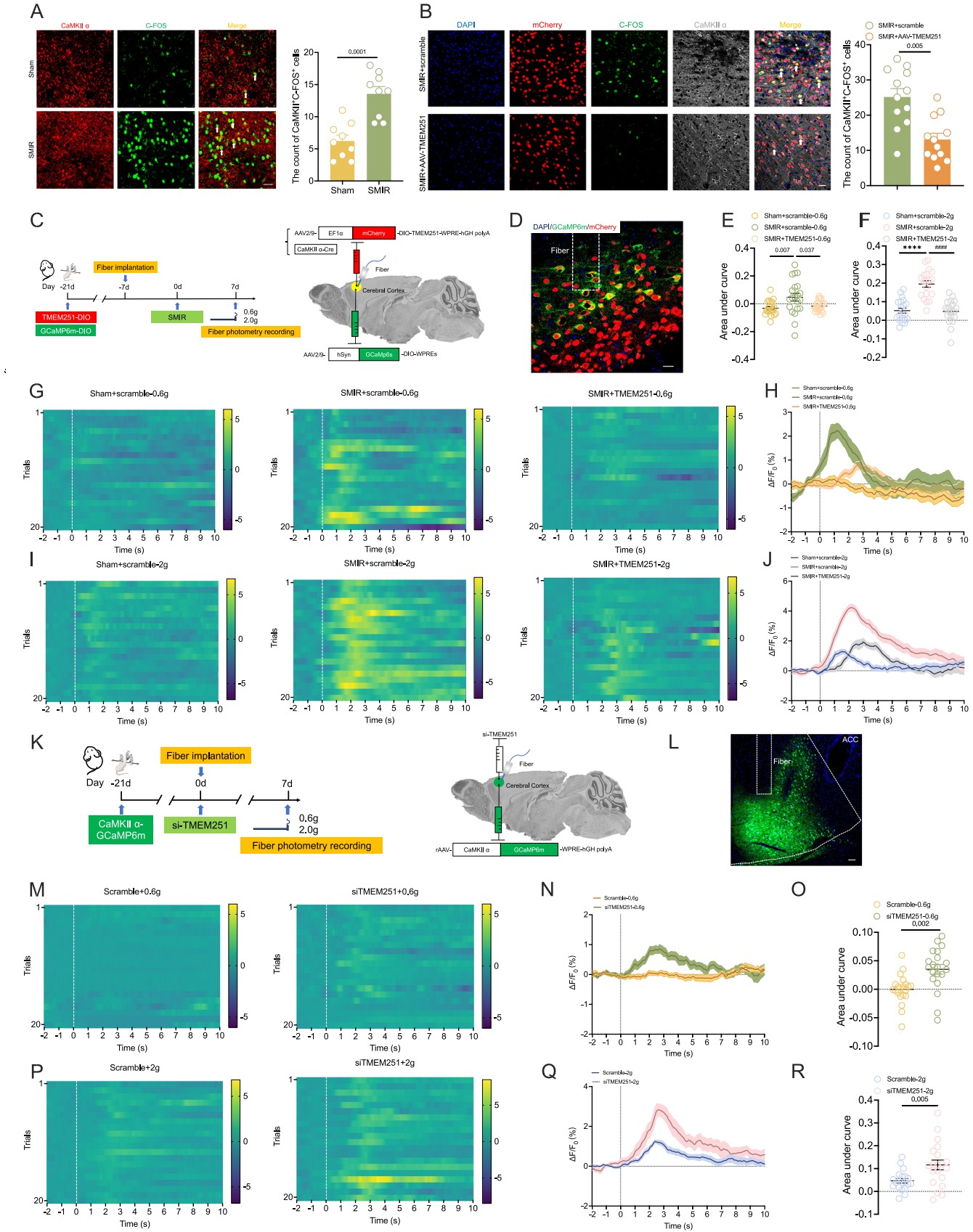

◀ **Figure 7. Damaged autophagy activates CaMKIIα-positive neurons in the ACC.**

(A) Representative images (left) and statistical analysis (right) of immunofluorescence colocalization of CaMKIIα and C-FOS (white arrows) in the sham and SMIR groups. $n = 9$ per group; two-tailed unpaired Student's $t$ tests. Scale bar: 50 μm. (B) The count of CaMKIIα⁺C-FOS⁺ cells (white arrows) was reduced in the ACC of SMIR + AAV-TMEM251 group. Representative fluorescence images (left) and statistical analysis (right) are shown. $n = 12$ per group; two-tailed unpaired Student's $t$ tests. Scale bar: 50 μm. (C) Experimental design for TMEM251 overexpression and fiber photometry recordings in CaMKIIα-positive neurons. Left: Experimental timeline. Right: Schematic of viral infection and fiber implantation in the ACC of CaMKII-Cre mice. AAV2/9-TMEM251-DIO and AAV2/9-GCaMP6m-DIO viruses were stereotaxically injected into the ACC, followed by optical fiber implantation. After 7 days of recovery, mice underwent SMIR surgery, with photometric recordings performed at POD7. (D) Representative images of viral expression and fiber implantation in the ACC. Scale bar: 20 μm. (E, F) Quantitative analysis of Ca²⁺ activity (area under curve, AUC) to mechanical stimulation: 0.6 g von Frey filament (E) and 2 g von Frey filament (F). $n = 21$ per group; ****$P < 0.0001$ vs. the sham+scramble group; ####$P < 0.0001$ vs. the SMIR+scramble group; one-way repeated-measures ANOVA with Bonferroni post hoc correction. (G) Heatmaps of Ca²⁺ activity (ΔF/F₀) in CaMKIIα-positive neurons stimulated with a 0.6 g von Frey filament. (H) Time-dependent curves of Ca²⁺ activity in CaMKIIα-positive neurons under 0.6 g von Frey filament stimulation. (I) Heatmaps of Ca²⁺ activity in CaMKIIα-positive neurons stimulated with 2 g von Frey filaments. (J) Time-dependent curves of Ca²⁺ activity in CaMKIIα-positive neurons in the ACC under 2 g von Frey filament stimulation. (K) Flowchart (left) and schematic diagram (right) of the fiber photometry recording experiment for CaMKIIα-positive neurons in mice. AAV-CaMKII-GCaMP6m was microinjected into the ACC. After stable virus transfection, RNAi-*Tmem251* was injected, and the fibers were implanted into the ACC. Fiber photometry recording experiments were performed on day 7 after microinjection. (L) Representative images of viral expression and fiber implantation in the ACC. The white dashed lines are the boundary of the ACC brain region. Scale bar: 200 μm. (M) Heatmaps of Ca²⁺ activity in CaMKIIα-positive neurons stimulated with 0.6 g von Frey filament. (N) Time–dependent curves of Ca²⁺ activity in CaMKIIα-positive neurons under 0.6 g von Frey filament stimulation. (O) The AUC of the si-TMEM251 group was significantly elevated with 0.6 g von Frey filament stimulation. $n = 20$ per group; two-tailed unpaired Student's $t$ tests. (P) Heatmaps of Ca²⁺ activity in CaMKIIα-positive neurons stimulated with 2 g von Frey filaments. (Q) Time-dependent curves of Ca²⁺ activity in CaMKIIα-positive neurons under 2 g von Frey filament stimulation. (R) The AUC of the si-TMEM251 group was significantly elevated with 2 g von Frey filament stimulation. $n = 22$ per group; two-tailed unpaired Student's $t$ tests. All data are presented as the mean ± SEM. Source data are available online for this figure.

VI. Previous study has indicated that deep-layer ACC neurons form extensive projection networks with subcortical regions including the spinal cord, critically contributing to chronic pain (Chen et al, 2018). These data suggest that autophagy impairment in the ACC may activate ACC-related top-down neural networks, involving in CPOP maintenance. Changes of synaptic plasticity in the ACC are the key factor in pain signal transmission (Bliss et al, 2016). Therefore, we evaluated the impact of neuronal autophagy damage on synapses in the ACC. The results indicated that SMIR surgery upregulated the expression of excitatory synaptic proteins (PSD95 and VGLUT1), but did not affect inhibitory synaptic proteins (Gephyrin and VGAT). Importantly, we identified significant accumulation of synaptophysin and PSD95 in autophagy substrates of SMIR mice. These findings align with prior reports showing that peripheral nerve injury enhances α-Amino-3-hydroxy-5-methyl-4-isoxazolepropionic acid (AMPA) receptor accumulation and membrane insertion in ACC layer V neurons, thereby facilitating nociceptive transmission (Chen et al, 2014a; Chen et al, 2014b). We also found that infusion of the excitatory synaptic transmission blockers (AP-5 and CNQX) in the ACC significantly reduced mechanical allodynia in SMIR mice. These results implicate excitatory synapse accumulation resulting from neuronal autophagy impairment as a critical mechanism driving postoperative pain chronification. Furthermore, previous studies have identified two forms of long-term potentiation (LTP) in the ACC: presynaptic LTP (pre-LTP) and postsynaptic LTP (post-LTP) (Koga et al, 2015). Pre-LTP has been shown to play a critical role in chronic pain and anxiety-like behaviors (Li et al, 2021b). It is worth noting that electrophysiological data is important for directly investigating the effects of autophagy damage on synaptic transmission and pre-LTP in ACC neurons. Therefore, the research on the electrophysiological characteristics of neuronal autophagy impairment is worth exploring in depth.

The impaired autophagy damage is mainly concentrated in CaMKIIα-positive neurons in the ACC of SMIR mice, accompanied by significant activation of these neurons. Furthermore, the chemogenetic inhibition of CaMKIIα-positive neurons effectively alleviated mechanical allodynia, and the accumulation of SQTM1

and LC3B in SMIR mice. Conversely, we observed that chronic activation of CaMKIIα-positive neurons in the ACC induced impaired autophagy flux and accumulation of synaptic proteins in autophagy substrates. Previous studies have suggested that CaMKII promotes autophagy through multiple signaling pathways, as an important kinase (Wu et al, 2024; Zhang et al, 2024a). Therefore, we speculate that the activation of CaMKII induces an increase in autophagy levels at the initiation of postoperative pain. As the chronic progression of postoperative pain, damaged autophagy induces the accumulation of excitatory synaptic proteins, exacerbating the excitation of CaMKII-positive neurons.

Lysosomes serve as critical organelles for the degradation of autophagy substrates, and their dysfunction can lead to the impaired autophagy (Ahn et al, 2023). In this study, our findings revealed an increase in lysosomal number in the ACC of SMIR mice. Further analysis of lysosomal proteins indicated a reduction in CTSB within lysosomes, indicating incorrect sorting of hydrolases. While recent studies have identified TMEM251 as a regulator of hydrolases sorting and autophagy (Qiao et al, 2023), its potential involvement in CPOP remains unexplored. Our data demonstrated markedly reduced TMEM251 expression in the ACC of SMIR mice, particularly in CaMKIIα-positive neurons. TMEM251 overexpression in the CaMKIIα-positive neurons increased the lysosomal localization of CTSB, alleviated autophagy barriers and neuronal activation in SMIR mice. Notably, overexpression of TMEM251 prevented the progression of postoperative pain without affecting its occurrence. Conversely, RNAi-mediated knockdown of TMEM251 resulted in reduction of CTSB in lysosomes, impaired autophagy, neuronal activation and chronic pain phenotype in naive mice. These results indicate that the deficiency of TMEM251 is one of the potential mechanisms underlying neuronal autophagy impairment in the ACC of CPOP mice. While this study highlights the role of TMEM251 in autophagy, the possibility that TMEM251 may influence pain through alternative pathways remains to be explored.

In summary, our study demonstrates that defective autophagy contributes to the chronicity of postoperative pain through the activation of CaMKIIα-positive neurons and the accumulation of

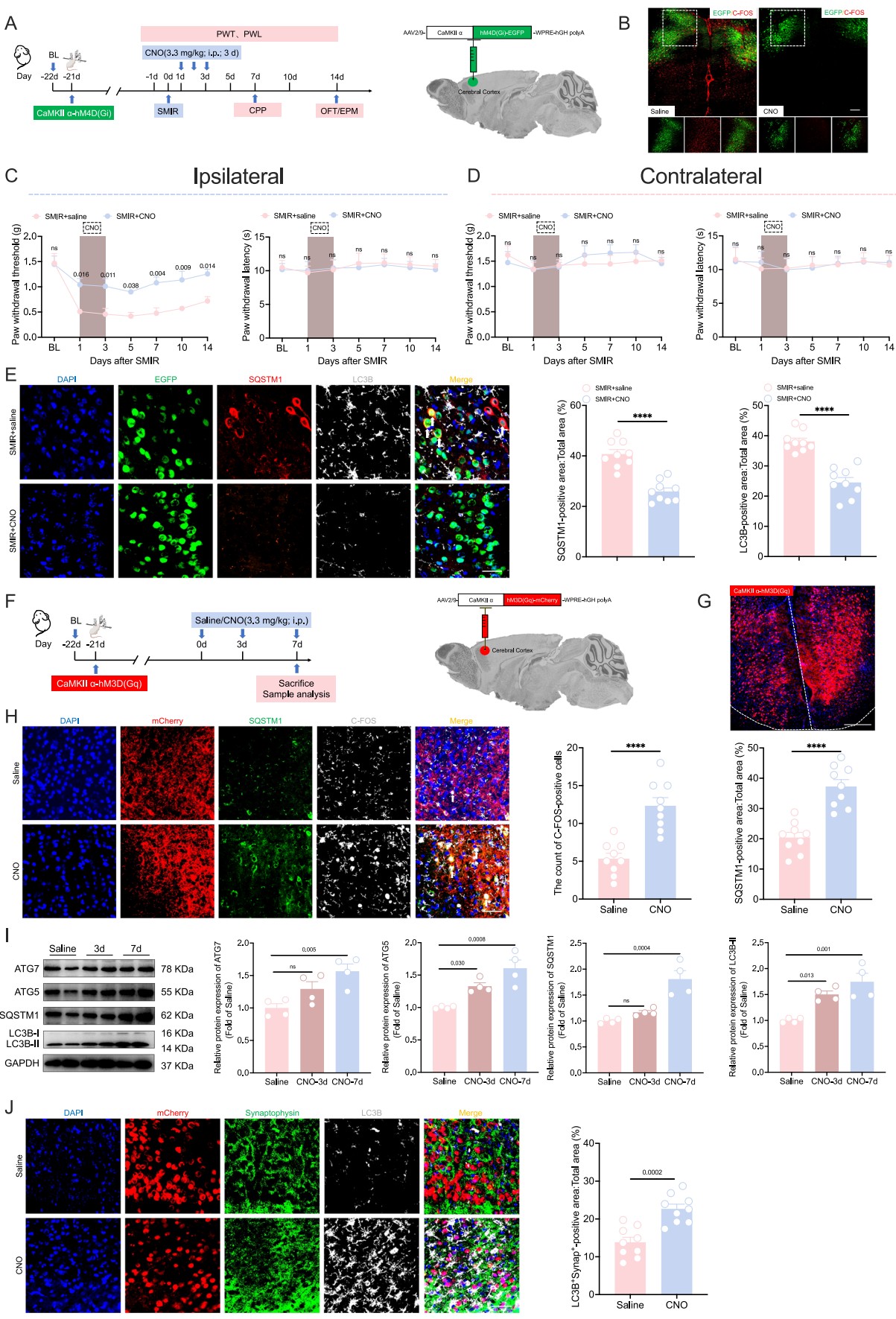

◄ **Figure 8. Continuous chemogenetic activation of CaMKIIα-positive neurons in the ACC induces autophagy impairment.**

(**A**) Flowchart (left) and schematic diagram (right) of the chemogenetic experiment for CaMKIIα-positive neurons in the ACC. AAV-CaMKIIα-hM4D(Gi) virus was injected into the bilateral ACC, and the SMIR model was established three weeks later. Saline or CNO (3.3 mg/kg) was injected intraperitoneally on POD1-POD3. PWT and PWL were measured at baseline (day -1) and POD1, 3, 5, 7, 10 and 14 d. The CPP was implemented at POD7. The OFT and EPM were performed at POD14. (**B**) Representative images showing CNO-mediated suppression of C-FOS expression in EGFP$^+$ regions. Scale bar: 200 μm. (**C**) Inhibition of activated CaMKIIα-positive neurons increased PWT (left), not PWL (right) in the ipsilateral paws of SMIR mice. $n = 6$ per group; two-way repeated-measures ANOVA with Bonferroni post hoc correction. (**D**) Inhibition of activated CaMKIIα-positive neurons did not affect PWT (left) or PWL (right) in the contralateral paws of the SMIR mice. $n = 6$ per group; two-way repeated-measures ANOVA with Bonferroni post hoc correction. (**E**) Inhibition of activated CaMKIIα-positive neurons reduced the accumulation of SQSTM1 and LC3B (white arrows) in the ACC of SMIR mice. $n = 9$ per group; ****$P < 0.0001$ vs. the SMIR+saline group; two-tailed unpaired Student's $t$ tests. Scale bar: 20 μm. (**F**) Flowchart (left) and schematic diagram (right) of the chemogenetic activation for CaMKIIα-positive neurons in the ACC. AAV-CaMKIIα-hM3D(Gq) virus was injected into the bilateral ACC. Saline or CNO (3.3 mg/kg) was injected intraperitoneally on POD1-POD7. (**G**) Representative virus-transfected images. The white dashed lines are the boundary of the ACC brain region. Scale bar: 200 μm. (**H**) The application of CNO significantly increased the number of C-FOS-positive cells (white arrows) and the accumulation of SQSTM1 in the ACC. $n = 9$ per group; ****$P < 0.0001$ vs. the saline group; two-tailed unpaired Student's $t$ tests. Scale bar: 20 μm. (**I**) Expression levels of autophagy-related proteins (ATG7, ATG5, SQSTM1, and LC3B) of saline, CNO-3d and CNO-7d groups. $n = 4$ per group; one-way repeated-measures ANOVA with Bonferroni post hoc correction. (**J**) The application of CNO significantly increased the colocalization signal of LC3B and synaptophysin in the ACC. $n = 9$ per group; two-tailed unpaired Student's $t$ tests. Scale bar: 50 μm. All data are presented as the mean ± SEM. Source data are available online for this figure.

excitatory synaptic proteins in autophagy substrates. Importantly, TMEM251 has emerged as a critical regulator of autophagy. These findings highlight a potential therapeutic strategy for managing postoperative pain by targeting autophagy pathways.

# Methods

### Reagents and tools table

| Reagent/resource | Reference or source | Identifier or catalog number |
|---|---|---|
| **Experimental models** | | |
| C57BL/6 (*M. musculus*) | Sbeff Biotechnology Co., Ltd | N/A |
| CaMKIIα-ires-Cre mice (*M. musculus*) | Shanghai Model Organisms Centre, Inc. | Cat# NM-KI-242500 |
| SH-SY5Y cell (human) | Procell Life | Cat# CL-0208 |
| PC12 cell (*R. norvegicus*) | Procell Life | Cat# CL-0412 |
| **Recombinant DNA** | | |
| N/A | | |
| N/A | | |
| **Antibodies** | | |
| Rabbit anti- ATG7 | Cell Signaling Technology | Cat# 2631 |
| Rabbit anti- ATG5 | Proteintech Biotechnology | Cat# 10181-2-AP |
| Mouse anti-SQSTM1/p62 | Invitrogen | Cat# MA5-27800 |
| Rabbit anti- SQSTM1/p62 | Proteintech Biotechnology | Cat# 18420-1-AP |
| Rabbit anti-LC3B | Proteintech Biotechnology | Cat# 18725-1-AP |
| Mouse anti-LC3B | Cell Signaling Technology | Cat# 83506S |
| Mouse anti-LAMP1 | Santa Cruz Biotechnology | Cat# sc-20011 |
| Rabbit anti-CTSB | Cell Signaling Technology | Cat# 31718S |

| Reagent/resource | Reference or source | Identifier or catalog number |
|---|---|---|
| Mouse anti-GAPDH | Proteintech Biotechnology | Cat# 60004-1-Ig |
| Rabbit anti-C-FOS | Synaptic Systems | Cat# 226008 |
| Mouse anti-C-FOS | Abcam | Cat# ab11959 |
| Rabbit anti-GAD67 | Abcam | Cat# ab213508 |
| Rabbit anti-CaMKIIα | Abcam | Cat# ab134041 |
| Rabbit anti-ATP6V0D1 | Proteintech Biotechnology | Cat# 18274-1-AP |
| Rabbit anti-LAMP2A | Abcam | Cat# ab18528 |
| Rabbit anti-GFAP | Cell Signaling Technology | Cat# 80788S |
| Rabbit anti-IBA1 | Abcam | Cat# ab178846 |
| Rabbit anti-NEUN | Abcam | Cat# ab177487 |
| Rabbit anti- Synaptophysin | Abcam | Cat# ab32127 |
| Rabbit anti-PSD95 | Proteintech Biotechnology | Cat# 20665-1-AP |
| Rabbit anti-VGLUT1 | Proteintech Biotechnology | Cat# 55491-1-AP |
| Rabbit anti-Gephyrin | Proteintech Biotechnology | Cat# 12681-1-AP |
| Rabbit anti-VGAT | Proteintech Biotechnology | Cat# 14471-1-AP |
| FITC-Goat Anti-Rabbit | Jackson ImmunoResearch | Cat# 111-095-003 |
| Cy3-Goat Anti-Rabbit | Jackson ImmunoResearch | Cat# 111-165-003 |
| FITC-Goat Anti-Mouse | Jackson ImmunoResearch | Cat# 115-095-003 |
| Cy3-Goat Anti-Mouse | Jackson ImmunoResearch | Cat# 115-165-003 |
| 647-Goat Anti-Rabbit | Abcam | Cat# ab150079 |
| 647-Goat Anti- Mouse | Abcam | Cat# ab150115 |
| **Oligonucleotides and other sequence-based reagents** | | |
| *Tmem251* siRNA in mice | Shanghai GenePharma | N/A |

| Reagent/resource | Reference or source | Identifier or catalog number |
|---|---|---|
| *Tmem251* siRNA in cell | Shanghai GenePharma | N/A |
| *Tmem251* nucleic acid fragments for in situ hybridization | BOSTER Biological | N/A |
| **Chemicals, enzymes and other reagents** | | |
| Clozapine N-oxide | MCE | Cat# HY-17366 |
| Isoflurane | RWD Life Science | Cat# R510-22-10 |
| Bafilomycin A1 | MCE | Cat# HY-100558 |
| CNQX | MCE | Cat# HY-15066 |
| AP-5 | MCE | Cat# HY-100714 |
| SYBR Green | Bioeast Biotechnology | Cat# M4QS01 |
| Trypsin | Gibco | Cat# 2520011 |
| **Software** | | |
| Graphpad Prism v9.0.0 | Graphpad software | https://www.graphpad.com/ |
| ImageJ | NIH | https://imagej.net/software/fiji/ |
| **Other** | | |
| rAAV2/9-CaMKIIα-GCaMp6m-WPRE-hGH polyA | BrainVTA Biotechnology | Cat# PT-0111 |
| rAAV2/9-EF1α-DIO-GCaMp6m-WPRE-hGH polyA | BrainVTA Biotechnology | Cat# PT-0283 |
| rAAV2/9-CaMKIIα-hM4D(Gi)-EGFP-hGH polyA | BrainVTA Biotechnology | Cat# PT-0524 |
| rAAV2/9-CaMKIIα-hM3D(Gq)-mCherry-hGH polyA | BrainVTA Biotechnology | Cat# PT-0049 |
| rAAV2/9-EF1α-DIO-Lyset-mCherry-WPRE-hGH polyA | BrainVTA Biotechnology | N/A |
| plenti-CMV-mCherry-EGFP-LC3B-IRE-PURO-WPRE | OBiO Biotechnology | N/A |
| Lysosomal protein extraction kit | Solarbio Science and Technology | Cat# EX1231 |
| BEAsy™ Chloroform-free Universal Total RNA Extraction Kit | Bioeast Biotechnology | Cat# M1TR03 |
| GeniScript™ RT SuperMix with gDNA Eraser 2.0 Kit | Bioeast Biotechnology | Cat# M5RT04 |
| Cell culture media for SH-SY5Y | Procell Life | Cat# CM-0208 |
| Cell culture media for PC-12 | Procell Life | Cat# CM-0412 |

## Animals

This study was conducted in strict accordance with the guidelines outlined in the National Institutes of Health Guide for the Care and Use of Laboratory Animals and received ethical approval from the Experimental Animal Platform of the School of Medical Sciences, Zhengzhou University (Approval No. ZZU-LAC20231201[11]). All experimental procedures were performed in a specific pathogen-free (SPF) facility at the animal center. We utilized C57BL/6 mice

(7–8 weeks old, 25 g–30 g) of both sexes, obtained from Sbeff Biotechnology Co., Ltd (Beijing, China), along with adult male CaMKIIα-ires-Cre mice (NM-KI-242500) generously provided by Dr. Songxue Su from Zhengzhou University, originally sourced from Shanghai Model Organisms Centre, Inc. The animals were maintained under controlled environmental conditions (23 °C–28 °C, 50%–60% humidity) with a 12-h light/dark cycle and ad libitum access to food and water. Mice were housed in groups of 3–5 per cage, with random assignment to experimental groups. The sample size for each experiment was determined through power analysis, incorporating pilot data and previous studies, while adhering to the principle of minimizing animal usage. Experimental groups typically consisted of 5–6 mice, except for behavioral assessments which required 8–10 mice per group to ensure result reliability. A predefined exclusion criterion was implemented to omit animals that died from non-experimental factors (e.g., aggression-related injuries) from the final analysis. To ensure objectivity, all data collection was performed by investigators blinded to the experimental conditions and group assignments.

## Cell lines

All cell cultures were maintained under sterile conditions at 37 °C in a humidified atmosphere containing 5% $CO_2$. Cell line authentication was performed through short tandem repeat (STR) profiling within six months prior to experimentation, and routine Mycoplasma contamination testing was conducted using the Myco-Lumi Luminescent Mycoplasma Detection Kit (Beyotime Biotechnology Co., Ltd, Shanghai, China, C0297S). The human neuroblastoma SH-SY5Y cell line (CL-0208) and rat pheochromocytoma PC-12 cell line (CL-0412), along with their respective culture media (CM-0208 for SH-SY5Y and CM-0412 for PC-12), were obtained from Procell Life Technology Co., Ltd (Wuhan, China). Cells were cultured in Dulbecco's Modified Eagle Medium (DMEM) supplemented with 10% fetal bovine serum, 2 mmol/L glutamine (Gibco), and antibiotic-antimycotic solution containing penicillin (100 U/mL) and streptomycin (100 μg/mL). Medium replacement was performed twice weekly to maintain optimal cell growth conditions.

## Skin/muscle incision retraction mouse model

Following the protocol established by Flatters (Flatters, 2008), we performed a mouse model of chronic postoperative pain using the SMIR procedure. Briefly, mice were anesthetized with 2–3% isoflurane (RWD Life Science Co., Shenzhen, China) and positioned in dorsal recumbency. A 1 cm longitudinal incision was made adjacent to the saphenous vein in the right hindlimb, followed by blunt dissection to separate the superficial gracilis muscle from the adductor magnus tendon fascia. A micro distractor (RWD Life Science Co. R22030-03) was then implanted to maintain skin and muscle retraction at 1–1.5 cm for 60 min. Throughout the procedure, the surgical site was irrigated with sterile saline and protected with moistened gauze, while core body temperature was maintained using a heating pad. Following the retraction period, the wound was closed with sutures, and prophylactic antibiotics were administered to minimize infection risk. Sham-operated controls underwent identical procedures except for the muscle retraction step, involving only exposure of the adductor magnus tendon fascia.

## Behavioral testing

The PWT was used to evaluate mechanical allodynia in the mice. In short, according to Chaplan et al (Chaplan et al, 1994), the up-down method is used for the PWT. All the mice were placed in a Plexiglas box (Xinruan Information Technology Co., Ltd., Shanghai, China) with a metal net bottom for 30 min per day for 3 days before the tests. After habituation to the environment, a series of von Frey hairs with logarithmically increasing stiffnesses (0.04, 0.07, 0.2, 0.4, 0.6, 1, 1.4, and 2 g) (Aesthesio) were used to stimulate the hind paws. The filaments were applied perpendicular to the plantar surface with an appropriate force to bend it for 5 s. Rapid lifting or retraction of the area behind the paw was considered a positive reaction. After a positive reaction, a smaller filament was used. If no positive reaction was obtained, the larger filament was replaced in the next test. The interval between each test was 5 min.

PWL was used to evaluate thermal hypersensitivity. The mice were placed in individual test compartments (Xinruan Information Technology Co., Ltd. XR1880) on a temperature-controlled glass platform maintained at 30 °C, and the lateral plantar surface of the hind paw was stimulated with a radiant heat source directed through an aperture. The time elapsed from the initiation of the stimulus until paw withdrawal was recorded as the thermal paw withdrawal latency. Each hind paw was tested five times at 5-min intervals, and the withdrawal latency values were averaged. To avoid tissue damage from prolonged thermal stimuli, the cut-off latency was set at 15 s.

Affective-motivational pain behaviors were assessed using CPP testing. Following a 3-day habituation period, animals underwent preconditioning assessment, during which they were allowed free access to all chambers of the CPP apparatus for 15 min. Chamber occupancy was automatically recorded and analyzed using an automated tracking system (Xinruan Information Technology Co., Ltd., XR-XT401) to confirm the absence of baseline chamber preference. After preconditioning, animals were randomly assigned to either sham or SMIR surgical procedures and returned to their home cages for recovery. On POD6, the conditioning phase was initiated. Animals first received an injection of saline (200 μL) into the injured tissue and popliteal fossa, followed by immediate placement (<2 min) into the designated pairing chamber. After a 4-h interval, animals received an injection of lidocaine (1% w/v, 200 μL) into the same anatomical sites and were placed in the opposite chamber. Chamber assignments were systematically counterbalanced across experimental groups. The CPP test was conducted on POD7, 20 h after the final conditioning session. Animals were allowed free access to all chambers for 15 min, and chamber preference was quantified using VisuTrack software (Xinruan Information Technology Co., Ltd.). The time spent in each chamber was automatically recorded and analyzed to determine treatment-induced changes in chamber preference (Navratilova et al, 2012).

The OFT was conducted to assess locomotor activity and anxiety-like behaviors in mice. Following a 2–3 day acclimation period to the testing environment, mice were individually placed in a square arena (50 × 50 × 50 cm, Xinruan Information Technology Co., Ltd., XR-XZ301) for a 5-min test session. The apparatus was thoroughly cleaned with 70% ethanol between trials to eliminate olfactory cues. Behavioral parameters, including central zone entries, time spent in the central area, and total distance travelled, were automatically recorded and analyzed using VisuTrack software (Xinruan Information Technology Co., Ltd.).

Anxiety-related behaviors were further evaluated using the EPM test. The EPM apparatus (Xinruan Information Technology Co., Ltd., XR-XG201) consisted of four arms (40 cm length × 10 cm width) elevated 50 cm above the floor, with two opposing open arms and two enclosed arms featuring 20 cm high black walls. Each mouse was placed in the central platform facing an open arm and allowed to explore the maze for 5 min. The time spent in the open arms was quantified using VisuTrack software, following established protocols from previous studies (Rodgers and Dalvi, 1997).

The rotarod test was performed to assess potential effects of SMIR surgery on motor coordination of mice. Following established protocols (Zhao et al, 2025), mice were acclimated to the testing environment for 30 min prior to experimentation. Testing was conducted using a rotarod apparatus (Xinruan Technology Co., Ltd., Shanghai, XR-6C). Mice were initially acclimated to the rotating rod at a constant speed of 4 rpm. After adaptation, the rotation speed was progressively increased from 4 to 40 rpm over a 5-min period, during which the latency to fall was recorded. Each mouse underwent three trials with beyond 30-min inter-trial intervals, and the average latency across trials was calculated. Throughout testing, environmental conditions were maintained with minimal noise, and the apparatus was kept clean and free of olfactory cues.

## Stereoscopic surgery and cannula implantation

The mice were anesthetized with 2% isoflurane and maintained with 1% isoflurane. The mouse head was fixed in a stereotactic framework (RWD Life Science Co., Ltd.) and placed on a stereotactic fixation device. Erythromycin was applied to the eyes of the mice to prevent corneal dryness. All skull measurements were taken relative to the bregma. A 200-nL virus mixture was injected at a rate of 20 nL/min via a microelectrode (RWD Life Technology Co., Ltd.) to deliver the virus. After virus injection, the microelectrode was kept in place for 10 min to allow diffusion of the virus. ACC: anterior posterior (AP): +0.85 mm; medial lateral (ML): ±0.55 mm; dorsal ventral (DV): −2.00 mm.

For autophagy, pLenti-EGFP-mCherry-LC3B (OBiO Biotechnology Co., Ltd., Shanghai, China) was injected into the ACC. For the TMEM251 overexpression experiment, rAAV2/9-EF1α-mCherry-DIO-TMEM251 (BrainVTA Biotechnology Co., Ltd., Wuhan, China) was injected into the ACC. The data were excluded from the experiments when viral injections were inaccurate. For cannula implantation, a guide cannula (RWD Life Technology Co., Ltd.) was implanted into the ACC. After surgery, the mice were placed under a homeothermic heating pad until awakening and were monitored daily. Bafilomycin A1 (BafA1, MCE Biotechnology Co., Ltd., New Jersey, USA, HY-100558) was minimally infused into the ACC via a cannula. Amino-5-phosphonovaleric acid (AP-5, 50 mmol/L, 0.4 μL, MCE Biotechnology Co., Ltd., NJ, USA, HY-100714) or cyano-2,6-cyano-7-nitroquinoxaline-2,3-dione (CNQX, 20 mol/L, 0.4 μL, MCE Biotechnology Co., Ltd., NJ, USA, HY-15066) was infused into the ACC (Ren et al, 2022).

## Fiber photometry recording

For fiber photometry, rAAV2/9-CaMKIIα-GCaMp6m (BrainVTA Biotechnology Co., Ltd.) was injected into the ACC. The optical fiber was implanted into the ACC, 100–200 μm higher than the virus injection point (200 μm OD, 0.39 NA, Inper Life Technology Co., Ltd.,

Hangzhou, China). The mice were divided and fed separately. After the mice recovered, the dynamic changes in neuronal calcium were recorded via a fiber photometry system (Inper Life Technology Co., Ltd.) under filament stimulation. Light (470 nm, 45 mW) was used to detect calcium-dependent signals. Moreover, calcium-independent signals were detected via 410 nm light (25 mW) as a balanced control. During the recording, von Frey stimulation was applied to the planar surface of each mouse. A 0.6 g filament represented non-nociceptive stimuli, and a 2 g filament represented nociceptive stimuli. Each mouse underwent five repeated stimulation trials. We subdivided the data on the basis of behavioral events in individual experiments and obtained the calcium signal change value ($\Delta F/F$) by calculating $(F-F_0)/F_0$ with InperPlot software (Inper Life Technology Co., Ltd.). For baseline measurements, the $\Delta F/F$ of two seconds prior to von Frey stimulation was used.

## Chemogenetic experiments

For chemogenetic experiments, rAAV2/9-CaMKIIα-hM4D(Gi)-EGFP and rAAV2/9-CaMKIIα-hM3D(Gq)-mCherry was injected into the ACC. After stable expression of the virus was achieved, all the mice were adapted to the behavioral room one day in advance. Clozapine N-oxide (CNO) (3.3 mg/kg in 1% DMSO, MCE Co., Ltd.) was injected intraperitoneally into the mice, and the behavioral tests were conducted 30 min later (Jendryka et al, 2019).

## RNA interference (RNAi) experiment for mice

For *Tmem251* knockdown by small interfering RNA (siRNA) in mice, after several siRNAs were designed, qPCR was used to verify the knockdown efficiency of the siRNAs in PC-12 cells. The siRNA with the highest knockdown efficiency was selected for cholesterol condensation and 2'-OMe modification, which aimed to increase the transfection and knockdown efficiency of siRNAs in mice. Then, 1 μl of siTmem251 (50 μM) was injected into the ACC via stereotactic surgery. For siTMEM251-Mus (5'-3'): sense: CAGUGGGCUACU-GUAUUAUTT; antisense: AUAAUACAGUAGCCCACUGTT.

## Transmission electron microscopy (TEM)

The tissues were fixed in 3% glutaraldehyde, postfixed in 1% osmium tetroxide, dehydrated through a graded series of acetone, infiltrated with Epox 812 and embedded. Semithin sections were stained with methylene blue, and ultrathin sections were cut with a diamond knife and stained with uranyl acetate and lead citrate. The sections were examined via transmission electron microscopy (JEM-1400-FLASH, Japan).

## Virus and small interfering RNA (siRNA) transfection

For viral transfection, cells in good condition were uniformly inoculated into a 24-well plate. When the cells adhered to the well and the density reached 30–40%, the culture medium was replaced, and 1–2 μL of pLenti-EGFP-mCherry-LC3B stock solution was added to each well. After mixing, the cells were placed in a cell culture incubator and cultured for 12–16 h, after which the medium was replaced with fresh culture medium.

For siRNA transfection, cells in good condition were uniformly inoculated into a 24-well plate. When the cells adhered to the well and the density reached 60–80%, siRNA transfection was performed. The

transfection-promoting reagent GP-transfect-Mate (GenePharma Co., Ltd., Shanghai, China) and the siRNA stock solution were diluted with OPTI-MEM (Gibco, USA, 51985091). The culture medium was replaced with fresh OPTI-MEM, and the prepared siRNA mixture was added. The OPTI-MEM was replaced with complete culture medium after 4–6 h. PCR was performed after 24–72 h. siTMEM251 (5'–3'): sense: GUGGGAUUGUAUCUGUUAGTT; antisense: CUAACAGAUACAA UCCCACTT.

## Western blotting

Briefly, the mice were sacrificed after anesthesia at the expected time points. The ACC of each mouse was obtained and quickly placed in liquid nitrogen. The samples were subsequently placed in precooled RIPA lysis buffer (CWBIO Biotech Co., Ltd., Beijing, China, CW2333) for homogenization. The RIPA lysis buffer contained 0.1 mM phenylmethylsulphonyl fluoride (PMSF)-protease inhibitors (CWBIO Biotech Co., Ltd., CW2200S). The lysis mixture was centrifuged at 3000 rpm for 15 min at 4 °C, and the supernatant was collected. After the protein concentration was measured via the BSA method, the sample was heated at 99 °C for 5 min. A total of 20–40 μg of protein was loaded onto a 10% or 12% SDS polyacrylamide gel, and protein electrophoresis products were transferred to a PVDF membrane. The membrane was blocked with 5% bovine serum albumin for 1 h and incubated with the primary antibody at 4 °C overnight. The membranes were then washed three times with TBST and incubated with HRP-conjugated goat anti-rabbit/mouse IgG secondary antibodies for 2 h. The enhanced chemiluminescence method (Abbkine Scientific Co., Ltd., Wuhan, China, BMU102-CN) was used with a ChemiDoc™ machine (Bio-Rad, USA) to observe and detect the bands. The Coomassie blue or colorimetric model was used to obtain western blot markers to anchor the target molecular weight bands. The chemiluminescence model was used to obtain original western blots for analysis. The relative levels of the target protein were normalized to the GAPDH level via gray level analysis via ImageJ software (NIH, USA).

For cells, precooled RIPA lysis solution was added to six-well plates when the cell growth density reached 60–80%, after which the mixture was bubbled, and mixed well. The supernatant was collected after centrifugation for subsequent Western blotting experiments. For tissue lysosomal proteins, a lysosomal protein extraction kit (Solarbio Science and Technology Co., Ltd., Beijing, China, EX1231) was used to extract lysosomal proteins from the ACC, followed by Western blotting experiments.

## Immunohistochemistry

The mice were terminally anesthetized and perfused through the ascending aorta with normal saline, followed by 4% paraformaldehyde in 0.1 M phosphate-buffered saline. After perfusion, the brain was removed and postfixed in the same fixative for 12 h, which was then replaced with 30% sucrose phosphate-buffered saline for 48 h. ACC sections (25 μm) were cut on a cryostat (Leica, CM1950) and prepared for immunohistochemistry. After being washed with PBS, the sections were blocked with 5% goat serum in 0.3% Triton X-100 for 1 h at 37 °C and incubated with primary antibody overnight at 4 °C. The secondary antibody was incubated with the sections for 2 h at 37 °C. The brain slices were fixed with DAPI Fluoromount-G mounting medium

(SouthBiotech, USA, 0100-20). The stained sections were examined with a Nikon AIRMP⁺ laser confocal microscope (Nikon, Japan).

SH-SY5Y cells were digested, resuspended in 0.25% trypsin (2520011, Gibco, USA), and inoculated into a 24-well plate, from which slides containing polylysine were obtained (WHB Biotechnology Co., Ltd.). After 24 h of culture and a growth density of 60%-80%, the cells were fixed with 4% paraformaldehyde and subjected to immunohistochemical staining as previously described.

## In situ hybridization experiment

Frozen mouse brain slices were fixed with 4% PFA containing 1/1000 DEPC for 30 min at room temperature. The target nucleic acid fragments were exposed via gastric protease. Twenty microliters of prehybridization solution were added to each brain slice and incubated at 38 °C for 4 h. After removal of the prehybridization solution, 20 µL of hybridization solution was added to each brain slice and incubated at 38 °C overnight incubation. After incubation, the brain slices were washed continuously with 2X SSC, 0.5X SSC, or 0.2X SSC and incubated with diluted SABC-FITC at 37 °C for 30 min. The brain slices were fixed with DAPI Fluoromount-G mounting medium (SouthBiotech, 0100-20). The stained sections were examined via a Nikon AIRMP⁺ laser confocal microscope (Nikon, Japan). The target nucleic acid fragments were (1) 5'–3': CACAC TCCTT GAAAG CTCGG TTACT TTCCC TGCCC TTTTG; (2) 5'–3': ACTGT ATTAT CCCCA TATGC TTGGC AGTTA TCTGC AATCG; and (3) 5'–3': CATCA CCGGT TTCCC TTATT GTAAG ACTGC CAGCA CTGTA (Boster Biological Technology Co., Ltd.).

## Quantitative PCR (qPCR)

QPCR was performed on SH-SY5Y cells and ACC samples obtained from the mice. Total RNA was isolated via the standard method with a BEAsy™ Chloroform-free Universal Total RNA Extraction Kit (Bioeast Biotechnology Co., Ltd., M1TR03). The isolated RNA was reverse transcribed into cDNA with the GeniScript™ RT SuperMix with gDNA Eraser 2.0 Kit (Bioeast Biotechnology Co., Ltd., Hangzhou, China, M5RT04) following standard protocols. Real-time quantitative PCR (qPCR) was performed via synthetic primers (Sangon Biotechnology Co., Shanghai, China, Ltd.) and SYBR Green (Bioeast Biotechnology Co., Ltd., M4QS01) on a QuantStudio 5 Real-Time PCR Detection System (Thermo Fisher Scientific, USA). The relative expression levels of the genes were calculated and quantified via the $2 - \Delta\Delta CT$ method after normalization to the reference gene GAPDH.

The following primers were used: *Tmem251*(mouse), FORWARD-GGATTGGAGTGGGACTGTATTT, REVERSE-TGTGTCCATGTG GTTCCTTC; *Gapdh*(mouse), FORWARD-AACAGCAACTCCCACT CTTC, REVERSE-CCTGTTGCTGTAGCCGTATT; *Tmem251*(cell), FORWARD- ATGGGATGGATTGGAGTGGGATTG, REVERSE-CT CAGGGTGCTGTTGAATGTGTTC; and *Gapdh*(cell), FORWARD-ACACCCACTCCTCCACCTTTG, REVERSE- TCCACCACCCTGT TGCTGTAG.

## Statistical analysis

The sample size was calculated via power analysis and then adjusted on the basis of pilot data and prior studies. Statistical analyses were performed using GraphPad Prism software (version 9.0; GraphPad Software, CA, USA). All datasets were assessed for normality distribution prior to statistical analysis. Data are presented as mean ± standard error of the mean (SEM). The behavioral data were analyzed using either One-way or Two-way analysis of variance (ANOVA) with Bonferroni post hoc correction, or paired/unpaired Student's $t$ tests, as appropriate. Biochemical and photometric measurements were evaluated using two-tailed unpaired Student's $t$ tests or repeated-measures One-way ANOVA with Bonferroni correction. A probability value of $P < 0.05$ was considered statistically significant for all analyses.

## Graphics

The synopsis graphics were created with BioRender.com.

# Data availability

The original data, statistical details, and original images have been submitted as source data.

The source data of this paper are collected in the following database record: biostudies:S-SCDT-10_1038-S44319-025-00646-8.

# Peer review information

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

## Acknowledgements

We thank Dr. Song-xue Su (Ph.D., Department of Human Anatomy, Zhengzhou University, China) for providing CaMKII-IRES-Cre mice, Dr. Wen-chao Zhao (Ph.D., Department of Physiology, Zhengzhou University, China) for providing technical assistance with the cell experiments, and Dr. Shu-sheng Zhang (Ph.D., Modern Analysis and Gene Sequencing Center, Zhengzhou University, China) for providing technical assistance with transmission electron microscopy. This work was supported by the National Natural Science Foundation of China [82002086, 82001187, 82071240], the Health Young and Middle-aged Discipline Leader Project of Henan Province [HNSWJ-2021012], the Medical Science and Technology Research Project of Henan Province (SBGJ202403023), and the Young and Middle-aged Health Science and Technology Innovation Leading Talent Project of Henan Province [YXKC2020059].

## Author contributions

**Yaowei Xu**: Conceptualization; Investigation; Visualization; Methodology; Writing—original draft. **Fei Xing**: Conceptualization; Formal analysis; Funding acquisition; Writing—review and editing. **Xin Wei**: Conceptualization; Formal analysis; Funding acquisition; Writing—review and editing. **Xiaoling Wang**: Investigation; Methodology. **Xiaoshan Shi**: Investigation; Methodology. **Zhongyu Wang**: Supervision; Validation. **Na Xing**: Supervision; Validation. **Jingjing Yuan**: Supervision; Validation. **Zhisong Li**: Supervision; Funding acquisition; Investigation; Writing—review and editing. **Wei Zhang**: Supervision; Funding acquisition; Project administration; Writing—review and editing.

Source data underlying figure panels in this paper may have individual authorship assigned. Where available, figure panel/source data authorship is listed in the following database record: biostudies:S-SCDT-10_1038-S44319-025-00646-8.

## Disclosure and competing interests statement

The authors declare no competing interests.

