## [Peer Review File · EMBO Reports]

TMEM251 loss-induced autophagy dysfunction in the anterior cingulate cortex contributes to chronic postoperative pain

Yaowei Xu, Fei Xing, Xin Wei, Xiaoling Wang, Xiaoshan Shi, Zhongyu Wang, Na Xing, Jingjing Yuan, Zhisong Li, and Wei Zhang

Corresponding author(s): Wei Zhang (lzs zd@gs.zzu.edu.cn) , Zhisong Li (wxlzd@gs.zzu.edu.cn)

Review Timeline:

Submission Date:	10th Apr 25
Editorial Decision:	10th Jun 25
Revision Received:	27th Aug 25
Editorial Decision:	7th Oct 25
Revision Received:	14th Oct 25
Accepted:	22nd Oct 25

Transaction Report:

Dear Dr. Zhang

Thank you for the submission of your research manuscript to our journal. We have now received the full set of referee reports that is copied below.

As you will see, the referees acknowledge that the findings are interesting but they also raise a number of concerns and have suggestions how to strengthen the data. While I realize that a significant revision will be required to address all concerns, I would like to give you the chance to revise your manuscript with the understanding that the referee concerns (as detailed above and in their reports) must be fully addressed and their suggestions taken on board. Please address all referee concerns in a complete point-by-point response. Acceptance of the manuscript will depend on a positive outcome of a second round of review. It is EMBO Reports policy to allow a single round of revision only and acceptance or rejection of the manuscript will therefore depend on the completeness of your responses included in the next, final version of the manuscript.

We realize that it is difficult to revise to a specific deadline. In the interest of protecting the conceptual advance provided by the work, we recommend a revision within 3 months (September 10th). Please discuss the revision progress ahead of this time with the editor if you require more time to complete the revisions.

I am also happy to discuss the revision further via e-mail or a video call, if you wish.

=====
IMPORTANT NOTE:

We perform an initial quality control of all revised manuscripts before re-review. Your manuscript will FAIL this control and the handling will be delayed IN CASE the following APPLIES:

- 1) A data availability section providing access to data deposited in public databases is missing. If you have not deposited any data, please add a sentence to the data availability section that explains that.
- 2) Your manuscript contains statistics and error bars based on $n=2$. Please use scatter blots in these cases. No statistics should be calculated if $n=2$.

=====

- 1) a .docx formatted version of the manuscript text (including legends for main figures, EV figures and tables). Please make sure that the changes are highlighted to be clearly visible.
- 2) individual production quality figure files as .eps, .tif, .jpg (one file per figure). Please download our Figure Preparation Guidelines (figure preparation pdf) from our Author Guidelines pages <https://www.embopress.org/page/journal/14693178/authorguide> for more info on how to prepare your figures.
- 3) a .docx formatted letter INCLUDING the reviewers' reports and your detailed point-by-point responses to their comments. As part of the EMBO Press transparent editorial process, the point-by-point response is part of the Review Process File (RPF), which will be published alongside your paper.
- 4) a complete author checklist, which you can download from our author guidelines (<<https://www.embopress.org/page/journal/14693178/authorguide>>). Please insert information in the checklist that is also reflected in the manuscript. The completed author checklist will also be part of the RPF.
- 5) Please note that all corresponding authors are required to supply an ORCID ID for their name upon submission of a revised

manuscript (<<https://orcid.org/>>). Please find instructions on how to link your ORCID ID to your account in our manuscript tracking system in our Author guidelines (<<https://www.embopress.org/page/journal/14693178/authorguide#authorshipguidelines>>)

6) We replaced Supplementary Information with Expanded View (EV) Figures and Tables that are collapsible/expandable online. A maximum of 5 EV Figures can be typeset. EV Figures should be cited as 'Figure EV1, Figure EV2' etc... in the text and their respective legends should be included in the main text after the legends of regular figures.

7) Please note that a Data Availability section at the end of Materials and Methods is now mandatory. In case you have no data that requires deposition in a public database, please state so instead of refereeing to the database. See also < <https://www.embopress.org/page/journal/14693178/authorguide#dataavailability>>. Please note that the Data Availability Section is restricted to new primary data that are part of this study.

Additional information on source data and instruction on how to label the files are available <<https://www.embopress.org/page/journal/14693178/authorguide#sourcedata>>

10) Figure legends and data quantification:
The following points must be specified in each figure legend:

- the name of the statistical test used to generate error bars and P values,
 - the EXACT p-values,
 - the number (n) of independent experiments (please specify technical or biological replicates) underlying each data point,
 - the nature of the bars and error bars (s.d., s.e.m.)
- If the data are obtained from n {less than or equal to} 5, show the individual data points in addition to the SD or SEM.
- If the data are obtained from n {less than or equal to} 2, use scatter blots showing the individual data points.

See also the guidelines for figure legend preparation:
<https://www.embopress.org/page/journal/14693178/authorguide#figureformat>

11) Our journal encourages inclusion of *data citations in the reference list* to directly cite datasets that were re-used and obtained from public databases. Data citations in the article text are distinct from normal bibliographical citations and should directly link to the database records from which the data can be accessed. In the main text, data citations are formatted as follows: "Data ref: Smith et al, 2001" or "Data ref: NCBI Sequence Read Archive PRJNA342805, 2017". In the Reference list, data citations must be labeled with "[DATASET]". A data reference must provide the database name, accession number/identifiers and a resolvable link to the landing page from which the data can be accessed at the end of the reference. Further instructions are available at <<https://www.embopress.org/page/journal/14693178/authorguide#referencesformat>>.

12) All Materials and Methods need to be described in the main text using our 'Structured Methods' format. According to this format, the Methods section includes a Reagents and Tools Table (listing key reagents, experimental models, software and relevant equipment and including their sources and relevant identifiers) followed by a Methods and Protocols section describing the methods, ideally using a step-by-step protocol format. The aim is to facilitate adoption of the methodologies across labs. Please download and fill our Reagents and Tools Table template (.docx), which you can find in our author guidelines:

13) As part of the EMBO publication's Transparent Editorial Process, EMBO Reports publishes online a Review Process File to accompany accepted manuscripts. This File will be published in conjunction with your paper and will include the referee reports, your point-by-point response and all pertinent correspondence relating to the manuscript.

Yours sincerely,

=====

Referee #1:

The current manuscript seeks to address how altering autophagy processes contributes to the production of pain, with a focus on (1) how autophagy is dysregulated in chronic postoperative pain and (2) how autophagy may be acting on the anterior cingulate cortex (ACC) to alter neural pain processing.

The authors focus on neuronal autophagy as a contributor to chronic pain is novel, timely, and interesting.

The authors use behavioral, molecular, genetic, and neurophysiological approaches to dissect how autophagy contributes to altered ACC activity and thereby to the maintenance of chronic pain.

They use a model of chronic postoperative pain (skin/muscle incision and retraction (SMIR)) that involves a minor surgical perturbation of the hindlimb muscles.

One of the holes in the manuscript is that it is left unsaid what exactly the increased autophagy is attempting to degrade? For example, are membrane proteins turning over faster from increased use due to higher neural activity in chronic pain conditions? What is in these autophagosomes?

Another major hole in the manuscript is that they reference that 80% of activated neurons are CamKIIa+ versus 20% that are GAD67+. They use this as evidence that excitatory neurons have preferential activation. However, this is exactly the number one would expect if both populations were similar activated. This is because cortical circuits are approximately 80% excitatory neurons and 20% inhibition neurons, by cell number. So the reasoning to focus on excitatory neurons over inhibitory neurons does not make sense. Also, looking through the supplemental data, I would not find the data they are referring to.

The BafA1 experiments are difficult to interpret because it is possible that the major effect is actually due to dysregulation of intracellular calcium dynamics rather than changes in autophagy. See comments below. I would recommend moving all BafA1 experiments to supplemental data, as the TMEM251 experiments are sound and likely do not run into similar issues.

I believe the paper would be acceptable for publication after major revisions.

Comments by figure:

In brief, the results presented by the authors are, by figure #:

Figure 1: Reproduction that a model of postoperative pain (SMIR) produces pain behavior phenotypes in mice, specifically:

- von Frey (vF) based mechanical hypersensitivity testing, where they find that both Male mice and Female mice both exhibit mechanical hypersensitivity following SMIR. The vF data indicate sex specific differences in the timing of healing to SMIR, however it is not clear if the two datasets (from Fig 1A and Fig 1G) are directly statistically compared to prove or disprove this point.
- Conditioned place preference (CPP) testing, which demonstrates that the SMIR have a preference for chambers that air paired with the injection of lidocaine into the injured hindpaw, but sham mice do not. This is true for both male and female mice.
- SMIR does not produce significant changes in anxiety related or gross locomotor behaviors, as measured by open field test and elevated plus maze. The authors consistently call these tests "emotional" tasks, but really they are anxiety related and should be indicated as such.
- - It does seem that the experiments in I and K may be underpowered to conclude that no change takes place. Can the authors please provide a power analysis for panels I and J in figure 1?

Figure 2: Using western blot, immunohistochemistry, and electron microscopy, the authors demonstrate that autophagy is impaired in the ACC after SMIR. Additionally, using a viral approach, they attempt to demonstrate that LC3B colocalizes with excitatory neurons to a greater degree in SMIR mice.

- Panels A through E are convincing evidence of their claims.
- However, from panels F on, the reviewer would like to raise concerns. This issue will continue to arise throughout the paper.
- - The authors use a viral approach to track LC3B, an autophagy related protein that is tagged with mCherry. A major problem here is that the injection of the virus into the ACC is going to cause an injury to the cortex, which is going to alter the native state of the ACC, likely biasing it toward autophagy as the neurons heal following the injection. Therefore, these results are likely biased by the injection of virus into the ACC. The virus will be most concentrated in the injection track, specifically where the most injury will occur. The injection track is clearly visible, indicating some level of injury response (immune infiltration, among others). On top of this, the virus, and the viral overexpression of proteins, will likely increase the level of autophagy. A transgenic line for this protein exists (PMID: 18425446), so an approach that involves injury to the cortex was not necessary.

Figure 3: The authors then explore the use of bafilomycin A1 (BafA1) as an inhibitor of autophagy, and whether direct injection of BafA1 into the ACC produces a similar phenotype to SMIR.

- The data are convincing that BafA1 inhibits autophagy (which was already known).
- The data that BafA1 injection into the ACC produces a chronic pain phenotype are also convincing.
- - However, the reviewer has issue with panel K. The representative image clearly demonstrates a very high degree of overlap between GFP and mCherry (~90-100%). The images from panel 1H and 5G do not show such a high degree of overlap. Did the authors use different threshold levels to perform their quantifications specifically for this panel? I am highly skeptical about the results pertaining to Figure 3J-K due to major differences between representative images in different figures, but similar quantification percentages.

Figure 4: The authors then use viral genetic tools to overexpress TMEM251 in the ACC, and demonstrate that overexpression blunts the maintenance of chronic pain after SMIR.

- The data here are convincing. Although there remains the issue of potentially increased autophagy after viral injection in and of itself, the experiment is well controlled with scrambled control virus, and the lack of effect in Sham mice for both TMEM251 and scrambled overexpression viruses.

Figure 5: TMEM251 knockdown using siRNA produces mechanical hypersensitivity.

- The data here are convincing, with a few minor issues.
- - The quantification for panel 1A, the unit is inches. Instead, do they mean pixels? Microns? Or are these inches on the computer screen? If the later, this is not a useful measurement, and should be converted into microns.
- - As the siRNA are delivered virally, would one not expect a more prolonged effect? It seems that mechanical hypersensitivity ends after 10 days.

Figure 6: Here, using calcium imaging, they take a neurophysiological approach to illuminate how stimulus evoked activity in the ACC is altered by SMIR and the role of TMEM251 in maintaining SMIR induced changes in nociceptive processing.

- As expected, they find enhanced activation of the ACC in chronic pain (this is well known).
- The overexpression of TMEM251 reduces SMIR-enhanced ACC stimulus evoked activity.
- In the absence of chronic pain, knockdown of TMEM251 increases ACC evoked activity, however, the increase seems less than occurs during SMIR alone, indicating partial contribution to altered ACC nociceptive processing.
- Direct administration of BafA1 increases evoked ACC activity.
- - There is an issue with this experiment though, and which is a larger issue for the manuscript. BafA1 alters intracellular calcium dynamics (PMID: 25337829). Therefore, using calcium imaging is not an appropriate way to identify how BafA1 contributes to altered nociceptive processing via autophagy, as the change in evoked responses is likely due to changes in intracellular calcium buffering. This is clearly evident, as dF/F values for the BafA1 experiment are 2-fold higher (see the scale bar for Panel S and compare to those for all other panels in the figure). This experiment is poorly conceived and should not be

included in the paper.

- - However, the larger issue is whether the administration of BafA1 provides useful information specifically about autophagy, or if the results are more indicative of altered intracellular calcium buffering within the ACC.

Figure 7: Lastly, the authors replicate previous literature to show that decreasing bulk ACC activity can decrease pain behaviors, and then demonstrate that inhibiting ACC activity can reverse chronic pain phenotypes produced by the siRNA-mediated knockdown of TMEM251.

- I'm not sure how informative this figure is. Decreasing bulk ACC activity is a well known way to produce analgesia. Showing that decreasing ACC activity after TMEM251 knockdown or BafA1 infiltration produces analgesia does not necessarily prove that inhibition is working specifically to reverse the autophagy phenotype. As a thought experiment, ablating the ACC would have had a similar effect, but one would not conclude that ablations are useful for assessing the contribution of autophagy to ACC pain processing.

- I think more useful here would be to see if altering neuronal activity directly within the ACC would produce changes in autophagy. For example, is the reason autophagy is dysregulated in chronic pain that ACC neurons fire more, and therefore proteins are degraded faster and have higher need to be cleared? If so, increasing ACC activity using excitatory DREADDs should increase autophagy demand (which would show up in their previous assays). Decreasing ACC activity may, through a similar mechanism, decrease the autophagy load.

Comments for text:

Line 37: Pharmacological should be pharmacogenetic is referring to the DREADD experiments.

Line 102: I think CPP would be better referred to as a test of affective-motivational pain behaviors, and not necessarily spontaneous pain. Spontaneous pain usually refers to tests where specific pain behaviors are monitored after injury or injection of an algogen or pruritogen.

Line 109: The open field test and elevated plus maze are generally considered to be tests for anxiety, not general emotional tests. This same issue occurs in a few places throughout the manuscript.

Line 125: The authors mention that lack of motor impairment due to SMIR model, but only point to open field data, which will give very limited information on motor impairment outside of the animals overall motivation to move (distance traveled). Did the authors try placing the mouse on a rotarod assay to test coordination?

Line 162-164: It is difficult to understand what is being said.

Line 167-169: Male and female mice clearly have different behavioral responses to SMIR and autophagy related responses. I would be useful to provide data on female mice for some of their more important findings to ensure that their findings are not sex specific. Specifically, the TMEM251 overexpression and siRNA knockdown experiments, as related to their alteration of pain behaviors.

Line 189: The authors mention "autophagy substrate", it would be helpful if this is more clearly defined. In fact, a graphical representation of all of the proteins involved and their relation to one another in the context of autophagy would considerably improve the manuscript.

Line 290: The wording for this header is confusing, please rephrase.

Line 584-586: After reading these lines, I was worried that this discussion was, at least partly, written by AI and not read over carefully. How does this paper in any way show how mechanical pressure sensitive channels in sensory neurons change? And how do they relate to temperature sensitive channels? I'm very concerned that this bit is in the manuscript and did not catch any of the authors' attention. In general, the discussion is difficult to read and should be rewritten and more closely edited.

Referee #2:

In this report authors found that autophagy damage in the ACC after peripheral injury contribute to chronic postsurgical pain. While the study is potentially interesting, the report is too preliminary for publication in the present form. Additional experiments are needed to confirm key findings.

Major:

1. It is still unclear how these autophagy damage affect ACC excitability. Electrophysiological experiments are needed to determine if the impact is on excitatory vs inhibitory synaptic transmission, as well as possible presynaptic changes of circuits within the ACC.

2. ACC deep neurons may project to subcortical areas as well as spinal dorsal horn neurons (Chen et al., 2018, Nature Communication). Authors need to perform detailed analyses if superficial layers of cells (II/III) and deep projection cells are all

affected by autophagy damage. Additional discussion of related to ACC projected network in chronic postsurgical pain should be added.

3. Insular cortex plays important roles in pain modulation and perception. Some comparison or additional data from the IC would increase impact of the study.

Dear Editor and Reviewers,

Re: Response to the manuscript “*TMEM251 deficiency-induced autophagy damage in anterior cingulate cortex contributes to chronic postoperative pain*” (ID: EMBOR-2025-61719V1).

We greatly appreciate the time and effort invested by the editors and reviewers in the manuscript. We cherish this precious opportunity for modification. The reviewers carefully reviewed each figure and paragraph, providing us with many valuable suggestions and modification plans. We are very grateful for the meticulousness and responsibility of the reviewers. We have revised the manuscript, according to these modification plans, which undoubtedly helped to greatly improve the quality of our manuscript. We have provided point-by-point responses to the questions raised by reviewers, and the revised portions are highlighted in **red**.

We sincerely hope that these responses can alleviate your concerns, and look forward to hearing from you.

Sincerely,

Dr. Wei Zhang

Professor, Department of Anesthesiology, Pain and Perioperative Medicine

The First Affiliated Hospital of Zhengzhou University

No. 1 East Construction Road, Zhengzhou, Henan, China

Email: zhangw571012@126.com

Encl. Responses to the comments from Reviewers 1 and 2.

Reply to Reviewer 1

Reviewer point #1: The current manuscript seeks to address how altering autophagy processes contributes to the production of pain, with a focus on (1) how autophagy is dysregulated in chronic postoperative pain and (2) how autophagy may be acting on the anterior cingulate cortex (ACC) to alter neural pain processing.

The authors focus on neuronal autophagy as a contributor to chronic pain is novel, timely, and interesting.

The authors use behavioral, molecular, genetic, and neurophysiological approaches to dissect how autophagy contributes to altered ACC activity and thereby to the maintenance of chronic pain.

They use a model of chronic postoperative pain (skin/muscle incision and retraction (SMIR)) that involves a minor surgical perturbation of the hindlimb muscles.

Author response #1: We sincerely appreciate your thorough and insightful comments, which have markedly improved the quality and clarity of our manuscript. In addition, we are pleased to receive your recognition of “The authors focus on neural autophagy as a contributor to chronic pain is novel, timely, and interesting.”. We have carefully discussed each of your comments. And based on these comments, relevant experiments were supplemented and the manuscript was revised. Sincerely hope that these modifications could alleviate your concerns.

Reviewer point #2: One of the holes in the manuscript is that it is left unsaid what exactly the increased autophagy is attempting to degrade? For example, are membrane proteins turning over faster from increased use due to higher neural activity in chronic pain conditions? What is in these autophagosomes?

Author response #2: Thanks very much for your valuable suggestion that “what

exactly the increased autophagy is attempting to degrade”. The suggestions of you have inspired us that “For example, are membrane proteins turning over faster from increased use due to higher neural activity in chronic pain conditions?”. The literatures have suggested that autophagy maintains synaptic function, plasticity, and energy supply by dynamically regulating the degradation and regeneration of synaptic components. Its functional impairment is associated with various neurological disorders (Kuijpers et al., 2021; Nikolettou and Tavernarakis, 2018). Autophagy impairments may lead to abnormal aggregation of synaptic proteins, causing functional impairment. Therefore, we speculate that the increase in autophagy within the ACC in the context of chronic postoperative pain may be related to the regulation of synaptic components.

To verify this hypothesis, we used LC3B and synaptophysin for immune co-staining. Synaptophysin is an important synaptic vesicle membrane protein widely distributed in neuroendocrine cells and neurons, involved in the secretion, endocytosis, and circulation of synaptic vesicles, and regulating neurotransmitter release (Wang et al., 2024a). Compared with the Sham group, the SMIR group presented significantly increased co-localization signals of LC3B and synaptophysin, indicating that synaptophysin may be one of the increased autophagy substrates of the ACC in SMIR mice (V2-Figure. 2E).

To further clarify whether excitatory or inhibitory synapses are involved in autophagy degradation, we analyzed the expression of excitatory synaptic proteins (PSD95,

VGLUT1) and inhibitory proteins (Gephyrin, VGAT) using western blot. In the ACC of SMIR mice, PSD95 and VGLUT1 were significantly up-regulated, whereas Gephyrin and VGAT remained unchanged (V2-Appendix Figure. S2E). Furthermore, immunostaining showed a significant increase in co-localization signals between LC3B and PSD95, while there was no significant change in co-localization signals with Gephyrin (V2-Figure. 2F and V2-Appendix Figure. S2G). These results suggest that under CPOP state, autophagy may primarily regulate the renewal of excitatory synaptic proteins. The inhibition of autophagy flux may promote synaptic function enhancement by accumulating synaptic proteins. This is consistent with previous studies that peripheral nerve injury promotes the accumulation and insertion of AMPA receptors in the V-layer neurons of the ACC, facilitating the transmission of nociceptive signals (Chen et al., 2014a; Chen et al., 2014b). These results suggest that the accumulation of excitatory synaptic proteins may promote the transmission of nociceptive signals. The revision of the manuscript is presented in Line 157-164 and the screen dump is provided below for your quick reference.

in the SMIR group (Figure. 2D). Importantly, we observed a significant increase in the
 colocalization signal of LC3B and synaptophysin in the SMIR group (Figure. 2E).
 SMIR surgery increased the expression of postsynaptic density protein-95 (PSD95) and
 vesicular glutamate transporter 1 (VGLUT1) in the ACC, but did not affect the
 expression of Gephyrin and vesicular GABA transporter (VGAT) (Appendix Figure.
 S2E). Consistently, the colocalization signals of LC3B and PSD95, rather than those of
 Gephyrin, significantly increased in the SMIR group (Figure. 2F and Appendix
 Figure. S2G).

In addition, overexpression of TMEM251 can alleviate the accumulation of synaptic proteins in autophagy substrates (V2-Figure. 5L and V2-Appendix Figure. S4E-S4G). The revision of the manuscript is presented in Line 256-260 and the screen dump is provided below for your quick reference.

increasing autolysosomes in ACC neurons (**Figure. 5K and 5M**). **Western blot analysis**
**also demonstrated that TMEM251 overexpression significantly reduced the expression**
**of PSD95 and VGLUT1 in SMIR group (Appendix Figure. S4E)**. Furthermore, the
**overexpression of TMEM251 reduced the accumulation of synaptophysin and PSD95**
**in autophagy substrates of SMIR mice (Figure. 5L and Appendix Figure. S4F-S4G).**

However, knocking down TMEM251 can also observe similar results to the SMIR group (**V2-Figure. 6J and V2-Appendix Figure. S5D-S5F**). The revision of the manuscript is presented in Line 287-292 and the screen dump is provided below for your quick reference. We thank you again for this constructive suggestion.

test (**Appendix Figure. S5B and S5C**). **Western blot analysis demonstrated that**
**TMEM251 knockdown significantly increased the expression of PSD95 and VGLUT1**
**in naïve mice (Appendix Figure. S5D)**. Furthermore, RNAi-TMEM251 also induced
**the accumulation of synaptophysin and PSD95 in autophagy substrates, which is**
**consistent with the findings in SMIR mice (Figure. 6J and Appendix Figure. S5E-**
**S5F)**. These findings collectively demonstrate that TMEM251-deficiency in the ACC

Reference #2:

1. Kuijpers M, Kochlamazashvili G, Stumpf A, Puchkov D, Swaminathan A, Lucht MT, et al. Neuronal Autophagy Regulates Presynaptic Neurotransmission by Controlling the Axonal Endoplasmic Reticulum. *Neuron*. 2021; 109: 299-313.e9.
2. Nikolettou V, Tavernarakis N. Regulation and Roles of Autophagy at Synapses. *Trends Cell Biol*. 2018; 28: 646-61.

3. Wang C, Jiang W, Leitz J, Yang K, Esquivies L, Wang X, et al. Structure and topography of the synaptic V-ATPase-synaptophysin complex. *Nature*. 2024; 631: 899-904.
4. Chen T, Koga K, Descalzi G, Qiu S, Wang J, Zhang LS, et al. Postsynaptic potentiation of corticospinal projecting neurons in the anterior cingulate cortex after nerve injury. *Mol Pain*. 2014; 10: 33.
5. Chen T, Wang W, Dong YL, Zhang MM, Wang J, Koga K, et al. Postsynaptic insertion of AMPA receptor onto cortical pyramidal neurons in the anterior cingulate cortex after peripheral nerve injury. *Mol Brain*. 2014; 7: 76.

Reviewer point #3: Another major hole in the manuscript is that they reference that 80% of activated neurons are CamKIIa+ versus 20% that are GAD67+. They use this as evidence that excitatory neurons have preferential activation. However, this is exactly the number one would expect if both populations were similar activated. This is because cortical circuits are approximately 80% excitatory neurons and 20% inhibition neurons, by cell number. So, the reasoning to focus on excitatory neurons over inhibitory neurons does not make sense. Also, looking through the supplemental data, I would not find the data they are referring to.

Author response #3: We are very grateful to you for pointing out this important problem. To clarify the distribution of activated neuron types in the ACC of SMIR mice, we compared the activation of CaMKII-positive neurons and GAD67-positive neurons in Sham group and SMIR group, respectively, following previous study (Gu et al., 2023). **Compared with the Sham group, the SMIR group presented a significantly increased count of activated CaMKII-positive neurons, while the count of activated GAD67-positive neurons remained unchanged (V2-Figure. 7A and V2-Appendix Figure. S6B).** The revision of the manuscript is presented in Line 301-304 and the screen dump is provided below for your quick reference. The raw data and statistical details are presented in the source data (**Figure. 7 and Appendix Figure. S6**).

(Appendix Figure. S6A). Notably, the number of activate
 significantly increased in the SMIR group, could be reduce
 TMEM251 (Figure. 7A and 7B). However, there was no
 number of activated GAD67-positive cells (Appendix Fi

Furthermore, as you noted the majority of neurons in the ACC are pyramidal excitatory neurons (70%-80%), while inhibitory neurons account for approximately 20%. Therefore, we synthesized the research results and anatomical features of the ACC, with CaMKII-positive neurons as the main research subjects. The ultimate source of the impact of neuronal activation on mice lies in the bidirectional regulation of excitation and inhibition. Therefore, it cannot be ruled out that inhibitory neurons represented by GAD67 positive neurons do not play a role in CPOP. In future research, we will delve into the excitation/inhibition imbalance of ACC neurons. Once again, thank you for drawing our attention to this crucial point.

Reference #3:

1. Gu HW, Zhang GF, Liu PM, et al. Contribution of activating lateral hypothalamus-lateral habenula circuit to nerve trauma-induced neuropathic pain in mice. *Neurobiol Dis.* 2023;182:106155. doi:10.1016/j.nbd.2023.106155.

Reviewer point #4: The BafA1 experiments are difficult to interpret because it is possible that the major effect is actually due to dysregulation of intracellular calcium dynamics rather than changes in autophagy. See comments below. I would recommend moving all BafA1 experiments to supplemental data, as the TMEM251

experiments are sound and likely do not run into similar issues.

Author response #4: Thank you very much for the key suggestions. **We have transferred BafA1-related research to supplementary materials (V2-Appendix Figure. S3).** The results of reduced lysosomal localization of CTSB, and TMEM251 expression are presented in the manuscript as **V2-Figure. 4**. The revision of the manuscript is presented in Line 187-229 and the screen dump is provided below for your quick reference. We sincerely thank you again for your invaluable suggestions.

To investigate the causal relationship between impaired autophagy and CPOP, we
employed bafilomycin A1 (BafA1), a specific inhibitor of lysosomal vacuolar-ATPase
[20]. The blocking effect of BafA1 on autophagy flux was validated in cellular
experiments (**Appendix Figure. S3A**). Microinfusion cannulas were surgically
implanted into the ACC, followed by the administration of BafA1 at varying
concentrations via a microinfusion pump (**Appendix Figure. S3B**). Quantitative
analysis revealed a significant reduction in the PWT from day 3 to day 10 post-infusion
in 10ng and 50ng groups (**Appendix Figure. S3C and S3D**). PWL analysis showed
significant decreases in the 10ng and 50ng groups from day 3 to day 7 (**Appendix**
**Figure. S3C and S3D**). Notably, BafA1 administration had no significant effect on
motor coordination or anxiety-like behaviors of mice (**Appendix Figure. S3E and**
**S3F**). Quantitative analysis revealed that BafA1 administration (10 ng and 50 ng doses)
significantly upregulated the autophagy levels and inhibited autophagy flux (**Appendix**
**Figure. S3G and S3H**). Notably, SQSTM1 expression exhibited a temporal pattern of
upregulation, with significant increases observed between day 3 and day 7, rather than
on day 14, which paralleled the PWT changes in the 10ng treatment group (**Appendix**
**Figure. S3H**). Furthermore, regression analysis demonstrated a significant negative
correlation between SQSTM1 levels and left PWT in the 10ng group (**Appendix**
**Figure. S3I**). Fluorescence analysis also revealed significantly more EGFP-mCherry-
LC3B puncta in BafA1 group than in the control group (**Appendix Figure. S3J and**
**S3K**). These findings provide that obstructive autophagy flux may be an important
factor in inducing pain hypersensitivity.↵
Lysosomes serve as the primary degradation sites for autophagosomes. To elucidate
the mechanism underlying impaired autophagy in the ACC of SMIR mice, we assessed
lysosomal activity via the following key molecular markers: LAMP1, ATPase H⁺
transporting V0 subunit D1 (ATP6V0D1, lysosome acidification), lysosomal associated
membrane protein 2A (LAMP2A, lysosome membrane fusion), and cathepsin B (CTSB,
lysosome hydrolase). Quantitative analysis revealed a significant increase in LAMP1
expression in the ACC of SMIR mice, while other markers remained unchanged
(**Figure. 4A**). Subsequent western blot analysis of lysosomal proteins demonstrated a
substantial reduction in CTSB levels in lysosomes of SMIR group (**Figure. 4B**).
Immunofluorescence analysis further confirmed a significant decrease in CTSB and
LAMP1 colocalization signals in the SMIR group, independent of the CTSB expression
level (**Figure. 4C**). These results suggest the incorrect sorting of lysosomal hydrolases
in the ACC of SMIR mice. On the basis of previous studies demonstrating the role of
TMEM251 in mediating lysosomal hydrolases sorting [15, 16], we investigated its
potential involvement in autophagy impairment under CPOP conditions. Quantitative
PCR analysis revealed a significant reduction in *Tmem251* mRNA levels in the ACC of
SMIR mice at POD7 and POD14, paralleling alterations in autophagy (Figure. 4D).
Furthermore, in situ hybridization analysis demonstrated decreased TMEM251
expression, specifically in CaMKII α -positive neurons, but not in GAD67-positive
neurons, in SMIR mice (Figure. 4E and 4F). The change of TMEM251 expression was
consistent with the accumulation of autophagy substrates in the ACC of SMIR mice.↵

Reviewer point #5: I believe the paper would be acceptable for publication after major revisions.

Author response #5: We are deeply moved to receive your encouragement. Thank you once again for your professional guidance and dedication. I hope that our modifications can alleviate your concerns.

Reviewer point #6: Figure1: Reproduction that a model of postoperative pain (SMIR) produces pain behavior phenotypes in mice, specifically:

von Frey (vF) based mechanical hypersensitivity testing, where they find that both Male mice and Female mice both exhibit mechanical hypersensitivity following SMIR. The vF data indicate sex specific differences in the timing of healing to SMIR, however it is not clear if the two datasets (from Fig 1A and Fig 1G) are directly statistically compared to prove or disprove this point.

Author response #6: Thank you very much for your professional suggestion. Based on your suggestion, we chose POD7 and POD21 to directly compare the ipsilateral PWT of Sham-male, SMIR-male, Sham-female, and SMIR-female. At POD7, SMIR

mice of both sexes exhibited a markedly lower ipsilateral PWT than their respective Sham controls, with no sex difference within the SMIR cohort. **At POD21, SMIR females alone retained this mechanical hypersensitivity, displaying a significantly lower ipsilateral PWT than both Sham females and SMIR males.** These findings indicate that female mice are more sensitive to mechanical hypersensitivity induced by SMIR surgery.

Reviewer point #7: Conditioned place preference (CPP) testing, which demonstrates that the SMIR has a preference for chambers that air paired with the injection of lidocaine into the injured hindpaw, but sham mice do not. This is true for both male and female mice.

Author response #7: Thanks very much for your constructive comment. Because

SMIR model induces chronic pain by continuous stretching of peripheral muscles. Nociceptive signals are transmitted to the brain through the sciatic nerve and spinal dorsal horn. The use of lidocaine at the surgical site and sciatic nerve can block the transmission of nociceptive signals and alleviate pain in SMIR mice. Therefore, SMIR mice may prefer the lidocaine-paired chamber.

Reviewer point #8: SMIR does not produce significant changes in anxiety related or gross locomotor behaviors, as measured by open field test and elevated plus maze. The authors consistently call these tests "emotional" tasks, but really, they are anxiety related and should be indicated as such.

Author response #8: Thank you for the sincere opinions. We have made revisions to this issue in the manuscript. **We have modified “emotion” to “anxiety-like behaviors”**. The screen dump is provided below for your quick reference. Thank you once more for identifying these oversights.

between SMIR group and sham group (**Appendix Figure. S1D**). Given the
multidimensional nature of pain, we further evaluated **anxiety-like behaviors** in SMIR
mice. Both open field test (OFT) and elevated plus maze (EPM) analyses indicated that
SMIR surgery did not significantly affect **anxiety-like behaviors** in male mice (**Figure.**
**1C-1F**).[↵]
(**Figure. S1F**). Moreover, OFT and EPM analyses also showed a similar non-significant
tendency towards **anxiety-like behaviors** (**Figure. 1I-1L**). Collectively, these findings
demonstrate that SMIR surgery induces chronic postoperative pain, characterized
primarily by mechanical allodynia in both male and female mice, with subtle sex-
specific differences in temporal progression. Importantly, SMIR surgery has no
significant impact on **anxiety-like behaviors** or motor functions in either sex.[↵]
hind paws, from post-RNAi day 3 through day 10 (**Figure. 6H and 6I**). However,
**anxiety-like behaviors** and damaged locomotion were not observed in the OFT or EPM
The OFT was conducted to assess locomotor activity and **anxiety-like behaviors** in
mice. Following a 2-3 day acclimation period to the testing environment, mice were
**Anxiety-related behaviors** were further evaluated using the EPM test. The EPM
apparatus (Xinruan Information Technology Co., Ltd., XR-XG201) consisted of four

Reviewer point #9: It does seem that the experiments in I and K may be underpowered to conclude that no change takes place. Can the authors please provide a power analysis for panels I and J in figure 1?

Author response #9: Thanks very much for your professional suggestions. In this study, sample sizes for the open-field and elevated cross-maze experiments were set at $n = 8-12$ per group on the basis of previous reports (Gao et al., 2024; Yan et al., 2022).

Furthermore, based on the literature, we chose G-power software (Germany) to analyze the statistical power of the data of mice, during open field and elevated cross-maze experiments (Su et al., 2023). The sample size of the Sham group was equal to that of the SMIR group, with 12 individuals in each group. Type I error (α) is 0.05. We referred to the literature (Jeppesen et al., 2021) for the selection of the effect size. The study indicated that the effect size of anxiety-related clinical studies was 0.52 (moderate effect). However, the effect size of animal experiments is usually large (such as Cohen's $d > 1.0$). The reason is that animal experiments have highly controlled experimental environments and homogeneous individuals. Due to the 3R principle, animal experiments tend to choose large effect sizes to reduce sample size requirements. Therefore, we chose an effect size of 1-1.5. **The calculated power analysis is 0.65-0.94.**

Jeppesen et al., 2021

Table 2. Change From Baseline in Primary and Key Secondary Outcomes at 18 Weeks (ITT Population)^a

Outcome measure	MMM group		MAU group		Difference	P value	Effect size
	No. of participants	Measure	No. of participants	Measure			
Primary							
SDQ Impact scale score (parent-reported) ^b	197	-2.34 (0.13)	198	-1.23 (0.13)	-1.10 (-1.45 to -0.75)	<.001	-0.60
Key secondary							
Anxiety, SCAS score (parent-reported) ^c	197	-6.24 (0.66)	199	-1.34 (0.67)	-4.90 (-6.68 to -3.12)	<.001	-0.52
Depressive symptoms, MFQ score (parent-reported) ^d	197	-5.82 (0.48)	199	-2.72 (0.49)	-3.10 (-4.40 to -1.81)	<.001	-0.45
Level of daily functioning, WFIRS score (parent-reported) ^e	197	-7.56 (0.62)	199	-2.78 (0.64)	-4.78 (-6.47 to -3.10)	<.001	-0.54
School attendance (parent-reported) ^f	197	0.03 (0.01)	199	0.00 (0.01)	0.03 (0.01 to 0.06)	.009	0.26
Top-problem score (parent-reported)	197	-3.08 (0.14)	199	-1.37 (0.14)	-1.71 (-2.08 to -1.33)	<.001	-0.87
KIDSCREEN-27 score, mean (SD), t value (self-reported)^g							
Physical Well-being scale	197	3.08 (0.54)	199	2.56 (0.59)	0.52 (-1.01 to 2.04)	.51	0.06
Psychological Well-being scale	197	2.74 (0.49)	199	1.03 (0.54)	1.71 (0.33 to 3.09)	NC	0.24
Behavioral problems, ECBI (parent-reported)							
Intensity score ^h	197	-13.68 (0.97)	199	-6.47 (1.00)	-7.20 (-9.84 to -4.56)	NC	-0.52
Problem score ⁱ	197	-3.62 (0.27)	199	-2.30 (0.28)	-1.32 (-2.05 to -0.59)	NC	-0.34
Emotional and behavioral problems, SDQ Total Difficulties score (parent-reported) ^j	197	-4.07 (0.24)	198	-1.93 (0.25)	-2.14 (-2.79 to -1.48)	NC	-0.62
Responder indices, No. (%) (parent-reported)							
SDQ Impact scale score \geq 1-point reduction from baseline	197	144 (73.1)	199	93 (46.7)	3.15 (2.06 to 4.81)	NC	1.58
SDQ scores below inclusion cutoff ^k	197	98 (49.7)	199	56 (28.1)	2.59 (1.70 to 3.95)	NC	1.80

Abbreviations: ECBI, Eyberg Child Behavior Inventory; ITT, Intention-to-treat; MAU, management as usual; MFQ, Mood and Feelings Questionnaire; MMM, Mind My Mind; NC, not calculated; SCAS, Spence Children's Anxiety Scale; SDQ, Strengths and Difficulties Questionnaire; WFIRS, Weiss Functional Impairment Rating Scale.

^a Data presented as least-squares means (with standard error) unless otherwise stated. The differences between groups are the difference in least-squares means (95% CI) for the continuous outcomes and odds ratios (95% CI) for the dichotomous outcomes. The effect sizes are standardized mean differences for continuous outcomes and risk ratios for dichotomous outcomes.

^b Scores range from 0 to 10, with higher scores indicating greater severity of distress and impairment.

^c Scores range from 0 to 14, with higher scores indicating greater severity of anxiety.

^d Scores range from 0 to 68, with higher scores indicating greater severity of depressive symptoms.

^e Scores range from 0 to 100, with higher scores indicating more functional impairment.

^f Indicates percentage of school days in the last 4 weeks (range, 0-100).

^g Determined using health-related quality of life with 5 dimensions, of which we used the Physical Well-being and Psychological Well-being scales.

^h Scores range from 36 to 252, with higher scores indicating greater intensity of behavioral problems.

ⁱ Scores range from 0 to 36, with higher scores indicating more behavioral problems.

^j Scores range from 0 to 40, with higher scores indicating greater severity of general psychopathology.

^k Indicates parent-reported SDQ Total Difficulties score of at least 14, Emotional Problems score of at least 5, and/or Conduct Problems score of at least 3, combined with an SDQ Impact score of at least 1.

G-power

```
[1] -- Tuesday, July 15, 2025 -- 15:30:36
t tests -Means: Difference between two independent means (two groups)

Analysis: Post hoc: Compute achieved power
Input: Tail(s) = Two
      Effect size d = 1
      alpha err prob = 0.05
      Sample size group 1 = 12
      Sample size group 2 = 12
Output: Noncentrality parameter delta = 2.4494897
        Critical t = 2.0738731
        Df = 22
        Power (1-beta err prob) = 0.6486426
```

```
[2] -- Tuesday, July 15, 2025 -- 15:30:57
t tests -Means: Difference between two independent means (two groups)

Analysis: Post hoc: Compute achieved power
Input: Tail(s) = Two
      Effect size d = 1.5
      alpha err prob = 0.05
      Sample size group 1 = 12
      Sample size group 2 = 12
Output: Noncentrality parameter delta = 3.6742346
        Critical t = 2.0738731
        Df = 22
        Power (1-beta err prob) = 0.9393886
```

Furthermore, we have provided accurate P-values for V2-Figure 11-1K for review. We thank you again for this constructive suggestion.

Reference #9:

1. Gao Y, Gao D, Zhang H, Zheng D, Du J, Yuan C, et al. BLA DBS improves anxiety and fear by correcting weakened synaptic transmission from BLA to adBNST and CeL in a mouse model of foot shock. *Cell Rep.* 2024; 43: 113766.
2. Yan L, Wei JA, Yang F, Wang M, Wang S, Cheng T, et al. Physical Exercise Prevented Stress-Induced Anxiety via Improving Brain RNA Methylation. *Adv Sci (Weinh).* 2022; 9: e2105731.
3. Su Y, Xu L, Hu J, Musha J, Lin S. Meta-Analysis of Enhanced Recovery After Surgery Protocols for the Perioperative Management of Pediatric Colorectal Surgery. *J Pediatr Surg.* 2023; 58: 1686-93.
4. Jeppesen P, Wolf RT, Nielsen SM, Christensen R, Plessen KJ, Bilenberg N, et al. Effectiveness of Transdiagnostic Cognitive-Behavioral Psychotherapy Compared With Management as Usual for Youth With Common Mental Health Problems: A Randomized Clinical Trial. *JAMA Psychiatry.* 2021; 78: 250-60.

Reviewer point #10: Figure 2: Using western blot, immunohistochemistry, and electron microscopy, the authors demonstrate that autophagy is impaired in the ACC after SMIR. Additionally, using a viral approach, they attempt to demonstrate that LC3B colocalizes with excitatory neurons to a greater degree in SMIR mice.

Panels A through E are convincing evidence of their claims.

Author response #10: Thanks very much for your recognition of our research results.

Reviewer point #11: However, from panels F on, the reviewer would like to raise concerns. This issue will continue to arise throughout the paper.

The authors use a viral approach to track LC3B, an autophagy related protein that is tagged with mCherry. A major problem here is that the injection of the virus into the ACC is going to cause an injury to the cortex, which is going to alter the native state of the ACC, likely biasing it toward autophagy as the neurons heal following the injection. Therefore, these results are likely biased by the injection of virus into the ACC. The virus will be most concentrated in the injection track, specifically where the most injury will occur. The injection track is clearly visible, indicating some level of injury response (immune infiltration, among others). On top of this, the virus, and the viral overexpression of proteins, will likely increase the level of autophagy. A transgenic line for this protein exists (PMID: 18425446), so an approach that involves injury to the cortex was not necessary.

Author response #11: At the constructive suggestion of you, we learned that the GFP-LC3B transgenic mouse system (PubMed: Gt (ROSA) 26Sor) has been applied to visualize autophagy flux (Mareninova et al., 2020), which undoubtedly provides us with a powerful tool for subsequent experiments related to autophagy and CPOP.

In order to exclude the influence of EGFP-mCherry-LC3B reporting system on ACC, we designed an experiment to analyze the expression of LC3B in the ACC of the Scramble group and pLenti group in the naïve mice. **The results showed that compared with the Scramble group, there was no significant change in the expression of LC3B in the pLenti group.**

It is worth noting that you pointed out issues such as the virus being concentrated near the injection channel and the injection trajectory being clearly visible. In the EGFP-mCherry-LC3B reporting system, we used lentivirus for transfection. The transfection efficiency of lentivirus in the cerebral cortex is faster than that of adenovirus, but the transfection effect is relatively limited. Therefore, we speculate that these issues may be limited by the effectiveness of lentiviral transfection. We will replace it with AAV-EGFP-mCherry-LC3B in subsequent experiments. We thank you again for this constructive suggestion.

Reference #11:

1. Mareninova OA, Jia W, Gretler SR, Holthaus CL, Thomas DDH, Pimienta M, et al. Transgenic expression of GFP-LC3 perturbs autophagy in exocrine pancreas and acute pancreatitis responses in mice. *Autophagy*. 2020; 16: 2084-97.

Reviewer point #12: Figure 3: The authors then explore the use of bafilomycin A1 (BafA1) as an inhibitor of autophagy, and whether direct injection of BafA1 into the ACC produces a similar phenotype to SMIR.

The data are convincing that BafA1 inhibits autophagy (which was already known).

The data that BafA1 injection into the ACC produces a chronic pain phenotype are

also convincing.

Author response #12: We are pleased to receive recognition from you, which is the driving force for us to persevere.

Reviewer point #13: However, the reviewer has issue with panel K. The representative image clearly demonstrates a very high degree of overlap between GFP and mCherry (~90-100%). The images from panel 1H and 5G do not show such a high degree of overlap. Did the authors use different threshold levels to perform their quantifications specifically for this panel? I am highly skeptical about the results pertaining to Figure 3J-K due to major differences between representative images in different figures, but similar quantification percentages.

Author response #13: We greatly appreciate the professional suggestions provided by you. We have updated the new representative images (V2-Appendix Figure. S3K).

As shown in V1-Figure. 3J (V2-Appendix Figure. S3J), the EGFP-mCherry-LC3B reporting system was microinjected into the ACC at day 14 prior to surgery. The microinjection catheter was implanted at the same position at day 7 before surgery, but 0.1 mm higher than the injection point. The representative image we selected in V1-Figure. 3K is the ACC around the catheter. Due to the compression of brain tissue during trocar implantation and slow diffusion during BafA1 infusion, the virus around

the trocator accumulates more. This may be one of the reasons for the high overlap between EGFP-LC3B and mCherry-LC3B.

Furthermore, in the V1 version, we quantified the percentages of Merge-LC3B-positive area (autophagosomes) and total area in V1-Figures. 2H, 3K, and 5G. Although EGFP-LC3B overlaps significantly with mCherry-LC3B in V1-3K, the proportion of Merge-LC3B in the total area remains similar to that of V1-2H and V1-5G. We sincerely thank you again for your invaluable suggestions.

Reviewer point #14: Figure 4: The authors then use viral genetic tools to overexpress TMEM251 in the ACC, and demonstrate that overexpression blunts the maintenance of chronic pain after SMIR.

The data here are convincing. Although there remains the issue of potentially increased autophagy after viral injection in and of itself, the experiment is well controlled with scrambled control virus, and the lack of effect in Sham mice for both TMEM251 and scrambled overexpression viruses.

Author response #14: We would like to express our gratitude once again for your recognition of our results.

Reviewer point #15: Figure 5: TMEM251 knockdown using siRNA produces mechanical hypersensitivity.

The data here are convincing, with a few minor issues.

The quantification for panel 1A, the unit is inches. Instead, do they mean pixels? Microns? Or are these inches on the computer screen? If the later, this is not a useful measurement, and should be converted into microns.

Author response #15: Thank you for the professional suggestions. In order to further investigate the regulation of TMEM251 on CTSB sorting, we validated the distribution of CTSB in lysosomes after knocking down TMEM251 in vitro cell

experiments. We used ImageJ's Plot Profile tool to perform correlation analysis on the co-localization signals of CTSB and LAMP1 in cultured SH-SY5Y cells (Yu and Zhou, 2021).

In short, for RGB images captured by confocal microscopy, Split channels tool is used to separate each channel. It is divided into channel 1, where the CTSB is located, and channel 2, where LAMP1 is located. Then, linear tool is used to delineate the cell body of individual cells. The grayscale values of channel 1 and channel 2 within the delineated area were measured via the Plot Profile tool. The horizontal axis represents each pixel on the line, and the vertical axis represents the corresponding grayscale value of each point. Therefore, the image represents the trend of grayscale changes.

We referred to the horizontal axis (Distance inches, red box) of the original statistical chart during the statistics and did not convert the inches. **Therefore, we updated the correspondence between the inches and scale, and converted the units of the horizontal axis to μm (V2-Figure. 6A).** We sincerely thank you again for your

invaluable suggestion.

Reference #15:

1. Yu J, Zhou J. Vacuolar accumulation and colocalization is not a proper criterion for cytoplasmic soluble proteins undergoing selective autophagy. *Plant Signal Behav.* 2021; 16: 1932319.

Reviewer point #16: As the siRNA are delivered virally, would one not expect a more prolonged effect? It seems that mechanical hypersensitivity ends after 10 days.

Author response #16: Thank you for your valuable suggestions. RNAi technology directly delivers mature siRNA to the cytoplasm, which can rapidly degrade target mRNA (taking effect within hours to days), but the duration is relatively short (usually 3-7 days). In addition, shRNA is integrated into the host genome through vectors such as lentivirus and AAV, continuously transcribing to generate siRNA. The silencing effect can last for weeks to months, but the onset is slow.

Because in **V2-Figure. 1**, we observed that male mice exhibited pain hypersensitivity on POD1, accompanied by recovery of CPOP on POD21. **In order to better align with SMIR induced CPOP, we chose siRNA to intervene in TMEM251 in the ACC of wild-type mice.** It was effective on day 3 after RNAi intervention and lasted until the 10th day. We also designed an experiment and found that repeated injection of siRNA can prolong the mechanical allodynia of mice. We thank you again for this

constructive suggestion.

Reviewer point #17: Figure 6: Here, using calcium imaging, they take a neurophysiological approach to illuminate how stimulus evoked activity in the ACC is altered by SMIR and the role of TMEM251 in maintaining SMIR induced changes in nociceptive processing.

As expected, they find enhanced activation of the ACC in chronic pain (this is well known).

The overexpression of TMEM251 reduces SMIR-enhanced ACC stimulus evoked activity.

In the absence of chronic pain, knockdown of TMEM251 increases ACC evoked activity, however, the increase seems less than occurs during SMIR alone, indicating partial contribution to altered ACC nociceptive processing.

Direct administration of BafA1 increases evoked ACC activity.

Author response #17: Thank you for the professional comments. As you pointed out,

we aimed to establish a link between autophagy damage and neuronal activation.

Reviewer point #18: There is an issue with this experiment though, and which is a larger issue for the manuscript. BafA1 alters intracellular calcium dynamics (PMID: 25337829). Therefore, using calcium imaging is not an appropriate way to identify how BafA1 contributes to altered nociceptive processing via autophagy, as the change in evoked responses is likely due to changes in intracellular calcium buffering. This is clearly evident, as dF/F values for the BafA1 experiment are 2-fold higher (see the scale bar for Panel S and compare to those for all other panels in the figure). This experiment is poorly conceived and should not be included in the paper.

Author response #18: We greatly appreciate your professional advice. We have removed this part of the results from the manuscript (V1-Figure. 6Q-6U).

Reviewer point #19: However, the larger issue is whether the administration of BafA1 provides useful information specifically about autophagy, or if the results are more indicative of altered intracellular calcium buffering within the ACC.

Author response #19: Thank you very much for your constructive guidance. We fully understand your concern about the intracellular calcium imbalance induced by BafA1. The literature suggests that BafA1 blocking lysosomal uptake of Ca^{2+} increases the amplitude of Ca^{2+} signals induced by carbachol, but does not affect the rate of Ca^{2+} recycling to the endoplasmic reticulum (López Sanjurjo et al., 2014). Meanwhile, the endoplasmic reticulum and sarcoplasmic reticulum (ER/SR) are the core calcium stores within cells, storing the majority of calcium ions (Zheng et al., 2023). Hence, we speculate that the application of BafA1 may cause transient changes in intracellular Ca^{2+} , but may not cause long-term imbalances in intracellular calcium homeostasis.

Our data also indicated that the application of 10ng-BafA1 induces pain

hypersensitivity in naïve mice at day 3 after infusion (V2-Appendix Figure. S3C and S3D). Consistently, autophagy disorders also appeared at day 3 after infusion (V2-Appendix Figure. S3H). These results collectively suggest that the effect of BafA1 on pain of wild-type mice may be more due to the blockade of autophagy flux. We thank you again for this constructive suggestion.

Reference #19:

- López Sanjurjo CI, Tovey SC, Taylor CW. Rapid recycling of Ca²⁺ between IP3-sensitive stores and lysosomes. *PLoS One*. 2014; 9: e111275.
- Zheng S, Wang X, Zhao D, Liu H, Hu Y. Calcium homeostasis and cancer: insights from endoplasmic reticulum-centered organelle communications. *Trends Cell Biol*. 2023; 33: 312-23.

Reviewer point #20: Figure 7: Lastly, the authors replicate previous literature to show that decreasing bulk ACC activity can decrease pain behaviors, and then demonstrate that inhibiting ACC activity can reverse chronic pain phenotypes produced by the siRNA-mediated knockdown of TMEM251.

I'm not sure how informative this figure is. Decreasing bulk ACC activity is a well-known way to produce analgesia. Showing that decreasing ACC activity after TMEM251 knockdown or BafA1 infiltration produces analgesia does not necessarily prove that inhibition is working specifically to reverse the autophagy phenotype. As a thought experiment, ablating the ACC would have had a similar effect, but one would not conclude that ablations are useful for assessing the contribution of autophagy to

ACC pain processing.

Author response #20: Thank you very much for the constructive comments. We have transferred the experimental results of the chemogenetic inhibition of CaMKII-positive neurons in the siTMEM251 group and BafA1 group to V2-Appendix Figure. S7D-S7M. The revision of the manuscript is presented in Line 334-336 and the screen dump is provided below for your quick reference.

Figure. S7B and S7C). Similarly, chemogenetic inhibition of CaMKII α -positive
neurons significantly attenuated nociceptive hypersensitivity in siTMEM251 and
BafA1 mice (Appendix Figure. S7D-S7M). Amino-5-phosphonovaleric acid (AP-5)

In the preliminary design of this experiment, we aimed to alleviate pain hypersensitivity caused by autophagy disorders (siTMEM251 and BafA1) by chemogenetically inhibiting the activation of neurons, in order to further verify the link between autophagy disorders and neuronal activation. As suggested by you “Showing that decreasing ACC activity after TMEM251 knockdown or BafA1 infiltration produces analgesia does not necessarily prove that inhibition is working specifically to reverse the autophagy phenotype”, there is a logical weakness. Therefore, on the basis of your suggestion, we have added relevant experiments. Please refer to **Author response #21** for details.

Reviewer point #21: I think more useful here would be to see if altering neuronal activity directly within the ACC would produce changes in autophagy. For example, is the reason autophagy is dysregulated in chronic pain that ACC neurons fire more, and therefore proteins are degraded faster and have higher need to be cleared? If so, increasing ACC activity using excitatory DREADDs should increase autophagy demand (which would show up in their previous assays). Decreasing ACC activity may, through a similar mechanism, decrease the autophagy load.

Author response #21: We greatly appreciate the tremendous help and guidance you have provided. We strongly agree with your proposed modification plan. At your suggestion “Decreasing ACC activity may, through a similar mechanism, decrease the autophagy load”, **we have observed that chemogenetic inhibition of CaMKII-positive neurons significantly reduced the accumulation of SQSTM1 and LC3B in the ACC of SMIR mice (V2-Figure. 8E).** The revision of the manuscript is presented in Line 341-343 and the screen dump is provided below for your quick reference.

**Immunofluorescence analysis revealed that inhibition of activated CaMKII α -positive**
**neurons significantly attenuated SQSTM1 and LC3B accumulation in the ACC of**
**SMIR mice (Figure. 8E). To further investigate the association between CaMKII α -**

According to the comment “If so, increasing ACC activity using excitatory DREADDs should increase autophagy demand (which would show up in their previous assays)”. **Furthermore, the long-term application of excitatory DREADDs induced autophagy disorders and the accumulation of synaptophysin and PSD95 in autophagy substrates in the ACC (V2-Figure. 8F-8G).** The revision of the manuscript is presented in Line 343-360 and the screen dump is provided below for your quick reference.

SMIR mice (Figure. 8E). To further investigate the association between CaMKII α -
 positive neuron activation and autophagy dysregulation, we employed chemogenetic
 approaches to activate CaMKII α -positive neurons in naïve mice (Figure. 8F and 8G).
 CNO administration markedly increased the number of C-FOS-positive cells in the
 ACC (Figure. 8H). Notably, the CNO-treated group demonstrated significant SQSTM1
 accumulation in C-FOS-positive neurons (Figure. 8H). Similarly, the expression of
 both ATG5 and LC3B-II was significantly elevated in the ACC of CNO-3d group
 (Figure. 8I). Intriguingly, prolonged activation (CNO-7d group) resulted in additional
 upregulation of ATG7 and SQSTM1 expression (Figure. 8I). TEM analysis showed a
 significant increase in the number of autophagosomes in the ACC neurons of CNO
group (**Appendix Figure. S8A**). Immunofluorescence analysis further demonstrated
that sustained CaMKII α -positive neuron activation enhanced the accumulation of
synaptophysin and PSD95 in autophagy substrates, while gephyrin levels remained
unaffected (**Figure. 8J and Appendix Figure. S8B-S8C**). These findings collectively
suggest that chronic activation of CaMKII α -positive neurons elevates autophagy
activity, impairs autophagy flux, and promotes synaptic protein accumulation within
autophagy substrates, thereby establishing a link between neuronal hyperactivation and
autophagy dysfunction.↵

These findings are consistent with previous studies that CaMKII is a multifunctional serine/threonine protein kinase that regulates the initiation and progression of autophagy through multiple pathways. Research has shown that exercise can significantly activate neuronal autophagy in the hippocampal CA1 region through the Wnt5a/CaMKII signaling pathway, thereby enhancing synaptic plasticity and alleviating depression like behavior induced by high-fat diet (Wu et al., 2024). CaMKII can also directly phosphorylate the key autophagy initiating protein ULK1, enhancing autophagic activity (Liu et al., 2020). Therefore, we speculate that activation of CaMKII induces an increase in autophagy levels at the initiation of postoperative pain. As the chronic progression of postoperative pain, damaged autophagy induces the accumulation of excitatory synaptic proteins, exacerbating the excitation of CaMKII-positive neurons. **We have discussed these research results in the discussion section.** The revision of the manuscript is presented in Line 442-452 and the screen dump is provided below for your quick reference. We thank you again for this constructive suggestion.

activation of these neurons. Furthermore, the chemogenetic inhibition of activated
CaMKII α -positive neurons effectively alleviated mechanical allodynia, and the
accumulation of SQTM1 and LC3B in SMIR mice. Conversely, we observed that
chronic activation of CaMKII α -positive neurons in the ACC induced impaired
autophagy flux and accumulation of synaptic proteins in autophagy substrates. Previous
studies have suggested that CaMKII promotes autophagy through multiple signaling
pathways, as an important kinase [43, 44]. Therefore, we speculate that the activation
of CaMKII induces an increase in autophagy levels at the initiation of postoperative
pain. As the chronic progression of postoperative pain, damaged autophagy induces the
accumulation of excitatory synaptic proteins, exacerbating the excitation of CaMKII-
positive neurons. ←

Reference #21:

1. Wu, J., et al., 2024. Regular exercise ameliorates high-fat diet-induced depressive-like behaviors by activating hippocampal neuronal autophagy and enhancing synaptic plasticity. *Cell Death Dis.* 15, 737.
2. Liu, X., et al., 2020. Artesunate reverses LPS tolerance by promoting ULK1-mediated autophagy through interference with the CaMKII-IP3R-CaMKK β pathway. *Int Immunopharmacol.* 87, 106863.

Reviewer point #22: Line 37: Pharmacological should be pharmacogenetic is referring to the DREADD experiments.

Author response #22: Thanks very much for the valuable suggestions provided by you. **We have revised them accordingly.** The revision of the manuscript is presented in Line 39-40 and the screen dump is provided below for your quick reference.

while chemogenetic activation of these neurons conversely exacerbated autophagy
impairment. Our study highlights the close relationship among impaired autophagy,

Reviewer point #23: Line 102: I think CPP would be better referred to as a test of affective-motivational pain behaviors, and not necessarily spontaneous pain. Spontaneous pain usually refers to tests where specific pain behaviors are monitored after injury or injection of an algogen or pruritogen.

Author response #23: Thanks very much for the professional suggestions provided by you. **We have revised them accordingly.** The screen dump is provided below for your quick reference.

**Figure. S1C).** To assess affective-motivational pain behaviors, we conducted
conditioned place preference (CPP) testing at the peak of evoked pain response (POD7).
**(Figure. 1A and 1G).** Behavioral assessment using the CPP test at POD7 confirmed
the presence of affective-motivational pain behaviors in female mice **(Figure. 1H).** The

331 positive neurons effectively increased the ipsilateral PWT and reduced affective-
332 motivational pain behaviors in SMIR mice **(Figure. 8C-8D and Appendix Figure.**

550 **Affective-motivational pain behaviors** were assessed using CPP testing. Following a

Reviewer point #24: Line 109: The open field test and elevated plus maze are generally considered to be tests for anxiety, not general emotional tests. This same issue occurs in a few places throughout the manuscript.

Author response #24: We greatly appreciate your professional guidance. **We have revised them accordingly.** For specific details, please refer to **Author response #8.**

Reviewer point #25: The authors mention that lack of motor impairment due to SMIR model, but only point to open field data, which will give very limited information on motor impairment outside of the animals overall motivation to moved (distance traveled). Did the authors try placing the mouse on a rotarod assay to test coordination?

Author response #25: Thank you for the sincere feedback. **In accordance with your suggestion, we have added the rotarod test to evaluate the motor coordination of the mice.** The experimental results showed that SMIR surgery did not affect the motor coordination ability of the mice (V2-Appendix Figure. S1D and S1F). The screen dump is provided below for your quick reference.

**1B)**. Furthermore, the rotarod test was used to evaluate the motor ability of mice at
POD7. The results revealed that there was no significant difference in the latency to fall

5

between SMIR group and sham group (**Appendix Figure. S1D**). Given the
the presence of **affective-motivational pain** behaviors in female mice (**Figure. 1H**). The
rotarod tests revealed that SMIR surgery did not affect the latency to fall of female mice
(**Figure. S1F**). Moreover, OFT and EPM analyses also showed a similar non-significant
The rotarod test was performed to assess potential effects of SMIR surgery on motor
coordination of mice. Following established protocols [49], mice were acclimated to
the testing environment for 30 minutes prior to experimentation. Testing was conducted
using a rotarod apparatus (Xinruan Technology Co., Ltd., Shanghai, XR-6C). Mice
were initially acclimated to the rotating rod at a constant speed of 4 rpm. After
adaptation, the rotation speed was progressively increased from 4 to 40 rpm over a 5-
minute period, during which the latency to fall was recorded. Each mouse underwent
three trials with beyond 30-minute inter-trial intervals, and the average latency across
trials was calculated. Throughout testing, environmental conditions were maintained
with minimal noise, and the apparatus was kept clean and free of olfactory cues.↵

Reviewer point #26: Line 162-164: It is difficult to understand what is being said.

Author response #26: Thank you very much for your professional advice. We apologize for the inappropriate writing and **have made the necessary corrections**. The revision of the manuscript is presented in Line 161-163 and the screen dump is provided below for your quick reference.

During the maintenance of CPOP, the expression of SQSTM1 in the ACC of male
and female mice significantly increased (Figure. 2A and 2B), indicating that autophagy
flux may be impaired. We further performed immunofluorescence co-staining for

Reviewer point #27: Line 167-169: Male and female mice clearly have different behavioral responses to SMIR and autophagy related responses. I would be useful to provide data on female mice for some of their more important findings to ensure that their findings are not sex specific. Specifically, the TMEM251 overexpression and siRNA knockdown experiments, as related to their alteration of pain behaviors.

Author response #27: Thank you very much for your recognition and assistance. It is worth noting that an increasing number of studies have pointed out sex differences in pain, which may be due to biological factors such as hormone differences (Hellman et al., 2021) and sex-specific activation of immune cells (such as microglia) (Sorge et al., 2015). Previous studies have suggested that females may have a higher sensitivity to pain (Bartley and Fillingim, 2013; Lee et al., 2023), which is consistent with the results of this study. The potential mechanisms behind this are still worth further exploration. **We also discussed this viewpoint in the discussion section of the revised manuscript.** The revision of the manuscript is presented in Line 388-392 and the screen dump is provided below for your quick reference.

primary sensory neurons [26, 27]. Interestingly, the mechanical allodynia induced by
SMIR surgery lasted longer in female mice. Similarly, abundant evidence from
epidemiologic studies has demonstrated that women are at substantially greater risk for
many clinical pain conditions [28]. Potential biological mechanisms include hormonal
influences and sex-dependent immune cell activation [29]. Although SMIR mice

Reference #27:

1. Hellman KM, Oladosu FA, Garrison EF, Roth GE, Dillane KE, Tu FF. Circulating sex steroids and bladder pain sensitivity in dysmenorrhea. *Mol Pain*. 2021; 17: 17448069211035217.
2. Sorge RE, Mapplebeck JC, Rosen S, Beggs S, Taves S, Alexander JK, et al. Different immune cells mediate mechanical pain hypersensitivity in male and female mice. *Nat Neurosci*. 2015; 18: 1081-3.
3. Bartley EJ, Fillingim RB. Sex differences in pain: a brief review of clinical and experimental findings. *Br J Anaesth*. 2013; 111: 52-8
4. Lee SE, Greenough EK, Oancea P, Scheinfeld AR, Douglas AM, Gaudet AD. Sex Differences in Pain: Spinal Cord Injury in Female and Male Mice Elicits Behaviors Related to Neuropathic Pain. *J Neurotrauma*. 2023; 40: 833-44.

Reviewer point #28: Line 189: The authors mention "autophagy substrate", it would be helpful if this is more clearly defined. In fact, a graphical representation of all of the proteins involved and their relation to one another in the context of autophagy would considerably improve the manuscript.

Author response #28: Thank you very much for the constructive comments, which have greatly improved the quality of our manuscript. As we mentioned in **Author response #2**, we investigated the accumulation of synaptic related proteins in autophagy substrates, providing a foundation for enhancing synaptic function. **We have also made revisions in the abstract and discussion sections of manuscript.** The screen dump is provided below for your quick reference.

27 poorly understood. In this study, we demonstrated impaired autophagy in the anterior
28 cingulate cortex (ACC), especially in CaMKII α -positive neurons, during the
29 maintenance of CPOP. Notably, synaptic proteins accumulate in autophagy substrates.

367 positive neurons of the ACC, serves as a critical driver of CPOP progression. We
368 observed significant accumulation of synaptic proteins in autophagy substrates.

Given the established role of synaptic plasticity in chronic pain, we evaluated the
impact of neuronal autophagy damage on synapses in the ACC. The results indicated
that SMIR surgery upregulated the expression of excitatory synaptic proteins (PSD95
and VGLUT1), but did not affect inhibitory synaptic proteins (Gephyrin and VGAT).
Importantly, we identified significant accumulation of synaptophysin and PSD95 in
autophagy substrates of SMIR mice. These findings align with prior reports showing
that peripheral nerve injury enhances AMPA receptor accumulation and membrane
insertion in ACC layer V neurons, thereby facilitating nociceptive transmission [39, 40].

Furthermore, we have created graphical abstract, submitted as the appendix. We hope that it could help readers better understand this study. The figure is provided below for your quick reference. We thank you again for this constructive suggestion.

Legend: Potential mechanism of damaged autophagy involved in the chronicity of postoperative pain. A: When TMEM251 is present, CTSB can be correctly sorted into lysosomes, helping autophagy substrates including PSD95 to degrade normally and maintain pyramidal neuronal homeostasis. B: When TMEM251 is deficient, the lysosomal localization of CTSB decreases, impairing autophagy. Autophagy substrates, including PSD95, accumulate in pyramidal neurons, which induces the activation of pyramidal neurons. These are involved in the maintenance of postoperative pain. AMPA: A-amino-3-hydroxy-5-methyl-4-isoxazole-propionic acid receptor; CTSB: cathepsin B; NMDA: N-methyl-D-aspartic acid receptor; PSD95: postsynaptic density protein-95; TMEM251: transmembrane protein 251; VGLUT1: vesicular glutamate transporter 1.

Reviewer point #29: Line 290: The wording for this header is confusing, please rephrase.

Author response #29: We apologize to you for the confusing title. According to the suggestion in **Reviewer point # 3**, we have transferred the results of BafA1 to **V2-Appendix Figure. S3**. Meanwhile, the reduced expression of TMEM251 is presented as **V2-Figure. 4** in the main text. The result of overexpressing TMEM251 is presented as **V2-Figure. 5** in the main text. The revision of the manuscript is presented in Line 185-186/231-232 and the screen dump is provided below for your quick reference.

The decrease in hydrolases in lysosomes is accompanied by decreased TMEM251
expression in SMIR mice ↵
The overexpression of TMEM251 within the ACC alleviates impaired autophagy
and prevents chronicity of postoperative pain in SMIR mice ↵

Reviewer point #30: Line 584-586: After reading these lines, I was worried that this discussion was, at least partly, written by AI and not read over carefully. How does this paper in any way show how mechanical pressure sensitive channels in sensory neurons change? And how do they relate to temperature sensitive channels? I'm very concerned that this bit is in the manuscript and did not catch any of the authors' attention. In general, the discussion is difficult to read and should be rewritten and more closely edited.

Author response #30: We sincerely apologize for our incorrect expression and carelessness. **We are deeply aware of this and have rewritten the discussion section.** On the premise of accurately expressing ideas, we strive to ensure the clarity, coherence, and conciseness of sentences as much as possible. In addition, experts who are native English speakers were invited to make further revisions. Regarding the mechanical pressure sensitivity channel, we want to combine previous research to infer the potential mechanisms that the SMIR-induced CPOP model in mice mainly

exhibits mechanical allodynia rather than thermal hyperalgesia. This may be due to the mechanical force induced by SMIR surgery preferentially activating mechanical pressure-sensitive channels in peripheral primary sensory neurons, not the temperature-sensitive channels (Arora et al., 2021; Coste et al., 2010). It is worth noting that the details of the modifications in the discussion section can be reviewed in the manuscript. We sincerely thank you again for this constructive suggestion.

Reference #30:

1. Arora V, Campbell JN, Chung MK. Fight fire with fire: Neurobiology of capsaicin-induced analgesia for chronic pain. *Pharmacol Ther.* 2021; 220: 107743.
2. Coste B, Mathur J, Schmidt M, Earley TJ, Ranade S, Petrus MJ, et al. Piezo1 and Piezo2 are essential components of distinct mechanically activated cation channels. *Science.* 2010; 330: 55-60.

We would like to express our gratitude once again for your recognition of our manuscript that “The authors focus on neuronal autophagy as a contributor to chronic pain is novel, timely, and interesting” and “I believe the paper would be acceptable for publication after major revisions”. Furthermore, thank you very much for your seriousness and responsibility. You carefully reviewed each figure and paragraph, and proposed a comprehensive modification plan. We have tried our best to improve the manuscript and have made some changes in the updated manuscript that will not influence the content and framework of the paper. We hope that you will find this revised version satisfactory.

Sincerely,

The Authors

-----End of Reply to Reviewer 1-----

Reply to Reviewer 2

Reviewer point #1: In this report authors found that autophagy damage in the ACC after peripheral injury contribute to chronic postsurgical pain. While the study is potentially interesting, the report is too preliminary for publication in the present form. Additional experiments are needed to confirm key findings.

Author response #1: Thanks very much for the time and effort you have put into the manuscript. We fully appreciate your concern regarding the “preliminary nature” of some findings and the need for additional validation. We have supplemented relevant experiments based on your professional guidance and suggestions, which undoubtedly helped us greatly improve the manuscript. We have responded to your suggestion point-by-point and provided a revised version for your review.

We are aware of your concerns and take them seriously. In this study, we confirmed the role of the interaction between neuronal autophagy dysfunction and neuronal sensitization in the maintenance of postoperative pain through multidisciplinary approaches. Therefore, we would like to report this manuscript to provide a new perspective for the chronicity of postoperative pain and strong support for our future research.

In the next stage of research, we are focusing on exploring the potential mechanisms by which autophagy disorders induce neuronal sensitization. In addition to the synaptic effects which are supplemented under your professional guidance, we plan to explore the impact of autophagy disorders on neuronal inflammation. These are closely related to this manuscript, but also independent of it. Thank you again for your constructive feedback and detailed modification plan. We sincerely hope that under your professional guidance, the revised manuscript can reach the level required for publication in the journal.

Reviewer point #2: It is still unclear how these autophagy damage affects ACC excitability. Electrophysiological experiments are needed to determine if the impact is on excitatory vs inhibitory synaptic transmission, as well as possible presynaptic changes of circuits within the ACC

Author response #2: Thank you very much for taking the time to review our manuscript and providing constructive feedback on the key issues. Changes of synaptic plasticity in the ACC are the key factor in pain signal transmission (Bliss et al, 2016). Therefore, we fully agree with your suggestion that “Electrophysiological experiments are needed to determine if the impact is on excitatory vs inhibitory synaptic transmission, as well as possible presynaptic changes of circuits within the ACC”.

We would like to honestly explain for you that mature in vivo or ex vivo electrophysiological recording technology platforms are still being developed in our experimental team. We are currently unable to supplement the key electrophysiological experimental data you suggested during this modification cycle. We deeply regret this.

Although electrophysiological validation cannot be conducted immediately, we hope to provide other supportive evidence for our conclusions as much as possible through in-depth exploration of existing data and supplementary molecular biology experiments, in response to your concerns.

To assess whether impaired autophagy alters synaptic plasticity, we used excitatory synaptic-associated proteins (PSD95 and VGLUT1) (Chung et al., 2022) and inhibitory synaptic-associated proteins (Gephyrin and VGAT) (Kanjhan et al., 2016) to evaluate excitatory and inhibitory synaptic transmission, respectively. **The results indicated that SMIR surgery induced the expression of excitatory synaptic**

proteins (PSD95 and VGLUT1), but did not affect inhibitory synaptic proteins (Gephyrin and VGAT) (V2-Appendix Figure. S2E). Furthermore, we found that synaptophysin and PSD95 accumulated in autophagy substrates, not the Gephyrin (V2-Figure. 2E-2F and V2-Appendix Figure. S2G). The revision of the manuscript is presented in Line 157-164 and the screen dump is provided below for your quick reference.

in the SMIR group (Figure. 2D). Importantly, we observed a significant increase in the
 colocalization signal of LC3B and synaptophysin in the SMIR group (Figure. 2E).
 SMIR surgery increased the expression of postsynaptic density protein-95 (PSD95) and
 vesicular glutamate transporter 1 (VGLUT1) in the ACC, but did not affect the
 expression of Gephyrin and vesicular GABA transporter (VGAT) (Appendix Figure.
 S2E). Consistently, the colocalization signals of LC3B and PSD95, rather than those of
 Gephyrin, significantly increased in the SMIR group (Figure. 2F and Appendix
 Figure. S2G). ↵

In addition, overexpression of TMEM251 can alleviate the accumulation of synaptic proteins in autophagy substrates (V2-Figure. 5L and V2-Appendix Figure. S4E-S4G). The revision of the manuscript is presented in Line 256-260 and the screen dump is provided below for your quick reference.

increasing autolysosomes in ACC neurons (Figure. 5K and 5M). Western blot analysis
 also demonstrated that TMEM251 overexpression significantly reduced the expression
 of PSD95 and VGLUT1 in SMIR group (Appendix Figure. S4E). Furthermore, the
 overexpression of TMEM251 reduced the accumulation of synaptophysin and PSD95
 in autophagy substrates of SMIR mice (Figure. 5L and Appendix Figure. S4F-S4G).

However, knocking down TMEM251 can also observe similar results to the SMIR group (V2-Figure. 6J and V2-Appendix Figure. S5D-S5F). The revision of the manuscript is presented in Line 287-292 and the screen dump is provided below for your quick reference.

test (**Appendix Figure. S5B and S5C**). Western blot analysis demonstrated that
TMEM251 knockdown significantly increased the expression of PSD95 and VGLUT1
in naïve mice (**Appendix Figure. S5D**). Furthermore, RNAi-TMEM251 also induced
the accumulation of synaptophysin and PSD95 in autophagy substrates, which is
consistent with the findings in SMIR mice (**Figure. 6J and Appendix Figure. S5E-**
**S5F**). These findings collectively demonstrate that TMEM251-deficiency in the ACC

Previous literature has shown that PSD-95 directly positively regulates the amplitude of mEPSCs and indirectly affects their frequency by recruiting and anchoring AMPA receptors (El-Husseini et al., 2000). From these results, we speculate that the accumulation of excitatory synaptic proteins in autophagy substrates helps to enhance the transmission of excitatory synapses. **These findings align with prior reports showing that peripheral nerve injury enhances AMPA receptor accumulation and membrane insertion in ACC layer V neurons, thereby facilitating nociceptive transmission** (Chen et al., 2014a; Chen et al., 2014b). **We have added this viewpoint in the discussion section of manuscript.** The revision of the manuscript is presented in Line 416-425 and the screen dump is provided below for your quick reference.

networks, involving in CPOP maintenance. Changes of synaptic plasticity in the ACC
are the key factor in pain signal transmission [7]. Therefore, we evaluated the impact
of neuronal autophagy damage on synapses in the ACC. The results indicated that
SMIR surgery upregulated the expression of excitatory synaptic proteins (PSD95 and
VGLUT1), but did not affect inhibitory synaptic proteins (Gephyrin and VGAT).
Importantly, we identified significant accumulation of synaptophysin and PSD95 in
autophagy substrates of SMIR mice. These findings align with prior reports showing
that peripheral nerve injury enhances α -Amino-3-hydroxy-5-methyl-4-
isoxazolepropionic acid (AMPA) receptor accumulation and membrane insertion in
ACC layer V neurons, thereby facilitating nociceptive transmission [39, 40]. We also
7. Bliss TV, Collingridge GL, Kaang BK, Zhuo M. Synaptic plasticity in the anterior cingulate cortex
in acute and chronic pain. *Nat Rev Neurosci.* 2016; 17: 485-96.↵
39. Chen T, Wang W, Dong YL, Zhang MM, Wang J, Koga K, et al. Postsynaptic insertion of AMPA
receptor onto cortical pyramidal neurons in the anterior cingulate cortex after peripheral nerve injury.
*Mol Brain.* 2014; 7: 76.↵
40. Chen T, Koga K, Descalzi G, Qiu S, Wang J, Zhang LS, et al. Postsynaptic potentiation of
corticospinal projecting neurons in the anterior cingulate cortex after nerve injury. *Mol Pain.* 2014; 10:
33.↵

Furthermore, the microinfusion of excitatory synaptic transmission blockers (CNQX and AP-5) into the ACC of SMIR mice showed that both CNQX and AP-5 significantly increased ipsilateral PWT in SMIR mice at POD7 (**V2-Appendix Figure. S7N and S7O**), on the basis of the previous research (Ren et al., 2022). Combined with the above results, it suggests that autophagy damage may have a greater impact on excitatory synapses. The revision screen dump is provided below for your quick reference.

BafA1 mice (**Appendix Figure. S7D-S7M**). Amino-5-phosphonovaleric acid (AP-5)
and cyano-2,6-cyano-7-nitroquinoxaline-2,3-dione (CNQX) are important excitatory
synaptic transmission blockers. The behavioral data indicated that microinjection of
AP-5 or CNQX into the ACC significantly also alleviated mechanical allodynia of
SMIR mice at POD7 (**Appendix Figure. S7N and S7O**).[↵]
We also found that infusion of the excitatory synaptic transmission blockers (AP-5 and
CNQX) in the ACC significantly reduced mechanical allodynia in SMIR mice. These
results implicate excitatory synapse accumulation resulting from neuronal autophagy
impairment as a critical mechanism driving postoperative pain chronification. Previous

612 USA, HY-100558) was minimally infused into the ACC via a cannula. Amino-5-
613 phosphonovaleric acid (AP-5, 50 mmol/L, 0.4 μL, MCE Biotechnology Co., Ltd., New
614 Jersey, USA, HY-100714) or cyano-2,6-cyano-7nitroquinoxaline-2,3-dione (CNQX, 20
615 mol/L, 0.4 μL, MCE Biotechnology Co., Ltd., New Jersey, USA, HY-15066) was
616 infused into the ACC [50].[↵]

953 50. Ren D, Li JN, Qiu XT, Wan FP, Wu ZY, Fan BY, et al. Anterior Cingulate Cortex Mediates
954 Hyperalgesia and Anxiety Induced by Chronic Pancreatitis in Rats. *Neurosci Bull.* 2022; 38: 342-58.[↵]

As is well known, long-term potentiation (LTP) is an important indicator of synaptic plasticity. Previous studies have shown that there are two types of LTP in ACC, namely presynaptic LTP (pre-LTP) and postsynaptic LTP (post-LTP) (Koga et al., 2015). **Pre-LTP in the ACC plays an important role in chronic pain and anxiety-like behaviors** (Li et al., 2021). Therefore, further exploration of the impact of autophagy disorders on pre-LTP is necessary.

We believe that the electrophysiological experiment you proposed is not only a necessary step in revising this article, but also the core direction of our future research.

Therefore, we will take this research as the next key topic and actively cooperate with laboratories with well-established electrophysiological platforms to prepare for brain patch clamp recording, in order to accurately evaluate indicators such as sEPSC/sIPSC, mEPSC/mIPSC and LTP of ACC neurons, directly verifying the hypothesis of this article. **We have demonstrated this limitation in the discussion section of manuscript.** The revision of the manuscript is presented in Line 428-435 and the screen dump is provided below for your quick reference. We sincerely thank you again for this constructive suggestion.

critical mechanism driving postoperative pain chronification. Furthermore, previous
studies have identified two forms of long-term potentiation (LTP) in the ACC:
presynaptic LTP (pre-LTP) and postsynaptic LTP (post-LTP) [41]. Pre-LTP has been
shown to play a critical role in chronic pain and anxiety-like behaviors [42]. It is worth
noting that electrophysiological data is important for directly investigating the effects
of autophagy damage on synaptic transmission and pre-LTP in ACC neurons. Therefore,
the research on the electrophysiological characteristics of neuronal autophagy
impairment is worth exploring in depth.↵
41. Koga K, Descalzi G, Chen T, Ko HG, Lu J, Li S, et al. Coexistence of two forms of LTP in ACC
provides a synaptic mechanism for the interactions between anxiety and chronic pain. *Neuron*. 2015; 85:
377-89.↵
42. Li XH, Matsuura T, Xue M, Chen QY, Liu RH, Lu JS, et al. Oxytocin in the anterior cingulate
cortex attenuates neuropathic pain and emotional anxiety by inhibiting presynaptic long-term
potentiation. *Cell Rep*. 2021; 36: 109411.↵

Reference #2:

1. Bliss TV, Collingridge GL, Kaang BK, Zhuo M. Synaptic plasticity in the anterior cingulate cortex in acute and chronic pain. *Nat Rev Neurosci* 2016; 17: 485-496
2. Chung DW, Geramita MA, Lewis DA. Synaptic Variability and Cortical Gamma Oscillation Power in Schizophrenia. *Am J Psychiatry*. 2022; 179: 277-87.
3. Kanjhan R, Noakes PG, Bellingham MC. Emerging Roles of Filopodia and Dendritic Spines in Motoneuron Plasticity during Development and Disease.

- Neural Plast. 2016; 2016: 3423267.
4. El-Husseini AE, Schnell E, Chetkovich DM, Nicoll RA, Brecht DS. PSD-95 involvement in maturation of excitatory synapses. *Science*. 2000;290(5495):1364-1368.
 5. Chen T, Koga K, Descalzi G, Qiu S, Wang J, Zhang LS, et al. Postsynaptic potentiation of corticospinal projecting neurons in the anterior cingulate cortex after nerve injury. *Mol Pain*. 2014; 10: 33.
 6. Chen T, Wang W, Dong YL, Zhang MM, Wang J, Koga K, et al. Postsynaptic insertion of AMPA receptor onto cortical pyramidal neurons in the anterior cingulate cortex after peripheral nerve injury. *Mol Brain*. 2014; 7: 76.
 7. Ren D, Li JN, Qiu XT, Wan FP, Wu ZY, Fan BY, et al. Anterior Cingulate Cortex Mediates Hyperalgesia and Anxiety Induced by Chronic Pancreatitis in Rats. *Neurosci Bull*. 2022; 38: 342-58.
 8. Koga K, Descalzi G, Chen T, Ko HG, Lu J, Li S, et al. Coexistence of two forms of LTP in ACC provides a synaptic mechanism for the interactions between anxiety and chronic pain. *Neuron*. 2015; 85: 377-89.
 9. Li XH, Matsuura T, Xue M, Chen QY, Liu RH, Lu JS, et al. Oxytocin in the anterior cingulate cortex attenuates neuropathic pain and emotional anxiety by inhibiting presynaptic long-term potentiation. *Cell Rep*. 2021; 36: 109411.

Reviewer point #3: ACC deep neurons may project to subcortical areas as well as spinal dorsal horn neurons (Chen et al., 2018, Nature Communication). Authors need to perform detailed analyses if superficial layers of cells (II/III) and deep projection cells are all affected by autophagy damage. Additional discussion of related to ACC projected network in chronic postsurgical pain should be added.

Author response #3: Thanks very much for your valuable guidance. Based on your feedback and the anatomical stratification of ACC in the literature (Chen et al., 2014a), we evaluated the expression of LC3B in different sublayers of the ACC. We found that the increased LC3B in the SMIR group was mainly concentrated in layers III, V,

and VI of the ACC (V2-Figure. 2D). The revision of the manuscript is presented in Line 151-153 and the screen dump is provided below for your quick reference.

151 autophagy structures (Figure. 2C). Furthermore, immunofluorescence analysis
152 revealed that increased LC3B was mainly concentrated in layers III, V, and VI of ACC
153 in the SMIR group (Figure. 2D). Importantly, we observed a significant increase in the

Previous study has indicated that deep-layer ACC neurons form extensive projection networks with subcortical regions including the spinal cord, critically contributing to chronic pain pathophysiology (Chen et al., 2018). These data suggest that autophagy impairment in the ACC may activate ACC-related top-down neural networks, involving in CPOP maintenance. We have addressed this point in the discussion section of the revised manuscript. The revision of the manuscript is presented in Line 416-422 and the screen dump is provided below for your quick reference. We sincerely thank you again for this constructive suggestion.

dimensions of pain and associated emotional responses [36, 37]. The potential role of
ACC in CPOP still needs further exploration. In the present study, we observed that the
autophagy alterations in the ACC were primarily localized to layers III, V, and VI.
Previous study has indicated that deep-layer ACC neurons form extensive projection
networks with subcortical regions including the spinal cord, critically contributing to
chronic pain [38]. These data suggest that autophagy impairment in the ACC may
activate ACC-related top-down neural networks, involving in CPOP maintenance.
38. Chen T, Taniguchi W, Chen QY, Tozaki-Saitoh H, Song Q, Liu RH, et al. Top-down descending
facilitation of spinal sensory excitatory transmission from the anterior cingulate cortex. Nat Commun.
2018; 9: 1886.↵

Reference #3:

1. Chen T, Koga K, Descalzi G, Qiu S, Wang J, Zhang LS, et al. Postsynaptic potentiation of corticospinal projecting neurons in the anterior cingulate cortex after nerve injury. Mol Pain. 2014; 10: 33.
2. Chen T, Taniguchi W, Chen QY, Tozaki-Saitoh H, Song Q, Liu RH, et al. Top-down descending facilitation of spinal sensory excitatory transmission from the anterior cingulate cortex. Nat Commun. 2018; 9: 1886.

Reviewer point #4: Insular cortex plays important roles in pain modulation and perception. Some comparison or additional data from the IC would increase impact of the study.

Author response #4: Thank you very much for the constructive guidance. The insular cortex (IC) integrates pain-related emotional and proprioceptive information, bridging physiological states with subjective pain perception (Li et al., 2024; Zhang et al., 2022). **We have supplemented the autophagy changes in IC of male and female mice.** The results showed that SMIR surgery increased the expression of ATG7 and ATG5 in the IC of male mice (**V2-Appendix Figure. S2B**). Meanwhile, SMIR surgery increased the expression of ATG7, ATG5, and LC3B-II in the IC of female mice (**V2-Appendix Figure. S2D**). Overall, SMIR surgery increased autophagy activity in the IC without causing autophagy damage. The revision of the manuscript is presented in Line 136-138 and the screen dump is provided below for your quick

reference.

136 To investigate the role of autophagy in CPOP, we analyzed key cortical and subcortical
137 brain regions, including the somatosensory cortex (S1), **insular cortex (IC)**, nucleus
138 accumbens (NAc), ACC, hippocampus (Hip), and basolateral amygdala (BLA).

Furthermore, we added a comparison between IC and ACC in the discussion section of the revised manuscript. The revision of the manuscript is presented in Line 395-398 and the screen dump is provided below for your quick reference. We sincerely thank you again for this constructive suggestion.

The cortical regions play important roles in nociceptive signal processing and sorting.
The insular cortex, for example, integrates pain-related emotional and proprioceptive

18

information [30, 31]. By contrast, the ACC is involved in processing the affective
dimensions of pain and associated emotional responses [32, 33]. Autophagy plays an

894 30. Li Y, Li C, Chen QY, Hao S, Mao J, Zhang W, et al. Alleviation of migraine related pain and anxiety

by inhibiting calcium-stimulating AC1-dependent CGRP in the insula of adult rats. *J Headache Pain.*
2024; 25: 81.↵
31. Zhang MM, Geng AQ, Chen K, Wang J, Wang P, Qiu XT, et al. Glutamatergic synapses from the
insular cortex to the basolateral amygdala encode observational pain. *Neuron.* 2022; 110: 1993-2008.e6.↵

Reference #4:

1. Li Y, Li C, Chen QY, Hao S, Mao J, Zhang W, et al. Alleviation of migraine related pain and anxiety by inhibiting calcium-stimulating AC1-dependent CGRP in the insula of adult rats. *J Headache Pain.* 2024; 25: 81.
2. Zhang MM, Geng AQ, Chen K, Wang J, Wang P, Qiu XT, et al. Glutamatergic synapses from the insular cortex to the basolateral amygdala encode observational pain. *Neuron.* 2022; 110: 1993-2008 e6.

We would like to take this opportunity to thank you for all your time involved and this great opportunity for us to improve the manuscript. The complete plan you proposed has increased the depth and breadth of the research. We express our deep gratitude once again for this. In addition, we are well aware that these additional molecular experiments cannot be completely equated with direct evidence from electrophysiology, but we hope that the above efforts can demonstrate our serious attitude towards the review comments, and enhance the completeness and

persuasiveness of this study as much as possible. We also sincerely hope that these responses and revised manuscripts would satisfy you.

Sincerely,

The Authors

----- **End of Reply to Reviewer 2** -----

Dear Dr. Zhang

Thank you for the submission of your revised manuscript to EMBO reports. We have now received the full set of referee reports that is copied below.

As you will see, both referees are very positive about the study and recommend publication. Before I can proceed with the official acceptance of your study, I need you to address some minor points below:

- 1) Please update the 'Conflict of interest' paragraph to our new 'Disclosure and competing interests statement'. For more information see <https://www.embopress.org/page/journal/14693178/authorguide#conflictsofinterest>
- 2) Regarding the Author Contributions, we now use CRediT to specify the contributions of each author in the journal submission system. Therefore, please remove the Author Contributions from the manuscript file and make sure that the author contributions in our online manuscript tracking system are correct and up-to-date. The information you specified in the system will be automatically retrieved and typeset into the article. You can enter additional information in the free text box provided, if you wish.
- 3) The references need to be alphabetical, not numerical; et al needs to be used after 10 author names.
- 4) Please move the reference to BioRender to the Methods section as follows:
Graphics:
(some of the... OR Figure #... OR synopsis) Graphics were created with BioRender.com.
- 5) The information on funding needs to be part of Acknowledgments; the Funding section heading is not needed.
- 6) Reagent and Tools table: Please remove the short Instructions paragraph on the first page and the Example table on page 3.
- 7) Materials and methods should be Methods.
- 8) Methods, header "Cell" could be changed to "Cell culture" or "Cell lines"?
- 9) Ethical approval and participation consent should be part of the Methods section.
- 10) The Abbreviations section needs to be removed from the manuscript. Abbreviations should be defined in brackets after their first mention in the text, not in a list of abbreviations.
- 11) Please remove the Supplemental information section from the manuscript.
- 12) Your methods section contains references to "supplementary methods" or "Supplemental Material". All materials and methods must be part of the main methods section in the manuscript. That said, I could not locate a file with supplementary methods, so the callouts might be outdated? Please check and update.
- 13) Appendix:
 - The Appendix PDF needs a title page with a table of content and page numbers.
 - Appendix Figure S1: you describe statistical tests used in the figure legend, but the panels do not display any p-values or 'ns'. Please check whether the information in the legend is needed. This also applies to several other figure panels in the Appendix.
 - In cases where p-values are shown, please provide the exact p-values in the figure legend (or panel) unless the p-value is smaller than 0.0001.
 - Appendix Fig. S2G, S5F: the zoomed image needs a scale bar and the scale bar present is difficult to see.
 - Appendix Fig. S6A: please define that white dashed lines in the legend.
 - Appendix Fig. S6H: please define the white arrowheads.
 - Appendix Fig. S8A: please define the red arrows.
- 14) Figure legends:
 - Please address the following points in the main figure legends:
 - Please note that the exact p values are not provided in the legends of figures 1A, B, G, H; 2A, B, D, E, F; 3A, D, E, H, I; 4A, B, C, D, E; 5D, E, F, G, I, J, L, M; 6B, C, F, G, H, I, J; 7A, B, E, F, O, R; 8C, E, H, I, J.
 - Please note that the dotted border is not defined in the legend of figure 3G, 7L, 8G. This needs to be rectified.
 - Please note that the arrow heads are not defined in the legend of figure 3J, C. This needs to be rectified.
 - Please note that the white arrows are not defined in the legend of figure 4E, F; 7A, B; . This needs to be rectified.
- 15) Some of the figures, like Fig. 5, are quite data-rich. Might it make sense to split Fig. 5 in two separate figures?

16) Please provide the synopsis image as a separate file (jpeg, TIFF or png format: 550 pixels wide x 200-600 pixels high).

17) Source data: Please upload the source data for each figure separately. We need one folder per figure, inside the folder we need separate files/subfolders, one per each panel. This also applies to .xls files. I think it is OK to leave the Western blot images as they are, i.e., combined on one PDF per figure panel, but the imaging data need to be provided as raw data with individual images for each channel (or a composite that can be opened by e.g., Image J or Fiji).

18) As a standard procedure, we edit the title and abstract of manuscripts to make them more accessible to a general readership. Please find the suggested versions below my signature. Please check these for scientific accuracy and edit is needed.

With kind regards,

=====

Referee #1:

The authors have done substantial work to address my concerns, which is much appreciated.

Referee #2:

Authors have performed additional experiments to address my most of concern. I agree that some electrophysiological experiments can be performed in another study. Thus, I recommend its publication in the present form.

=====

TMEM251 loss-induced autophagy dysfunction in the anterior cingulate cortex contributes to chronic postoperative pain
Macroautophagy/autophagy plays a crucial role in maintaining nervous system homeostasis but its role in chronic postoperative pain (CPOP) remains poorly understood. Here, we identify impaired autophagy and the accumulation of synaptic proteins in the anterior cingulate cortex (ACC) during the maintenance of CPOP after skin/muscle incision and retraction (SMIR). Lysosomal hydrolase levels are reduced upon SMIR, accompanied by a deficiency of the lysosomal trafficking protein transmembrane protein 251 (TMEM251, also named LYSET). TMEM251 overexpression alleviates impaired autophagy, accumulation of synaptic proteins within autophagy substrates, and maintenance of CPOP in SMIR mice. Conversely, TMEM251 knockdown induces autophagy impairment, accumulation of synaptic proteins, and chronic pain phenotypes in naïve mice. Autophagy dysfunction is most pronounced in CaMKII α -positive neurons in the ACC post-surgery, resulting in their activation, which is mitigated by TMEM251 overexpression. Chemogenetic activation of CaMKII α neurons exacerbates autophagy impairment and CPOP, while their inhibition rescues SMIR-induced autophagy and pain phenotypes. Taken together, our study highlights the close relationship between impaired autophagy and neuronal activation in the promotion of chronic postoperative pain.

All editorial and formatting issues were resolved by the authors.

Dr. Wei Zhang
The First Affiliated Hospital of Zhengzhou University
Department of Anesthesiology, Pain and Perioperative Medicine
Zhengzhou, Henan 450001
China

Dear Dr. Zhang,

I am very pleased to accept your manuscript for publication in the next available issue of EMBO reports. Thank you for your contribution to our journal.

Yours sincerely,
